# Why Do We Need Warm-up? A Theoretical Perspective

**Foivos Alimisis** [* 1]   **Rustem Islamov** [* 1]   **Aurelien Lucchi** [1]

## Abstract

Learning rate warm-up – increasing the learning rate at the beginning of training – has become a ubiquitous heuristic in modern deep learning, yet its theoretical foundations remain poorly understood. In this work, we provide a principled explanation for why warm-up improves training. We rely on a generalization of the $(L_0, L_1)$-smoothness condition, which bounds local curvature as a linear function of the loss suboptimality and exhibits desirable closure properties. We show – both theoretically and empirically – that this condition is satisfied by common neural architectures and accurately captures the curvature of the optimization landscape early in training. Adapting the learning rate in response to this curvature condition naturally induces a warm-up–like schedule, and we show that this choice yields provably faster convergence guarantees than using a fixed learning rate. Experiments on language and vision models show that the resulting one-parameter warm-up schedule can match tuned linear warm-up and improve over no warm-up.

## 1. Introduction

A common learning-rate (LR) schedule linearly increases the LR at the beginning of training, *warm-up stage* (Goyal et al., 2017; Vaswani et al., 2017), and gradually decreases it near the end, *decay stage* (Loshchilov & Hutter, 2017; Vaswani et al., 2017; Hoffmann et al., 2022b; Zhang et al., 2023; Dremov et al., 2025).

Decaying the LR is a classical requirement in the theoretical analysis of SGD, ensuring convergence under broad conditions (Defazio et al., 2023; Gower et al., 2021), and it has been consistently observed to improve empirical per-

formance (Loshchilov & Hutter, 2017; Hu et al., 2024; Hägele et al., 2024). Recent work further demonstrates that *decaying* step-sizes can improve theoretical guarantees by yielding tighter bounds (Schaipp et al., 2025). By contrast, the practice of linearly increasing the LR at the start of training (warm-up phase) has become nearly ubiquitous in modern deep learning (He et al., 2016; Hu et al., 2024; Hägele et al., 2024), yet a clear theoretical understanding of why it helps optimization remains elusive. A growing body of empirical work points to several advantages of warm-up, including: $(i)$ mitigating training instabilities (Kosson et al., 2024; Goyal et al., 2017; Zhang et al., 2023), reducing the variance of stochastic gradients (Liu et al., 2019), and improving the robustness to the choice of the peak LR (Wortsman et al., 2024; Kalra & Barkeshli, 2024). However, these explanations remain fragmented and likely point to secondary benefits rather than the core mechanism. This raises the central question we address in this paper:

*Is there an intrinsic property of neural network loss landscapes that justifies the use of a learning-rate warm-up schedule?*

We identify such a property and make the following contributions:

1. We introduce a generalized smoothness condition, $(H_0, H_1)$-smoothness, in which local curvature is bounded by an affine function of the loss suboptimality. This condition enjoys useful closure properties under finite sums and affine transformations and generalizes $(L_0, L_1)$-smoothness.

2. We provide both theoretical and empirical evidence that the $(H_0, H_1)$-smoothness condition holds for common neural network architectures, under both mean-squared error (MSE) and cross-entropy (CE) losses. During the initial phase of training, $(H_0, H_1)$-smoothness realistically captures the loss-curvature dynamics.

3. We theoretically demonstrate that, for functions that satisfy the $(H_0, H_1)$ condition, Gradient Descent (GD) achieves faster convergence with a warm-up schedule than with a fixed LR. We prove this by deriving upper complexity bounds for GD with the adaptive schedule and lower complexity bounds for fixed-step GD.

---

*Equal contribution [1]University of Basel, Switzerland. Correspondence to: Foivos Alimisis <foivos.alimisis@unibas.ch>, Rustem Islamov <rustem.islamov@unibas.ch>.

*Proceedings of the 43rd International Conference on Machine Learning*, Seoul, South Korea. PMLR 306, 2026. Copyright 2026 by the author(s).

4. We provide empirical evidence that the resulting tuned one-parameter schedule matches the performance of tuned linear warm-up on language and vision models, and improves upon training without warm-up.

## 2. Related Works

**Warm-up.** LR scheduling plays a central role in the success of modern deep learning training pipelines. A wide range of scheduling strategies, including LR decay, annealing, and warm-up, have been developed to improve convergence and generalization (McCandlish et al., 2018; Sutskever et al., 2013; Touvron et al., 2023).

Among these different strategies, warm-up has become a key component in modern training pipelines, particularly for Transformers (Vaswani et al., 2017; Goyal et al., 2017). It is commonly credited with enhancing training stability (Kosson et al., 2024; Gotmare et al., 2019), improving robustness to the choice of LR (Wortsman et al., 2024), and enabling the use of larger peak LR (Kalra & Barkeshli, 2024). Warm-up has also been linked to improved generalization, either by reducing mini-batch gradient noise (Liu et al., 2019), encouraging convergence to flatter minima (Smith et al., 2020), or by complementing other scheduling techniques (Huang et al., 2020; Xiong et al., 2020; Wortsman et al., 2024). From a geometric perspective, Gilmer et al. (2022); Roulet et al. (2024) observed that warm-up induces a curvature reduction phase in which the largest Hessian eigenvalue decreases.

Although warm-up is well supported by empirical evidence (Vaswani et al., 2017; Wortsman et al., 2024; Dremov et al., 2025), its theoretical foundations remain limited. Most existing convergence analyses of (stochastic) gradient-based optimizers focus on the decay phase. For example, Wen et al. (2025) uses a river-valley model to study neural loss landscapes, but their framework focuses on the stable and decay stages of the LR. Likewise, Schaipp et al. (2025); Attia & Koren (2025) showed that decaying LR provides theoretical benefits and that convergence bounds closely align with empirical training curves, yet their analysis does not account for the warm-up phase. Kondo & Iiduka (2025) analyze a scheme with exponentially increasing batch size and LR, showing faster convergence for (GD). Yet, the requirement of rapidly growing batches limits its practicality.

Finally, several complementary explanations for the role of warm-up have been proposed. For instance, Xiong et al. (2020) attribute the necessity of warm-up in Transformer training primarily to the placement of layer normalization. In a different vein, Kosson et al. (2024) demonstrate that explicitly constraining the norm of parameter updates—similar to gradient clipping—can only partially reduce the reliance on warm-up.

Despite extensive prior research on warm-up, we are not aware of any theoretical framework that explains its benefits in terms of convergence. In this work, we address this gap by relying on a smoothness-type condition that upper bounds the curvature of the landscape using an affine expression of the function suboptimality. This condition represents well the loss landscape at the initial stage of training. For this function class, the curvature-adaptive step-size naturally increases as the loss decreases, giving a warm-up mechanism.

**Curvature dynamics.** The distinct phases of curvature evolution observed during neural network training have been studied extensively. Cohen et al. (2021) showed empirically that curvature typically increases over the course of training until it reaches a learning-rate–dependent threshold. Beyond this point, gradient descent often enters the so-called *edge-of-stability* regime, in which curvature fluctuates around this threshold. These progressive sharpening and edge-of-stability phases are now relatively well understood within the deep learning community. More recent work has identified an additional *initial* phase that precedes progressive sharpening and is characterized by a reduction in curvature (Kalra et al., 2023; Kalra & Barkeshli, 2024). This behavior is consistent with both our theoretical analysis and our experimental observations. Unlike the later phases, however, this early curvature-reduction regime remains poorly understood and is likely the phase in which learning-rate warm-up plays a critical role.

**Generalized Smoothness.** The conventional smoothness assumption in optimization theory requires the Hessian to satisfy a uniform bound $\|\nabla^2 f(w)\| \leq L$, but this constraint proves to be overly restrictive when applied to neural network training, as noted by Zhang et al. (2020b). To address this limitation, they introduced the more flexible $(L_0, L_1)$-smoothness condition, which allows the Hessian norm to grow linearly with the gradient magnitude: $\|\nabla^2 f(w)\| \leq L_0 + L_1 \|\nabla f(w)\|$ for non-negative constants $L_0, L_1 \geq 0$. This relaxed framework naturally motivates gradient normalization techniques—both soft normalization and hard clipping—as optimal LR strategies that can significantly improve gradient descent convergence rates (Zhang et al., 2020a; Zhao et al., 2021; Faw et al., 2023; Wang et al., 2023; Gorbunov et al., 2025; Vankov et al., 2025; Li et al., 2023).

Despite its advantages, the $(L_0, L_1)$-smoothness condition has important limitations in practice, especially in explaining warm-up schedules. More precisely, at the beginning of training, the gradient-dependent nature of the $(L_0, L_1)$-smoothness condition leads to counterintuitive implications for LR scheduling: in some cases, the gradient norm is observed to increase during the early iterations (Xie et al.,

2023; Defazio et al., 2023; Defazio, 2025). As a result, the $(L_0, L_1)$-bound becomes increasingly loose, which theoretically prescribes *decreasing* step-sizes through gradient clipping. This stands in direct contrast to empirical best practices, where *increasing* LR is typically employed at the beginning of training.

These theoretical and practical inconsistencies highlight the need for a more sophisticated smoothness characterization that can adequately capture and explain LR warm-up dynamics. Since dependence on the gradient norm can lead to the wrong qualitative step-size prediction during early training, a natural replacement is the function-value suboptimality, which decays monotonically and gives a direct measure of the optimization target. We name this modified smoothness class as $(H_0, H_1)$-smoothness. Interestingly, a recent work by Vaswani & Harikandeh (2025) made a similar observation in a different context, showing that Armijo line search can achieve faster convergence than `GD` with a constant step-size. Their analysis verifies this condition for several simple models but relies on additional assumptions from Taheri & Thrampoulidis (2023): $(i)$ bounding the gradient norm by the function suboptimality, $(ii)$ adopting the unrealistic exponential loss, $(iii)$ assuming data separability, and $(iv)$ restricting trainability to the input layer. In contrast, our analysis establishes the validity of the $(H_0, H_1)$-smoothness condition under a mild regularity assumption on the weights, which can be ensured either explicitly through balancedness or implicitly via standard L2 regularization. Our work proposes a curvature-based condition tailored to the early sharpness-reduction phase and uses it to justify warm-up as a stability-speed tradeoff.[1]

## 3. The $(H_0, H_1)$-Smoothness Condition

As an initial proof of concept, we analyze the *derivative of the spectral norm of the Hessian* of a simple $1 \times 1 \times 1$ network with linear activation $\phi(x) = x$:

$$f(u, v) = (y - uvx)^2$$

over the course of the gradient flow ODE, assuming that $x, y \neq 0$. For this model, we provide the following result.

**Proposition 3.1.** *During the trajectory* $(u(t), v(t))$ *of gradient flow starting sufficiently close to* $(0, 0)$*, the Hessian* $H(t)$ *of the loss function* $f(u(t), v(t)) = (y - u(t)v(t)x)^2$*, satisfies*

*(i)* $\frac{d}{dt}\|H(t)\|_2 < 0$*, if* $\frac{x^2(u(0)-v(0))^2}{4} \leq (y - 2u(t)v(t)x)^2$*.*

*(ii)* $\frac{d}{dt}\|H(t)\|_2 > 0$*, when* $y < 2u(t)v(t)x$*.*

The proof can be found in Appendix C. Proposition 3.1 re-

---

[1] A concurrent work (Liu et al., 2025) studies a warm-up stage using a similar condition. We discuss the differences in Appendix A.

veals two distinct phases of curvature evolution under initialization close to $0$, as is standard in practice. During the initial phase of training, condition $(i)$ holds, leading to curvature reduction. This behavior has also been observed in prior work (Kalra et al., 2023; Kalra & Barkeshli, 2024), though it remains poorly understood. As training progresses, gradient flow drives the parameters into a regime in which condition $(ii)$ is satisfied, resulting in progressive sharpening. This phenomenon was identified early in the study of the edge of stability (Cohen et al., 2021) and has attracted significant attention in recent years. There are three remarks worth highlighting about this proposition:

1. If $u(0) = v(0)$, the two phases are continuous, and the transition from curvature reduction to progressive sharpening occurs precisely when $u(t)v(t) = y/2$.

2. The switching behavior predicted by this simple model closely matches our empirical observations in language model training (see Figure 2).

3. The proposition does not capture later training stages, such as the edge of stability, as it relies on gradient flow dynamics. These regimes can be characterized more precisely using alternative approaches, such as the central flow framework of Cohen et al. (2025), which is beyond the scope of this work.

These observations motivate us to focus on an earlier curvature reduction phase, which has been largely overlooked but is closely tied to LR warm-up. We characterize this early stage by relating curvature to function value suboptimality, as summarized in the following definition.

**Definition 3.1.** *A function* $f: \mathbb{R}^d \rightarrow \mathbb{R}$ *with minimum* $f^* > -\infty$ *is called* $(H_0, H_1)$*-smooth for some* $H_0, H_1 \geq 0$*, if for any* $w \in \mathbb{R}^d$ *it holds*

$$\|\nabla^2 f(w)\|_2 \leq H_0 + H_1(f(w) - f^*).$$

$\mathcal{H} \coloneqq \{f: \mathbb{R}^d \rightarrow \mathbb{R} \mid f \text{ is } (H_0, H_1)\text{-smooth}\}$ *denotes the class of all* $(H_0, H_1)$*-smooth functions.*

Notably, any $(L_0, L_1)$-smooth function also satisfies $(H_0, H_1)$-smoothness, implying that the $(H_0, H_1)$-smooth class contains the previously studied $(L_0, L_1)$-smooth class, demonstrating the generality of Definition 3.1. Moreover, we show that $\mathcal{H}$ is closed under finite sums and affine transformations, in contrast to the $(L_0, L_1)$-smooth class, for which neither operation is preserved, as demonstrated by simple counterexamples. Formal statements and proofs are deferred to Appendix B. Finally, Definition 3.1 admits a natural extension in which the linear dependence on the suboptimality $f(w) - f^*$ is replaced by $\mathcal{L}(f(w) - f^*)$, where $\mathcal{L}$ is a general increasing function, in the spirit of Li et al. (2023). We leave the study of this extension to future work.

### 3.1. Theoretical Justification of $(H_0, H_1)$-Smoothness

In this section, we show that under mild regularity conditions on the weights, enforced either explicitly via constraints or implicitly through $\ell_2$ regularization, the $(H_0, H_1)$-smoothness condition holds for several basic deep learning architectures.

#### 3.1.1. STANDARD FEEDFORWARD NETWORKS

We begin with vanilla feedforward networks and distinguish two regimes: $(i)$ the initial phase of training and $(ii)$ the global regime. We focus on the initial phase, where both sharpness and loss suboptimality decay, making our bounds predictive. Detailed proofs are in Appendix D.

**Results under Balancedness.** A well-known property of gradient flow in feedforward neural networks is the balanced evolution of the weight matrices $\{W_i\}_{i=1}^\ell$. For linear networks, this implies $W_i(t)^\top W_i(t) = W_{i+1}(t)W_{i+1}(t)^\top$, while for networks with homogeneous nonlinear activations it yields $\|W_i(t)\|_{\mathrm{F}} = \|W_{i+1}(t)\|_{\mathrm{F}}$ (Du et al., 2018, Theorem 2.2 and Corollary 2.1). The latter condition is weaker than the former.

**Proposition 3.2.** *Consider a deep linear network with $\ell$ layers and MSE loss:*

$$f(W) \equiv f(W_1, \ldots, W_\ell) = \|Y - W_1 W_2 \ldots W_\ell X\|_{\mathrm{F}}^2,$$

*where $Y \in \mathbb{R}^{c \times m}$ are the labels, $X \in \mathbb{R}^{d \times m} (d \leq m)$ is the input, and $W_i \in \mathbb{R}^{n_{i-1} \times n_i}$, where $n_0 = c$ and $n_\ell = d$ are layer dimensions. In the weakly balanced subspace, $\|W_1\|_F = \|W_2\|_F$, we have:*

$(i)$ *If $\|W_1\|_{\mathrm{F}}^\ell \|X\|_2 < (1 - \frac{1}{\ell})\sqrt{f(W)}$, then it holds*

$$\|\nabla^2 f(W)\| \leq H_0^{warm-up} + H_1^{warm-up}(f(W) - f^*)$$

*with constants defined in Eq. (7).*

$(ii)$ *If we additionally assume strong balancedness, then it holds*

$$\|\nabla^2 f(W)\| \leq H_0 + H_1(f(W) - f^*) \ \text{ for all } W,$$

*with constants defined in Eq. (13).*

We note that condition $(i)$ is consistent with the early phase of training, where weight norms are typically small while the loss remains large, especially under commonly used near-zero initializations. Under standard conditions on the data matrices, $H_0^{\text{warm-up}}, H_1^{\text{warm-up}}$ are smaller than the global constants $H_0, H_1$ respectively. In Proposition D.1, we generalize this result to a non-linear network with leaky-ReLU non-linearities.

**Results under L2 Regularization.** Analogous to balancedness, another approach to constrain the weight space is through L2 regularization. Here, we present a result that validates the $(H_0, H_1)$-smoothness condition for two-layer neural networks with general activation functions and cross-entropy loss. Its proof can be found in Appendix D.

**Proposition 3.3.** *Consider a 2-layer non-linear model with cross-entropy loss and L2 regularization:*

$$f(W) \equiv f(W_1, W_2) = $$
$$- Y \log(P)^\top - (\mathbb{1} - Y) \log(\mathbb{1} - P)^\top$$
$$+ \frac{\lambda_1}{2}\|W_1\|_{\mathrm{F}}^2 + \frac{\lambda_2}{2}\|W_2\|_{\mathrm{F}}^2,$$

*where $Y \in \mathbb{R}^{1 \times m}$ are true labels, and $P = \sigma(W_1 \phi(W_2 X))$ is the output of the model with the activation function $\phi$ such that $|\phi(s)| \leq C_1|s|, |\phi'(s)| \leq C_2$ and $|\phi''(s)| \leq C_3$ for all $s \in \mathbb{R}$, sigmoid function $\sigma$, and weight matrices $W_1 \in \mathbb{R}^{1 \times n_1}, W_2 \in \mathbb{R}^{n_1 \times d}$. Then, it holds*

$(i)$ *If $\|W_1\|_F, \|W_2\|_F$ are sufficiently small compared to the loss value in the sense of Eq. (29), then $(H_0, H_1)$-smoothness holds with constants defined as in Eq. (30).*

$(ii)$ *For all $W$, $(H_0, H_1)$-smoothness holds for constants $H_0$ and $H_1$ defined as in Eq. (31).*

Similarly to Proposition 3.2, the bound derived in case (i) is sharper than the one derived in case (ii). This is natural, as case (i) concerns only the initial slice of the landscape.

Notably, we show in Appendix F that $(L_0, L_1)$-smoothness fails to hold even for simple two-layer networks under $\ell_2$ regularization or weight balancedness.

#### 3.1.2. TRANSFORMERS

In this section, we study a simple transformer architecture with a single attention layer trained under $\ell_2$ regularization, following the setup of Zhang et al. (2024). We defer the description of the model to Appendix E and state only an informal version of the main result here: the corresponding in-context loss function is $(H_0, H_1)$-smooth. The proof of this result can be found in Appendix E (see formal version in Prop. E.2).

**Proposition 3.4.** *(informal) Consider a transformer model for in-context learning in continuous variables with MSE loss, or in binary variables with CE loss. Under mild regularity conditions on the distributions of the input and output spaces, it holds*

$$\|\nabla^2 f(P, Q)\|_2 \leq H_0 + H_1(f(P, Q) - f^*),$$

*where $f$ is the in-context loss function, $f^*$ the minimum, $P, Q$ are parameters of the model, and $H_0$ and $H_1$ are some positive constants. When $\|P\|_F$ and $\|Q\|_F$ are small*

*compared to the loss value, one can sharpen $H_0$ and $H_1$ similarly to Propositions 3.2 and 3.3.*

### 3.2. Empirical Justification of $(H_0, H_1)$-Smoothness

We next turn to verifying the proposed condition in practical settings. Specifically, we examine Transformer-based language models with 70M, 160M, and 410M parameters using the NanoGPT implementation (Radford et al., 2019; Karpathy, 2022). Experiments are conducted on the FineWeb dataset (Penedo et al., 2024) using SGD with gradient clipping at 1 and a small constant learning rate of $10^{-4}$. Using such a conservative LR and gradient clipping allows the optimizer to progress slowly, thereby probing the landscape around initialization in more detail. As a stochastic proxy for local smoothness at iteration $k$, we compute

$$\frac{\|\nabla f_{S_k}(w_{k+1}) - \nabla f_{S_{k-1}}(w_k)\|}{\|w_{k+1} - w_k\|},$$

where $S_k$ denotes the mini-batch at iteration $k$, following prior work (Zhang et al., 2020b; Riabinin et al., 2026). As shown in Figure 1, the estimated smoothness decays approximately linearly, indicating that the proposed condition provides a reasonable smoothness approximation for real-world models.

To provide a more complete picture of smoothness dynamics over the full training trajectory and the role of initialization, we train a 70M language model using larger constant learning rates and two different initialization schemes. The results are shown in Figure 2 . We observe an initial curvature reduction phase, previously identified in (Kalra & Barkeshli, 2024; Kalra et al., 2023), followed by progressive sharpening and the edge of stability phases (Cohen et al., 2021).

Notably, the initial sharpness is higher when the initialization is scaled with depth, a common practice in language model training (Sun et al., 2025; Zhang et al., 2019). We hypothesize that this effect may partly explain the need for learning rate warm-up in Transformer training, and defer a more detailed discussion of initialization effects to Appendix K.2. Moreover, the transition from curvature reduction to progressive sharpening consistently occurs at the same loss value, independent of the learning rate and initialization, further highlighting the link between sharpness and loss. The initial curvature reduction phase accounts for approximately 20% of the total training loss reduction.

We next turn to image classification on ImageNet-32 (Chrabaszcz et al., 2017), training both ResNet50 (He et al., 2016) and ViT-Tiny (Dosovitskiy et al., 2021). The results, shown in Figure 3, indicate that a linear function provides a good approximation of the relationship between local smoothness and training loss. Compared to language

models, however, the points are more widely dispersed and have larger variance. Taken together, Figures 1 and 3 support the view that $(H_0, H_1)$-smoothness offers a reasonable approximation of smoothness in early training.

## 4. Theoretical Analysis under $(H_0, H_1)$-Smoothness

We study the minimization problem $\min_w f(w)$, which appears in various machine learning applications. Here $w \in \mathbb{R}^d$ denotes parameters of some model, $d$ is the number of parameters, and $f$ is the loss that measures the performance. We define $f^* := \min_w f(w) > -\infty$ as the optimal loss. The set $\mathcal{S}$ contains all global minimizers of the objective $f$. Importantly, our theoretical results apply to the regime in which Definition 3.1 holds, which in our setting is most relevant during the early sharpness-reduction phase of neural network training. The proofs of this section are deferred to Appendix H and J.

### 4.1. Notation and Assumptions

We conduct our analysis for well-known classes of non-convex functions, presented below.

**Definition 4.1** (Liu et al. (2023)). *A function $h$ satisfies the Aiming condition with a constant $\theta > 0$ around the set $\mathcal{X}$, if $\langle \nabla h(w), w - \pi_{\mathcal{X}}(w) \rangle \geq \theta(h(w) - h^*)$ holds for all $w \in \mathbb{R}^d$. Here, $\pi_{\mathcal{X}}(w)$ is the projection of $w$ onto the set $\mathcal{X}$, and $h^* := \min_{w \in \mathbb{R}^d} h(w)$.*

**Definition 4.2** (Polyak (1963)). *A function $h$ satisfies Polyak-Łojasiewicz (PL) condition with a constant $\mu > 0$, if $\|\nabla h(w)\|^2 \geq 2\mu(h(w) - h^*)$ holds for all $w \in \mathbb{R}^d$.*

### 4.2. Lower Bounds and Convergence of GD with Constant Step-size

To enable a meaningful comparison between the step-size schedule suggested by the $(H_0, H_1)$-condition and an alternative fixed step-size strategy, we derive lower complexity bounds for the latter. This approach follows the idea of Theorem 4 in (Zhang et al., 2020b): we first consider a rapidly growing function and show that, for GD to converge, the step-size must be sufficiently small. Next, we examine a slowly growing function and demonstrate that this previously derived step-size constraint leads to slow convergence of the algorithm. The complete proof can be found in Appendix J.

**Theorem 4.1.** *Let $f$ belong to the class $\mathcal{H}$ of $(H_0, H_1)$-smooth functions. Then it holds:*

1. *To satisfy $\|\nabla f(w_K)\| \leq \varepsilon$ for a general non-convex function $f$, GD with constant step-size initialized at*

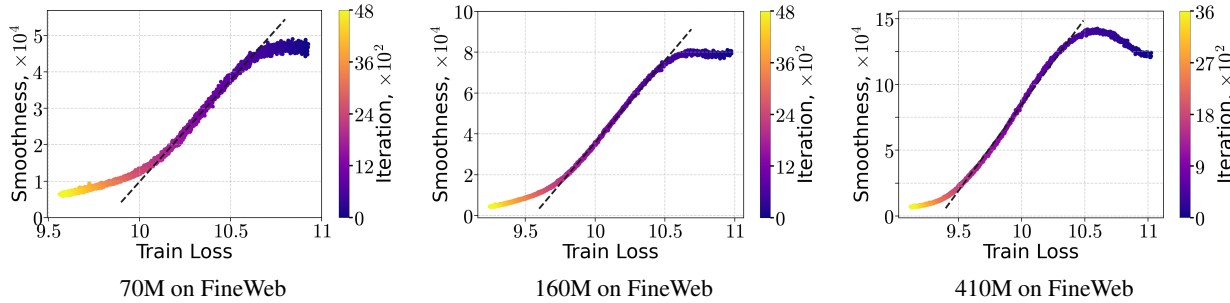

*Figure 1.* Local smoothness approximation versus training loss for language models of varying sizes on the FineWeb dataset, using `SGD` with gradient clipping 1 with a constant LR of $10^{-4}$. Each dot represents estimated local smoothness and stochastic training loss, with color indicating training progress, while the black dashed line shows the best linear fit. For much of early training, the relation is well-approximated by a line.

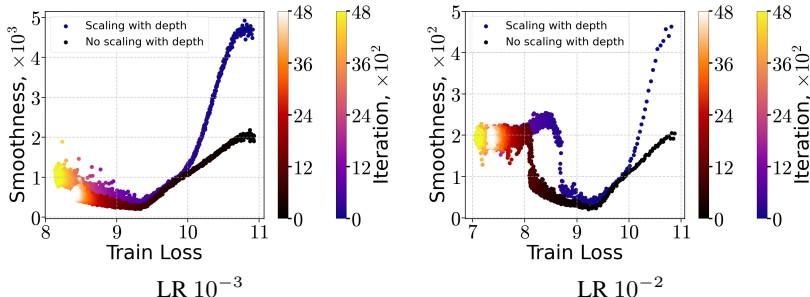

*Figure 2.* Training of a 70M model on FineWeb dataset with `SGD` with gradient clipping to 1, varying fixed learning rate and initialization scheme. Each left color bar corresponds to initialization around 0 with variance that depends on the depth of the model (as is usually done in modern transformers), while the right one to initialization around 0 with fixed variance.

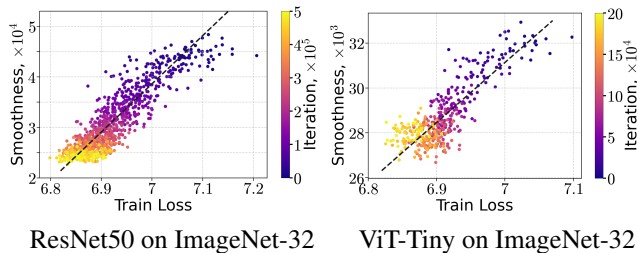

ResNet50 on ImageNet-32    ViT-Tiny on ImageNet-32

*Figure 3.* Stochastic local-smoothness proxy versus training loss for ResNet50 (left) and ViT-Tiny (right) on ImageNet-32, trained with `SGD` and a constant LR $10^{-4}$.

$w_0$, *needs at least*

$$K \geq \frac{H_1(f(w_0) - f^*)}{\log(f(w_0) - f^*) + 1} \frac{f(w_0) - f^* - 2\epsilon^2}{8\epsilon^2}$$

*iterations.*

2. *To satisfy $f(w_K) - f^* \leq \varepsilon$ for convex function $f$,* `GD` *with constant step-size initialized at $w_0$, needs at least*

$$K \geq \frac{H_1(f(w_0) - f^*)}{\log(f(w_0) - f^*) + 1} \frac{f(w_0) - f^* - \epsilon}{4\epsilon}$$

*iterations.*

3. *To satisfy $f(w_K) - f^* \leq \varepsilon$ for $\mu$-PL function $f$ (but not necessarily convex),* `GD` *with constant step-size initialized at $w_0$, needs at least*

$$K \geq \frac{H_1}{4\mu} \frac{(f(w_0) - f^*)}{\log(f(w_0) - f^*) + 1} \log\left(\frac{f(w_0) - f^*}{\epsilon}\right)$$

*iterations.*

This result covers the one in (Zhang et al., 2020b) as a special case, and it also covers convex and $\mu$-PL functions. These lower bounds concern fixed step-sizes chosen to be stable over the relevant $(H_0, H_1)$-smooth class; the initialization dependence appears because such a step-size must account for the worst early curvature of a non-uniformly smooth landscape. We show next that, for $(H_0, H_1)$-smooth functions, a gradually increasing step-size schedule can accelerate convergence while maintaining training stability.

### 4.3. Convergence of GD with Adaptive Warm-up

Next, we turn to the analysis of `GD` under Assumption 3.1 with an adaptive step-size of the form

$$\eta_k := \frac{1}{10H_0 + 20H_1(f(w_k) - f^*)} \tag{1}$$

prescribed by $(H_0, H_1)$-smoothness. The constants $H_0, H_1, f^*$ are unknown in practical training; Section 5 therefore uses a one-parameter surrogate in which a threshold C controls when the loss-dependent denominator becomes active. Since the function suboptimality decreases at the beginning of training, the theoretical step-size follows a warm-up-like scheme. In the general non-convex case, the derived upper bound in Theorem H.1 provides only numerical (non-asymptotic) improvement over a constant schedule.

To achieve tangible improvements, additional convexity-like assumptions are necessary. This is justified by the fact that the loss landscape of neural networks exhibits additional structure; prior studies indicate that neural network loss surfaces often display a convex-like geometry locally (Kleinberg et al., 2018; Guille-Escuret et al., 2024; Islamov et al., 2024; Tran et al., 2024). This observation has motivated relaxations of convexity, such as the aiming condition (Liu et al., 2023) and quasar-convexity (Hardt et al., 2018), which have been leveraged in the analysis of various gradient-based algorithms (Gower et al., 2021; Hinder et al., 2020; Fu et al., 2023). Importantly, these conditions are satisfied by certain classes of non-convex functions (Hardt et al., 2018; Liu et al., 2023).

**Theorem 4.2.** *Assume that $f$ is $(H_0, H_1)$-smooth, and it satisfies the Aiming condition with constant $\theta$ around the set of global minimizers $\mathcal{S}$. Then the iterates of GD with adaptive step-size $\theta \cdot \eta_k$ satisfy*

$f(w_K) - f^* \leq \varepsilon$ *after at most*

$$\frac{40H_0\text{dist}(w_0, \mathcal{S})^2}{\theta^2\varepsilon} + \frac{40H_1\text{dist}(w_0, \mathcal{S})^2}{\theta^2} \quad \text{iterations.}$$

To analyze convergence under increasing step-size, we split the iterations into two regimes corresponding to large and small function values and study them separately. The convex rate is recovered by setting $\theta = 1$. Notably, the $1/\varepsilon$ term depends only on $H_0$, as in standard convex GD theory, while $H_1$ affects only the constant term. Comparing the bounds in Theorem 4.2 and Theorem 4.1, we find that *GD with a warm-up adaptive step-size converges faster than with a fixed step-size when the initialization factor $H_1(f(w_0) - f^*)/\varepsilon$ is large*, that is, when the initial loss is large. This factor can be substantial, potentially even exponential in $H_1\text{dist}(w_0, \mathcal{S})$ (Gaash et al., 2026).

Next, we consider another widely studied class of structured non-convex functions, which encompasses the $\mu$-PL functions–known to hold for sufficiently over-parameterized networks (Liu et al., 2022). Moreover, PL is considered the weakest sufficient condition ensuring linear convergence of GD (Karimi et al., 2016).

**Theorem 4.3.** *Assume that $f$ is $(H_0, H_1)$-smooth, and it satisfies $\mu$-PL condition. Then the iterates of GD with adap-*

*tive step-size $\eta_k$ satisfy*

$f(w_K) - f^* \leq \varepsilon$ *after at most*

$$\frac{40H_1}{\mu}(f(w_0) - f^*) + \frac{20H_0}{\mu}\log\frac{H_0}{2H_1\varepsilon} \quad \text{iterations.}$$

Similar to the convex case, the $\varepsilon$-dependent term in GD with a warm-up adaptive step-size leads to faster convergence whenever $H_1(f(w_0) - f^*)$ is substantially larger than $H_0$.

In Appendix A, we show that our proof techniques in both Theorems 4.2 and 4.3 extend to a more general class of functions, where the function suboptimality in Definition 3.1 is raised to the power $\rho \geq 1$, extending the benefits of the theoretical warm-up to a broader class of functions.

### 4.4. Extension to the Stochastic Setting

In a standard training setup, the function $f$ has a finite sum structure, namely,

$$f(w) := \frac{1}{n}\sum_{i=1}^{n} f_i(w) \tag{P}$$

where $n$ is the size of the training dataset, and each $f_i$ represents a loss on $i$-th sample. We define the minimum of each loss $f_i^* = \min_w f_i(w)$. To study the convergence in the stochastic setting, we need an interpolation condition, which is typically satisfied for over-parameterized networks (Ma et al., 2018). Analytically, it means that $f^* = f_i^*$ for all $i \in [n]$.

**Theorem 4.4.** *Assume that the problem (P) satisfies the interpolation condition. Assume that each $f_i$ is $(H_0, H_1)$-smooth and satisfies the Aiming condition around the set of global minimizers $\mathcal{S}$. Then the iterates of SGD $w_{k+1} = w_k - \eta_k\nabla f_{S_k}(w_k)$ with batch $S_k \subseteq [n]$ and a step-size*

$$\eta_k = \frac{\theta}{10H_0 + 20H_1(f_{S_k}(w_k) - f_{S_k}^*)}$$

*satisfy*

$$\frac{1}{K+1}\sum_{k=0}^{K} \mathbb{E}\left[\min\left\{f(w_k) - f^*, \frac{H_0}{2nH_1}\right\}\right]$$

$$\leq \frac{20H_0\text{dist}(w_0, \mathcal{S})^2}{\theta^2(K+1)}.$$

We observe that the convergence rate depends on $H_0$, mirroring the deterministic result in Theorem 4.2. The convergence metric we use is non-standard, adopted because uniform convergence over all component functions $\{f_i\}_{i=1}^{n}$ cannot be ensured. Equivalently, the probability that all iterates remain outside the small-error region $f(w_k) - f^* \leq \frac{H_0}{2nH_1}$ is at most $\frac{40nH_0\text{dist}(w_0,\mathcal{S})^2}{\theta^2(K+1)}$ the suboptimality $f(w_k) - f^*$ can be larger than $\frac{H_0}{2nH_1}$ for any

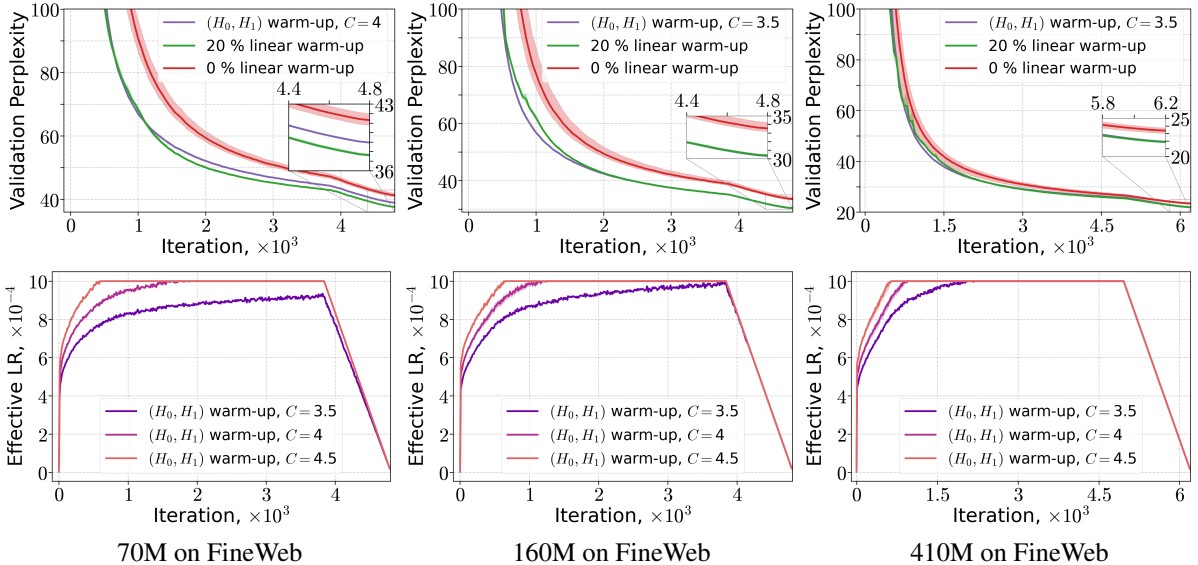

*Figure 4.* **Top row:** validation perplexity for 70M, 160M, and 410M language models on FineWeb under no warm-up, tuned linear warm-up, and $(H_0, H_1)$ warm-up. **Bottom row:** effective LR induced by $(H_0, H_1)$ warm-up for peak LR $10^{-3}$ and different values of C. All runs use a final 20% linear decay to $10^{-5}$.

$k \in \{0, \dots, K\}$. Thus, this failure probability vanishes as $K \to \infty$, implying convergence after a sufficiently large number of iterations with high probability.

As discussed earlier, our guarantees are predictive when the $(H_0, H_1)$ condition captures early sharpness reduction. Our theory shows that SGD with an increasing step-size stabilizes by adapting to local curvature, whereas a fixed step-size must be unrealistically small under poor initialization.

**Remark 4.1.** *Interestingly, both Aiming (Liu et al., 2023) and PL (Liu et al., 2022) conditions hold at least locally around the initial point of training. This matches well the study of warm-up, which concerns the initial phase of training.*

## 5. Experiments

We next evaluate the warm-up schedule derived from $(H_0, H_1)$-smoothness on two benchmarks: transformer language modeling on FineWeb and ViT-Tiny training on ImageNet-32. In both settings, warm-up is standard and empirically beneficial, as confirmed by our experiments. This section aims to highlight the merits of warm-up, particularly the gains obtained from the $(H_0, H_1)$-smoothness–driven schedule rather than to achieve state-of-the-art performance. To test the theoretically motivated warm-up schedule, we compare no warm-up and tuned linear warm-up against the following $(H_0, H_1)$-condition-based schedule:

$$\frac{\eta_k}{\max\{1, f_{S_k}(w_k)/C\}}.$$

Here $f_{S_k}(w_k)$ is the stochastic loss at iteration $k$, and $C$ is a tunable threshold that controls when the denominator becomes active and therefore determines the effective warm-up length. Here $\eta_k$ follows the WSD schedule (language modeling) or cosine annealing (ViT training) with no warm-up. All training details are reported in Appendix K.

**Language Modeling.** We train language models of three sizes: 70M, 160M, and 410M near Chinchilla optimum (Hoffmann et al., 2022a). For the 70M and 160M models, we use Adam with a WSD schedule, a 20% decay phase, and a tuned linear warm-up length in $\{0\%, 10\%, 20\%\}$. For the 410M model, we use AdamW with weight decay $\lambda = 0.1$ (Loshchilov & Hutter, 2019). We report the mean of 3 runs, with the shaded area showing the min–max range.

In this setup, $\eta_k$ in $(H_0, H_1)$ warm-up follows the WSD schedule without warm-up (i.e., the LR starts directly at its peak) with a 20% decay phase. This can be viewed as a hard counterpart of the theoretical step-size considered in our convergence analysis. We tune the parameter $C$, which determines the length of the $(H_0, H_1)$ warm-up, over the set $\{3.5, 4, 4.5\}$ (which was found to yield good results empirically). For all warm-up schedules, we tune the peak LR over $\{3 \cdot 10^{-4}, 10^{-3}, 3 \cdot 10^{-3}, 10^{-2}\}$. Figure 4 shows that the theoretically motivated $(H_0, H_1)$ warm-up performs competitively with linear warm-up, which is the standard choice in practice, and both warm-up schedules improve over training without warm-up. Figure 4, bottom row, plots the effective LR. Unlike linear warm-up, the $(H_0, H_1)$-based schedule increases nonlinearly because it

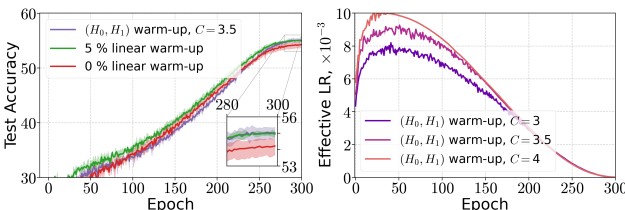

*Figure 5.* Test accuracy and effective LR for ViT-Tiny on ImageNet-32 with `AdamW` ($\lambda = 0.05$) under no warm-up, tuned linear warm-up, and $(H_0, H_1)$ warm-up. All schedules use cosine decay after the warm-up phase.

is driven by the loss trajectory.

**Image Classification.** Next, we repeat the study on ViT-Tiny using cosine annealing for $\eta_k$ (replacing WSD) while keeping the same warm-up mechanisms. For the $(H_0, H_1)$ warm-up, we sweep $C \in \{3, 3.5, 4\}$; for linear warm-up, we vary the warm-up length in $\{0\%, 5\%, 10\%\}$. For each schedule, we grid-search the peak LR over $\{3 \cdot 10^{-4}, 10^{-3}, 3 \cdot 10^{-3}, 10^{-2}, 3 \cdot 10^{-2}\}$. As in the previous setting, Figure 5 shows that $(H_0, H_1)$ warm-up matches linear warm-up, and both outperform training with no warm-up. The right panel of Figure 5 plots the effective LR, showing again that the loss-driven warm-up profile is nonlinear. Similar to the previous case, the warm-up substantially differs from the linear warm-up. We report the mean of three runs, with the shaded area showing the min–max range.

## 6. Conclusion, Limitations, and Future Work

We proposed $(H_0, H_1)$-smoothness as a curvature–loss model for the early sharpness-reduction phase of neural-network training. The condition yields a curvature-adaptive step-size that increases as the loss decreases, providing a theoretical mechanism for learning-rate warm-up: fixed step-sizes must be conservative near initialization, whereas the adaptive schedule can remain stable while increasing the step size in lower-curvature regions. We proved faster convergence guarantees than fixed-step gradient descent under convex, Aiming, and PL-type structure, extended the analysis to a stochastic finite-sum setting under interpolation, and showed empirically that the resulting loss-driven schedule can match tuned linear warm-up on language and vision tasks.

Future work should characterize the duration of this phase, develop layer-wise or block-wise variants of the condition, and generalize the results beyond Euclidean geometry (Bernstein & Newhouse, 2024; Islamov et al., 2026b). Another important direction is to extend the analysis to practical optimizers such as `Adam` (Kingma & Ba, 2015), `AdamW` (Loshchilov & Hutter, 2019), `Muon` (Jordan et al., 2024), `Scion` (Pethick et al., 2025), and `SCG` (Mokhtari et al.,

2020). Finally, analyzing the connection between Polyak-type step-sizes (Loizou et al., 2021; Orvieto & Xiao, 2024; Islamov et al., 2026a) and warm-up is a promising direction, as recent work suggests that such step-sizes can adapt to curvature and enable automatic warm-up (Defazio, 2026).

## Acknowledgements

Foivos Alimisis, Rustem Islamov, and Aurelien Lucchi acknowledge the financial support of the Swiss National Foundation, SNF grants No 207392 and 223031. The authors thank Eduard Gorbunov for fruitful discussions, which allowed us to improve the work.

## Impact Statement

This work studies theoretical and empirical explanations of learning-rate warm-up. Its most direct potential benefit is improved training stability and reduced hyperparameter-search cost, which may reduce wasted compute. The work does not introduce new datasets, deployed systems, or capabilities; its societal risks are therefore indirect and primarily those associated with more efficient training of machine-learning models.

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

# Appendix

## A. Comparison to Liu et al. (2025)

In this section, we provide a detailed comparison against a concurrent work by Liu et al. (2025). In the following section, we discuss in more detail the differences between our work and their work.

### A.1. The proposed Conditions

Liu et al. (2025) proposed the following condition with a general power $\rho > 0$:

$$\|\nabla^2 f(w)\| \leq K_0 + K_1(f(w) - f^*)^\rho. \tag{2}$$

The condition we study in the main part of the paper is a special case of (2) with $\rho = 1$. Liu et al. (2025) proves the convergence in the convex setting under (2), demonstrating benefits of the theoretical warm-up schedule. The proposed theoretical step-size is similar to ours in (1). However, their results can be simply recovered from our analysis for the $\rho = 1$ case.

Indeed, assuming $\rho > 1$ and that the iterates $\{w_k\}_{k=0}^K$ stay in the set $\{w \mid f(w) - f^* \leq f(w_0) - f^*\}$, which is the case for GD, we can simplify (2) as follows

$$\|\nabla^2 f(w)\|_2 \leq K_0 + K_\rho(f(w) - f^*)^\rho \leq K_0 + K_\rho(f(w_0) - f^*)^{\rho-1}(f(w) - f^*), \tag{3}$$

i.e., Definition 3.1 holds with $H_0 = K_0$ and $H_1 = K_\rho(f(w_0) - f^*)^{\rho-1}$. Therefore, the results of Theorem 4.2 apply, leading to the iteration complexity of GD with adaptive warm-up schedule of the form

$$K = \mathcal{O}\left(\frac{K_0 \text{dist}(w_0, \mathcal{S})^2}{\theta^2 \varepsilon} + \frac{K_\rho(f(w_0) - f^*)^{\rho-1}\text{dist}(w_0, \mathcal{S})^2}{\theta^2}\right).$$

This matches the bound in Liu et al. (2025) up to constants when $\theta = 1$, and shows that the adaptive schedule converges faster whenever $(f(w_0) - f^*)/\varepsilon \gg 1$. Given the simplification in (3), it remains open whether the convergence under the general condition (2) can be further tightened.

In Proposition D.1, we show that deep non-linear networks with Leaky-ReLU activations satisfy (2), albeit under stronger assumptions than Proposition 3.2. Moreover, Proposition 3.3 covers L2-regularized networks with two layers and arbitrary activations. If one considers deeper networks, $\rho$ increases with the number of layers $\ell$.

### A.2. Theoretical Evidence

We show that Definition 3.1 holds for several standard architectures: $(i)$ an $\ell$-layer linear network with MSE loss under balancedness (Proposition 3.2); $(ii)$ an $\ell$-layer nonlinear network with leaky-ReLU activations (Proposition D.1); and $(iii)$ two-layer network with cross-entropy losses (Proposition 3.3) under L2 regularization. These results provide clear theoretical evidence that the proposed condition is a good proxy for neural-network smoothness. By contrast, concurrent work by Liu et al. (2025) validates the condition only on a toy MLP model and a recurrent model with fewer than four parameters and sigmoid activation, leaving its extension beyond such simple cases unclear. Moreover, the representativeness of the proposed condition at the initial phase of training does not seem to be discussed.

### A.3. Experimental Evidence

Liu et al. (2025) empirically test the proposed condition on two setups: ResNet18 trained on CIFAR-10 (Krizhevsky, 2009) and a small NanoGPT-style transformer (6 blocks, 384-dimensional embeddings, 6 attention heads) trained on the Tiny Shakespeare dataset (Karpathy, 2015). They present smoothness versus training loss in a log-log scale, which leaves open how well the condition aligns with empirical behavior. Moreover, they show that these models train efficiently without learning-rate warm-up, and that warm-up offers no benefit (see Table 1 in Liu et al. (2025)). These settings leave open how the condition behaves in larger regimes where warm-up is empirically important. Our experiments complement theirs by evaluating larger language and vision models and by comparing against tuned linear warm-up. In contrast, we validate the condition on much larger models, where warm-up is crucial for achieving improved performance (see Figures 4 and 5).

### A.4. Theoretical Analysis

**Deterministic Setting.** Liu et al. (2025) establish convergence guarantees for GD in both convex and general non-convex regimes. In the non-convex case, their worst-case analysis shows that an adaptive warm-up schedule offers no advantage over a constant step-size beyond numerical factors, which is consistent with our findings. Benefits emerge in the convex setting, which is again similar to our results. In contrast, we extend the analysis beyond convexity, demonstrating the benefits of warm-up under Aiming and PL conditions, which are known to hold for sufficiently wide networks.

**Stochastic Setting.** The convergence guarantees in Liu et al. (2025) and our work are not directly comparable. We focus on finite-sum minimization in expectation under the Aiming condition and interpolation, showing that, under these assumptions, SGD with an adaptive warm-up schedule attains performance comparable to GD in the deterministic case. By contrast, Liu et al. (2025) analyzes general non-convex objectives under almost surely bounded variance or the almost-sure ABD assumption (Khaled & Richtárik, 2023) with high probability, where their results do not demonstrate warm-up benefits analogous to those in the deterministic setting.

**Lower Bounds.** Both works provide lower bounds for constant step-size GD, however, the "hard-to-optimize" functions and proof techniques are different.

### A.5. Empirical Results

We test the theoretically inspired LR warm-up schedule of the form $\frac{\eta_k}{\max\{1, f_{S_k}(w_k)/C\}}$, where $\eta_k$ follows WSD or cosine annealing schedule, $f_{S_k}(w_k)$ is the current stochastic loss and $C$ is the parameter of $(H_0, H_1)$ warm-up schedule. The proposed $(H_0, H_1)$ warm-up schedule is used when training language models of size 70M, 160M, and 410M on the FineWeb dataset and the ViT-Tiny model on the ImageNet-32 dataset. Our empirical results demonstrate that the proposed theoretical warm-up with tuned $C$ matches the performance of a linear warm-up and improves over a no-warm-up baseline. In contrast, Liu et al. (2025) conducts experiments when training the ResNet18 model on the CIFAR10 dataset. Their results demonstrate that the theoretical warm-up schedule $\frac{1}{4\sqrt{2}+4} \max\{\frac{1}{K_0}, \frac{1}{3K_1 f_{S_k}(w_k)}\}$ with tuned $K_0$ and $K_1$ matches the performance of linear and no-warm-up baselines.

## B. Arithmetics of $(H_0, H_1)$-smooth Functions

First, we provide a formal proof of the conjecture mentioned in Section 3. In other words, the following result demonstrates that the class of $(H_0, H_1)$-smooth functions contains all $(L_0, L_1)$-smooth functions.

**Proposition B.1.** *Assume that $f$ is $(L_0, L_1)$-smooth and bounded from below, i.e., $\|\nabla^2 f(w)\| \le L_0 + L_1\|\nabla f(w)\|$ and $f^* > -\infty$. Then $f$ satisfies Definition 3.1 with*

$$H_0 = L_0 + \frac{L_0 L_1}{\nu}, \quad H_1 = \frac{4L_1^2 + \nu L_1}{2\nu},$$

*where $\nu$ satisfies the equality $\nu = e^{-\nu^2}$.*

*Proof.* We start with Lemma 2.2 in Gorbunov et al. (2025)

$$\|\nabla f(w)\|^2 \le \frac{2}{\nu}(L_0 + L_1\|\nabla f(w)\|)(f(w) - f^*)$$

$$\Leftrightarrow \|\nabla f(w)\|^2 - \frac{2L_1}{\nu}\|\nabla f(w)\|(f(w) - f^*) - \frac{2L_0}{\nu}(f(w) - f^*) \le 0.$$

We need to solve this quadratic inequality w.r.t. $\|\nabla f(w)\|$. The discriminant is

$$\frac{4L_1^2}{\nu^2}(f(w) - f^*)^2 + 4 \cdot 1 \cdot \frac{2L_0}{\nu}(f(w) - f^*) > 0 = \frac{4L_1^2}{\nu^2}(f(w) - f^*)^2 + \frac{8L_0}{\nu}(f(w) - f^*) > 0,$$

---

[2] One can check numerically that $\nu \in (0.56, 0.57)$.

i.e., it is positive. Since $\|\nabla f(w)\| \geq 0$, we should also satisfy

$$\|\nabla f(w)\| \leq \frac{\frac{2L_1}{\nu}(f(w) - f^*) + \sqrt{\frac{4L_1^2}{\nu^2}(f(w) - f^*)^2 + \frac{8L_0}{\nu}(f(w) - f^*)}}{2}$$

$$\overset{(i)}{\leq} \frac{L_1}{\nu}(f(w) - f^*) + \sqrt{\frac{L_1^2}{\nu^2}(f(w) - f^*)^2} + \sqrt{\frac{2L_0}{\nu}(f(w) - f^*)}$$

$$\overset{(ii)}{\leq} \frac{2L_1}{\nu}(f(w) - f^*) + \frac{L_0}{\nu} + \frac{1}{2}(f(w) - f^*)$$

$$= \frac{L_0}{\nu} + \frac{4L_1 + \nu}{2\nu}(f(w) - f^*),$$

where $(i)$ follows from the inequality $\sqrt{a+b} \leq \sqrt{a} + \sqrt{b}$ for any $a, b \geq 0$, $(ii)$ – from the inequality $\sqrt{ab} \leq \frac{a}{2} + \frac{b}{2}$ for any $a, b \geq 0$. Therefore, we obtain

$$\|\nabla^2 f(w)\| \leq L_0 + L_1\|\nabla f(w)\|$$

$$\leq L_0 + \frac{L_0 L_1}{\nu} + \frac{4L_1^2 + \nu L_1}{2\nu}(f(w) - f^*),$$

which means that the function $f$ is $(H_0, H_1)$-smooth.

$\square$

Next, we demonstrate that operations like summation preserve $(H_0, H_1)$-smoothness. First, we show that the class of $(H_0, H_1)$-smooth functions is closed under summation.

**Proposition B.2.** *Let $f$ and $g$ be $(H_0^f, H_1^f)$-smooth and $(H_0^g, H_1^g)$-smooth respectively. Then $h := f + g$ is $(H_0, H_1)$-smooth with*

$$H_0 = (H_0^f + H_0^g + \max\{H_1^f, H_1^g\}h^* - H_1^f f^* - H_1^g g^*), \quad and \quad H_1 = \max\{H_1^f, H_1^g\}.$$

*Proof.* By Definition 3.1, we have

$$\|\nabla^2 f(w)\| \leq H_0^f + H_1^f(f(w) - f^*), \quad \|\nabla^2 g(w)\| \leq H_0^g + H_1^g(g(w) - g^*).$$

$\square$

Therefore, we have

$$\|\nabla^2 h(w)\| = \|\nabla^2 f(w) + \nabla^2 g(w)\|$$

$$\leq \|\nabla^2 f(w)\| + \|\nabla^2 g(w)\|$$

$$\leq H_0^f + H_1^f(f(w) - f^*) + H_0^g + H_1^g(g(w) - g^*)$$

$$\leq (H_0^f + H_0^g) + \max\{H_1^f, H_1^g\}(f(w) + g(w)) - H_1^f f^* - H_1^g g^*$$

$$= \underbrace{(H_0^f + H_0^g + \max\{H_1^f, H_1^g\}h^* - H_1^f f^* - H_1^g g^*)}_{:=H_0} + \underbrace{\max\{H_1^f, H_1^g\}}_{:=H_1}(h(w) - h^*).$$

Note that $h^* \geq f^* + g^*$. Therefore, we have

$$\max\{H_1^f, H_1^g\}h^* - H_1^f f^* - H_1^g g^* \geq H_1^f h^* + H_1^g h^* - H_1^f f^* - H_1^g g^* \geq 0,$$

i.e., $H_0 \geq 0$.

The next proposition shows that the class of $(H_0, H_1)$-smooth functions is closed under affine transformation.

**Proposition B.3.** *Let $g: \mathbb{R}^q \to \mathbb{R}$ be $(H_0^g, H_1^g)$-smooth, $A \in \mathbb{R}^{q \times p}$ be an arbitrary matrix, and $b \in \mathbb{R}^q$ be an arbitrary vector. We define $f: \mathbb{R}^p \to \mathbb{R}$ as $f(w) := g(Aw + b)$. Then $f$ is $(H_0^f, H_1^f)$-smooth with*

$$H_0^f = \|A\|^2(H_0^g + H_1(f^* - g^*)), \quad H_1^f = \|A\|^2 H_1^g,$$

*where $f^* = \min_{w \in \mathbb{R}^p} f(w), g^* = \min_{y \in \mathbb{R}^q} g(y)$.*

*Proof.* First, note that

$$f^* = \min_{w \in \mathbb{R}^p} g(Aw + b) \geq \min_{y \in \mathbb{R}^q} g(y) = g^*,$$

since the first minimum is taken in $\mathrm{Im}(A)$. Second, note that $\nabla^2 f(w) = A^\top \nabla^2 g(Aw + b)A$. Therefore,

$$\begin{aligned}
\|\nabla^2 f(w)\| &= \|A^\top \nabla^2 g(Aw + b)A\| \\
&\leq \|A^\top\|\cdot\|\nabla^2 g(Aw + b)\|\cdot\|A\| \\
&\leq \|A\|^2\cdot(H_0^g + H_1^g(g(Aw + b) - g^*)) \\
&= \|A\|^2 H_0^g + \|A\|^2 H_1^g(f(w) - f^* + f^* - g^*) \\
&= \|A\|^2(H_0^g + H_1(f^* - g^*)) + \|A\|^2 H_1^g(f(w) - f^*).
\end{aligned}$$

$\square$

In the next proposition, we demonstrate that the class of $(L_0, L_1)$-smooth functions is not closed under summation.

**Proposition B.4.** *There exist two $(L_0, L_1)$-smooth functions $f_1, f_2 \colon \mathbb{R} \to \mathbb{R}$ such that their sum $f = f_1 + f_2$ does not belong to the class of $(L_0, L_1)$-smooth functions.*

*Proof.* Let us consider two functions $f_1$ and $f_2$ defined as

$$f_1(w) = \int_0^w (u + \sin(u^2))\mathrm{d}u, \quad f_2(w) = \int_0^w (-v + \sin(v^2))\mathrm{d}v.$$

Then we have

$$f_1'(w) = w + \sin(w^2), \quad f_1''(w) = 1 + 2w\cos(w^2), \quad f_2'(w) = -w + \sin(w^2), \quad f_2''(w) = -1 + 2w\cos(w^2).$$

Therefore, we have

$$|f_{1,2}''(w)|\leq 1 + |2w\cos(w^2)|\leq 1 + 2|w|,$$

and

$$|f_{1,2}'(w)|\geq |\pm w + \sin(w^2)|\geq |w|-|\sin(w^2)|\geq |w|-1.$$

This implies that for $|w|\geq 1$

$$|f_{1,2}''(w)|\leq 1 + 2|w|\leq 3 + 3(|w|-1) \leq 3 + 3|f_{1,2}'(w)|.$$

For $|w|\leq 1$, we have $|f_{1,2}''(w)|\leq 3$. Thus, both functions are $(L_0, L_1)$-smooth with $L_0 = L_1 = 3$. They sum is $f(w) = 2\sin(w^2)$, for which we have

$$f'(w) = 2\sin(w^2), \quad f''(w) = 4w\cos(w^2).$$

Now we consider points $\{w_m\}_{m=1}^\infty$ with $w_m = \sqrt{m\pi}$. At these points, we have

$$f'(w_m) = 0, \quad f''(w_m) = 4w_m \to \infty.$$

If $f$ were $(L_0, L_1)$-smooth, then we would have

$$|f''(w_m)|\leq L_0 + L_1|f'(w_m)|\leq L_0.$$

This contradiction concludes the proof. $\square$

We now show that there exists an affine transformation that does not preserve $(L_0, L_1)$-smoothness.

**Proposition B.5.** *There exist a $(L_0, L_1)$-smooth function $g \colon \mathbb{R}^2 \to \mathbb{R}$ and a matrix $A \in \mathbb{R}^{2 \times 1}$ such that a function $f(w) = g(Aw)$ does not belong to the class of $(L_0, L_1)$-smooth functions.*

*Proof.* Let us consider $A = \begin{pmatrix} 1 \\ 0 \end{pmatrix}$, $b = 0$, and $g(y_1, y_2) = h(y_1)e^{y_2}$ with $h(y_1) = \cos(y_1)e^{y_1}$. We know that

$$h'(y_1) = e^{y_1}(\cos(y_1) - \sin(y_1)), \quad h''(y_1) = -2\sin(y_1)e^{y_1}.$$

Therefore,

$$\nabla g(y) = e^{y_2} \begin{pmatrix} h'(y_1) \\ h(y_1) \end{pmatrix}, \quad \nabla^2 g(y) = e^{y_2} \begin{pmatrix} h''(y_1) & h'(y_1) \\ h'(y_1) & h(y_1) \end{pmatrix}.$$

Note that

$$|h''(y_1)| = 2e^{y_1}|\sin(y_1)| \leq 2e^{y_1}|\cos(y_1)| + 2e^{y_1}|\cos(y_1) - \sin(y_1)| = 2|h(y_1)| + 2|h'(y_1)|.$$

Therefore, we have

$$\begin{aligned}
\|\nabla^2 g(y)\|_2 &\leq \|\nabla^2 g(y)\|_{\mathrm{F}} \\
&= e^{y_2}\sqrt{(h''(y_1))^2 + 2(h'(y_1))^2 + (h(y_1))^2} \\
&\leq e^{y_2}\sqrt{4(h(y_1) + h'(y_1))^2 + 2(h'(y_1))^2 + (h(y_1))^2} \\
&\leq e^{y_2}\sqrt{8(h(y_1))^2 + 8(h'(y_1))^2 + 2(h'(y_1))^2 + (h(y_1))^2} \\
&\leq \sqrt{10}e^{y_2}\sqrt{(h(y_1))^2 + (h'(y_1))^2}
\end{aligned}$$

Note that $\|\nabla g(y)\| = e^{y_2}\sqrt{(h(y_1))^2 + (h'(y_1))^2}$. Therefore, we obtain the bound $\|\nabla^2 g(y)\|_2 \leq \sqrt{10}\|\nabla g(y)\|$. Now we consider the function $f(w) = g(Aw) = g(w, 0) = h(w)$. For $f$, we have

$$f'(w) = e^w(\cos(w) - \sin(w)), \quad f''(w) = -2\sin(w)e^w.$$

We consider the points $\{w_m\}_{m=1}^\infty$ with $w_m = \frac{\pi}{4} + 2\pi m$. Therefore, $\cos(w_m) = \sin(w_m) = \sqrt{2}/2$. This implies, that at these points $f'(w_m) = e^{w_m}(\sqrt{2}/2 - \sqrt{2}/2) = 0$ and $f''(w_m) = -\sqrt{2}e^{w_m}$. Thus, we obtain that $|f''(w_m)| \to \infty$ with $m \to \infty$, while $|f'(w_m)| = 0$. This implies that $f$ does not satisfy $(L_0, L_1)$-smoothness for any $L_0, L_1 \geq 0$.

$\square$

## C. Proof of Proposition 3.1

**Proposition 3.1.** *During the trajectory $(u(t), v(t))$ of gradient flow starting sufficiently close to $(0, 0)$, the Hessian $H(t)$ of the loss function $f(u(t), v(t)) = (y - u(t)v(t)x)^2$, satisfies*

*(i) $\frac{d}{dt}\|H(t)\|_2 < 0$, if $\frac{x^2(u(0) - v(0))^2}{4} \leq (y - 2u(t)v(t)x)^2$.*

*(ii) $\frac{d}{dt}\|H(t)\|_2 > 0$, when $y < 2u(t)v(t)x$.*

*Proof.* The parameters $u(t)$ and $v(t)$ evolve over time via gradient flow as: $\frac{du}{dt} = -g_u$ and $\frac{dv}{dt} = -g_v$, where $g_u = \frac{\partial f(u,v)}{\partial u}, g_v = \frac{\partial f(u,v)}{\partial v}$
Let $R := y - uvx$ be the residual error. We have

$$\begin{aligned}
g_u &= \frac{\partial f}{\partial u} = 2(y - uvx) \cdot (-vx) = -2vxR \\
g_v &= \frac{\partial f}{\partial v} = 2(y - uvx) \cdot (-ux) = -2uxR.
\end{aligned}$$

The gradient vector is $g = \begin{pmatrix} g_u \\ g_v \end{pmatrix}$.

The Hessian matrix is $H = \nabla^2 f = \begin{pmatrix} H_{11} & H_{12} \\ H_{21} & H_{22} \end{pmatrix}$. We can calculate its components directly:

$$A := H_{11} = \frac{\partial g_u}{\partial u} = \frac{\partial}{\partial u}(-2vyx + 2uv^2x^2) = 2v^2x^2$$

$$C := H_{22} = \frac{\partial g_v}{\partial v} = \frac{\partial}{\partial v}(-2uyx + 2u^2vx^2) = 2u^2x^2$$

$$B := H_{12} = \frac{\partial g_u}{\partial v} = \frac{\partial}{\partial v}(-2vyx + 2uv^2x^2) = -2yx + 4uvx^2 = 2x(2uvx - y)$$

$$H_{21} = \frac{\partial g_v}{\partial u} = \frac{\partial}{\partial u}(-2uyx + 2u^2vx^2) = -2yx + 4uvx^2 = B.$$

So, the Hessian is:

$$H(t) = \begin{pmatrix} 2v^2x^2 & 2x(2uvx - y) \\ 2x(2uvx - y) & 2u^2x^2 \end{pmatrix}.$$

The spectral norm $\|H\|_2$ of a $2 \times 2$ symmetric matrix of the form $\begin{bmatrix} A & B \\ B & C \end{bmatrix}$ is its largest eigenvalue in absolute value. The two eigenvalues $\lambda_\pm$ are found by solving the characteristic Eq. $\det(H - \lambda I) = 0$:

$$\lambda^2 - (A + C)\lambda + (AC - B^2) = 0.$$

The solution of this quadratic Eq. gives the two eigenvalues:

$$\lambda_\pm = \frac{(A + C) \pm \sqrt{(A + C)^2 - 4(AC - B^2)}}{2} = \frac{(A + C) \pm \sqrt{(A - C)^2 + 4B^2}}{2}.$$

The spectral norm is the larger of these in magnitude, i.e.

$$\|H\|_2 = \lambda_+ = \frac{(A + C) + \sqrt{(A - C)^2 + 4B^2}}{2}.$$

For simplicity, we define $S_\lambda := \frac{1}{2}\sqrt{(A - C)^2 + 4B^2}$. Then we have: $\|H\|_2 = \lambda_+ = \frac{1}{2}(A + C) + S_\lambda$.

Our goal is to compute $\frac{d}{dt}\|H\|_2$ and analyze its sign.

First, we must find $\dot{H}$, which requires the 3rd derivatives of $f$:

- $\frac{\partial H_{11}}{\partial u} = \frac{\partial}{\partial u}(2v^2x^2) = 0$

- $\frac{\partial H_{11}}{\partial v} = \frac{\partial}{\partial v}(2v^2x^2) = 4vx^2$

- $\frac{\partial H_{12}}{\partial u} = \frac{\partial}{\partial u}(4uvx^2 - 2xy) = 4vx^2$

- $\frac{\partial H_{12}}{\partial v} = \frac{\partial}{\partial v}(4uvx^2 - 2xy) = 4ux^2$

- $\frac{\partial H_{22}}{\partial u} = \frac{\partial}{\partial u}(2u^2x^2) = 4ux^2$

- $\frac{\partial H_{22}}{\partial v} = \frac{\partial}{\partial v}(2u^2x^2) = 0$

Now we find the components of $\dot{H}$ using the chain rule:

$$\dot{H}_{ij} = \frac{\partial H_{ij}}{\partial u}\dot{u} + \frac{\partial H_{ij}}{\partial v}\dot{v} = -g_u\left(\frac{\partial H_{ij}}{\partial u}\right) - g_v\left(\frac{\partial H_{ij}}{\partial v}\right).$$

- $\dot{H}_{11} = -g_u(0) - g_v(4vx^2) = -g_v(4vx^2)$

- $\dot{H}_{22} = -g_u(4ux^2) - g_v(0) = -g_u(4ux^2)$

- $\dot{H}_{12} = -g_u(4vx^2) - g_v(4ux^2)$

We substitute $g_u = -2vxR$ and $g_v = -2uxR$ and we get

- $\dot{H}_{11} = -(-2uxR)(4vx^2) = 8uvx^3R$

- $\dot{H}_{22} = -(-2vxR)(4ux^2) = 8uvx^3R$

This reveals that $\dot{H}_{11} = \dot{H}_{22}$, which will be useful later. We denote these terms by $D_H$:

$$D_H := \dot{H}_{11} = \dot{H}_{22} = 8uvx^3R.$$

The off-diagonal term is:

$$E_H := \dot{H}_{12} = -(-2vxR)(4vx^2) - (-2uxR)(4ux^2)$$
$$= 8v^2x^3R + 8u^2x^3R = 8x^3(u^2 + v^2)R$$

So, $\dot{H} = \begin{pmatrix} D_H & E_H \\ E_H & D_H \end{pmatrix}$.

For a $2 \times 2$ symmetric matrix $H$ with $A = H_{11}$ and $C = H_{22}$, the derivative of the largest eigenvalue $\lambda_+$ is:

$$\frac{d\lambda_+}{dt} = \frac{1}{2}(\dot{A} + \dot{C}) + \frac{(A - C)(\dot{A} - \dot{C}) + 4B\dot{B}}{4S_\lambda}.$$

Our simplification $\dot{A} = \dot{C} = D_H$ gives a cleaner expression:

$$\frac{d\lambda_+}{dt} = \frac{1}{2}(D_H + D_H) + \frac{(A - C)\cdot 0 + 4BE_H}{4S_\lambda} = D_H + \frac{BE_H}{S_\lambda}.$$

Substituting $S_\lambda = \frac{1}{2}\sqrt{(A-C)^2 + 4B^2}$, we get

$$\frac{d}{dt}\|H\|_2 = D_H + \frac{2BE_H}{\sqrt{(A-C)^2 + 4B^2}}.$$

Substituting our components $A, B, C, D_H, E_H$ gives the final formula:

$$\frac{d}{dt}\|H\|_2 = 8uvx^3R + \frac{2 \cdot [2x(2uvx - y)] \cdot [8x^3(u^2 + v^2)R]}{\sqrt{(2v^2x^2 - 2u^2x^2)^2 + 4(2x(2uvx - y))^2}}.$$

To simplify the notation, we denote $B_{\text{sign}} := y - 2uvx$. Then $2uvx - y = -B_{\text{sign}}$ and plugging it in the previous formula, we have

$$\frac{d}{dt}\|H\|_2 = 8uvx^3R + \frac{2 \cdot (2x(-B_{\text{sign}})) \cdot (8x^3(u^2 + v^2)R)}{2\sqrt{x^4(v^2 - u^2)^2 + 4x^2(-B_{\text{sign}})^2}}$$
$$= 8uvx^3R - \frac{32x^4(u^2 + v^2)RB_{\text{sign}}}{2|x|\sqrt{x^2(v^2 - u^2)^2 + 4B_{\text{sign}}^2}}.$$

Let $c = u^2 - v^2$ (a constant, as we will prove later). To simplify the fraction, we use the identity $\frac{x^4}{|x|} = \frac{x^4}{x \cdot \text{sign}(x)} = x^3 \cdot \text{sign}(x)$ and we get

$$\frac{d}{dt}\|H\|_2 = 8uvx^3R - 16 \cdot \text{sign}(x) \cdot x^3(u^2 + v^2)R \cdot \frac{B_{\text{sign}}}{\sqrt{x^2c^2 + 4B_{\text{sign}}^2}}.$$

Factoring out the common term $8x^3R$, we arrive at the general formula for the derivative, valid for all $x, y, u, v$:

$$\frac{d}{dt}\|H\|_2 = 8x^3R\left(uv - 2 \cdot \text{sign}(x) \cdot (u^2 + v^2) \cdot \frac{B_{\text{sign}}}{\sqrt{x^2c^2 + 4B_{\text{sign}}^2}}\right). \tag{4}$$

Before analyzing specific paths, we prove that the quantity $u^2 - v^2$ stays constant over the course of gradient flow.

Indeed, we compute the time-derivative of the quantity $(u^2 - v^2)$:

$$\frac{d}{dt}(u^2 - v^2) = 2u\dot{u} - 2v\dot{v}.$$

From the gradient flow definition, $\dot{u} = -g_u = 2vxR$ and $\dot{v} = -g_v = 2uxR$. Substituting these in:

$$\frac{d}{dt}(u^2 - v^2) = 2u(2vxR) - 2v(2uxR) = 4uvxR - 4uvxR = 0.$$

Because the time-derivative is always zero, the quantity $(u^2 - v^2)$ is a conserved quantity. Any trajectory is constrained to a manifold where $u^2 - v^2 = c$, where $c$ is determined by the random initial conditions $c = u(0)^2 - v(0)^2$.

We now pass to the analysis of the sign of $\frac{d}{dt}\|H\|_2$. For notational simplicity, let $S = u^2 + v^2$ and we have

$$\frac{d}{dt}\|H\|_2 = \underbrace{8x^3R}_{\text{Term 1}} \cdot \underbrace{\left(uv - \frac{2\text{sign}(x) \cdot S \cdot B_{\text{sign}}}{\sqrt{x^2c^2 + 4B_{\text{sign}}^2}}\right)}_{\text{Term 2}}$$

We can see that the trajectory consists in general of three phases, defined by the boundaries $x^2c^2 = 4B_{\text{sign}}^2$ and $B_{\text{sign}} = 0$.

**First phase:** We define the initial regime by the condition $x^2c^2 \leq 4B_{\text{sign}}^2$. This holds near the origin because $c \approx 0$ and $B_{\text{sign}} \approx y \neq 0$. In this regime: $\text{sign}(R) \approx \text{sign}(y)$ and $\text{sign}(B_{\text{sign}}) \approx \text{sign}(y)$.

1. **Term 1:** $\text{sign}(\text{Term 1}) = \text{sign}(x^3R) \approx \text{sign}(x^3y) = \text{sign}(xy)$.

2. **Term 2:** Let $F = \frac{2S|B_{\text{sign}}|}{\sqrt{x^2c^2 + 4B_{\text{sign}}^2}}$. We use the identity $uv = \text{sign}(uv) \cdot \frac{1}{2}\sqrt{S^2 - c^2}$ (easy to prove).

We must analyze two scenarios

- **Case A: $\text{sign}(uv) = \text{sign}(xy)$.** Then,

$$\text{Term 2} = \text{sign}(xy)\left[\frac{1}{2}\sqrt{S^2 - c^2} - |F|\right].$$

   We have $\frac{1}{2}\sqrt{S^2 - c^2} \leq \frac{S}{2}$. We also have $|F| \geq \frac{2S|B_{\text{sign}}|}{\sqrt{8B_{\text{sign}}^2}} = \frac{S}{\sqrt{2}} > \frac{S}{2}$. Thus, the term inside the bracket is negative and

$$\text{sign}\left(\frac{d}{dt}\|H\|_2\right) = -\text{sign}(xy)^2 = -1.$$

- **Case B: $\text{sign}(uv) = -\text{sign}(xy)$**

$$\text{Term 2} = -\text{sign}(xy)\left[\frac{1}{2}\sqrt{S^2 - c^2} + |F|\right].$$

   In this case, it holds

$$\text{sign}\left(\frac{d}{dt}\|H\|_2\right) = -\text{sign}(xy)^2\text{sign}\left(\frac{1}{2}\sqrt{S^2 - c^2} + |F|\right) = -1.$$

In both cases, the derivative of the spectral norm is negative, which means that we enter a flattening phase.

**Second phase:** This initial flattening phase is only guaranteed as long as our assumption $x^2 c^2 \leq 4 B_{\text{sign}}^2$ holds. As the flow moves, $uvx$ increases, so $B_{\text{sign}} = y - 2uvx$ gets smaller. Eventually, we will enter an ambiguous phase where $x^2 c^2 > 4 B_{\text{sign}}^2$, but we still have $\text{sign}(B_{\text{sign}}) = \text{sign}(y)$ (because the flow has not yet reached $uvx = y/2$). In this region, our bounding logic for $|F|$ is inconclusive, and the sign of the derivative is unknown. However, if $c = 0$, then this phase is vacuous and we pass directly to the next progressive sharpening phase.

**Third phase:** The flow then crosses the mid-point at $uvx = y/2$, which means $\text{sign}(B_{\text{sign}})$ finally flips to $-\text{sign}(y)$. The flow is now in a "correct" quadrant (having been repelled by the origin), so $\text{sign}(uv) = \text{sign}(xy)$. We have:

- $\text{sign}(\text{Term 1}) = \text{sign}(x^3 R) = \text{sign}(x^3 y) = \text{sign}(xy)$ (since $\text{sign}(R) = \text{sign}(y)$ always).

- The sign of $F$ is now $\text{sign}(F) = \text{sign}(x \cdot B_{\text{sign}}) = \text{sign}(x) \cdot (-\text{sign}(y)) = -\text{sign}(xy)$.

- This means $\text{sign}(uv)$ and $\text{sign}(F)$ are now opposites.

- Term 2 $= uv - F$.

    - If $\text{sign}(xy) = +1$: $\text{sign}(uv) = +1$ and $\text{sign}(F) = -1$. Term 2 is $(\text{pos}) - (\text{neg}) = +$.
    - If $\text{sign}(xy) = -1$: $\text{sign}(uv) = -1$ and $\text{sign}(F) = +1$. Term 2 is $(\text{neg}) - (\text{pos}) = -$.

- In both sub-cases, $\text{sign}(\text{Term 2})$ is $\text{sign}(xy)$.

The total sign is

$$\text{sign}\left(\frac{d}{dt}\|H\|_2\right) = \text{sign}(\text{Term 1}) \cdot \text{sign}(\text{Term 2}) = \text{sign}(xy) \cdot \text{sign}(xy) = +1.$$

Thus, a sharpening phase is guaranteed after the $uvx = y/2$ boundary.

$\square$

## D. Proofs for Section 3.1.1

**Proposition 3.2.** *Consider a deep linear network with $\ell$ layers and MSE loss:*

$$f(W) \equiv f(W_1, \ldots, W_\ell) = \|Y - W_1 W_2 \ldots W_\ell X\|_{\text{F}}^2,$$

*where $Y \in \mathbb{R}^{c \times m}$ are the labels, $X \in \mathbb{R}^{d \times m} (d \leq m)$ is the input, and $W_i \in \mathbb{R}^{n_{i-1} \times n_i}$, where $n_0 = c$ and $n_\ell = d$ are layer dimensions. In the weakly balanced subspace, $\|W_1\|_{\text{F}} = \|W_2\|_{\text{F}}$, we have:*

$(i)$ *If $\|W_1\|_{\text{F}}^\ell \|X\|_2 < (1 - \frac{1}{\ell})\sqrt{f(W)}$, then it holds*

$$\|\nabla^2 f(W)\| \leq H_0^{\text{warm-up}} + H_1^{\text{warm-up}}(f(W) - f^*)$$

*with constants defined in Eq. (7).*

$(ii)$ *If we additionally assume strong balancedness, then it holds*

$$\|\nabla^2 f(W)\| \leq H_0 + H_1(f(W) - f^*) \quad \text{for all } W,$$

*with constants defined in Eq. (13).*

*Proof.* The proof goes as follows: first we obtain a general upper bound for the spectral norm of the Hessian. Then, we consider cases depending on whether we want a local bound close to initialization or a global bound.

**Upper bound for the Hessian norm:** One can find an explicit formula for the Hessian of such neural network in (Kawaguchi, 2016), Lemma 4.3.

The Hessian of $f$ in vectorized form has blocks in the $(i, j)$ position for $j < i$, that are of the form

$$\frac{\partial^2 f}{\partial \text{vec}(W_i)\text{vec}(W_j)} = 2((W_1 \ldots W_{i-1}) \otimes (W_{i+1} \ldots W_\ell X)^\top)^\top ((W_1 \ldots W_{j-1}) \otimes (W_{j+1} \ldots W_\ell X)^\top)$$
$$+ 2((W_{j+1} \ldots W_{i-1})^\top \otimes (W_{i+1} \ldots W_\ell X))(I_{n_j} \otimes ((W_1 \ldots W_\ell X - Y)^\top W_1 \ldots W_{j-1})),$$

where $W_1 W_0, W_{\ell+1} W_\ell := I$.

For $j = i$, we have

$$\frac{\partial^2 f}{\partial \text{vec}(W_i)\text{vec}(W_j)} = 2((W_1 \ldots W_{i-1}) \otimes (W_{i+1} \ldots W_\ell X)^\top)^\top ((W_1 \ldots W_{j-1}) \otimes (W_{j+1} \ldots W_\ell X)^\top).$$

The spectral norm of the Hessian in vectorized form is upper bounded by the sum of the spectral norms of each such block. Indeed, let $M$ be an $N \times N$ block symmetric matrix:

$$M = \begin{pmatrix} M_{11} & M_{12} & \cdots & M_{1N} \\ M_{12}^\top & M_{22} & \cdots & M_{2N} \\ \vdots & \vdots & \ddots & \vdots \\ M_{1N}^\top & M_{2N}^\top & \cdots & M_{NN} \end{pmatrix}$$

where each $M_{ij}$ is a matrix block.

A fundamental result for block matrices states that the spectral norm of a block matrix is bounded by the spectral norm of the matrix formed by the spectral norms of its blocks. Let us define a real symmetric $N \times N$ matrix $\tilde{M}$ where each element $(\tilde{M})_{ij}$ is the spectral norm of the corresponding block $M_{ij}$:

$$\tilde{M} = \begin{pmatrix} \|M_{11}\|_2 & \|M_{12}\|_2 & \cdots & \|M_{1N}\|_2 \\ \|M_{12}\|_2 & \|M_{22}\|_2 & \cdots & \|M_{2N}\|_2 \\ \vdots & \vdots & \ddots & \vdots \\ \|M_{1N}\|_2 & \|M_{2N}\|_2 & \cdots & \|M_{NN}\|_2 \end{pmatrix}$$

The inequality is then:

$$\|M\|_2 \leq \|\tilde{M}\|_2$$

Since the spectral norm is always upper bounded by the Frobenius norm, it holds

$$\|\tilde{M}\|_2 \leq \|\tilde{M}\|_F = \sqrt{\sum_{i=1}^N \sum_{j=1}^N \|M_{ij}\|_2^2} \leq \sum_{i=1}^N \sum_{j=1}^N \|M_{ij}\|_2.$$

Thus, indeed, it holds

$$\|M\|_2 \leq \sum_{i=1}^N \sum_{j=1}^N \|M_{ij}\|_2. \tag{5}$$

Going back to the Hessian, we can upper bound the spectral norm of the $(i, j)$ block using only the weak form of balancedness $\|W_i\|_F = \|W_{i+1}\|_F$ (which is implied by the strong form of balancedness).

For $1 < j < i < \ell$, we have

$$
\begin{aligned}
\left\| \frac{\partial^2 f}{\partial \mathrm{vec}(W_i)\mathrm{vec}(W_j)} \right\|_2 &= 2\|((W_1 \ldots W_{i-1}) \otimes (W_{i+1} \ldots W_\ell X)^\top)^\top ((W_1 \ldots W_{j-1}) \otimes (W_{j+1} \ldots W_\ell X)^\top) \\
&\quad + 2((W_{j+1} \ldots W_{i-1})^\top \otimes (W_{i+1} \ldots W_\ell X))(I_{n_j} \otimes ((W_1 \ldots W_\ell X - Y)^\top W_1 \ldots W_{j-1}))\|_2 \\
&\leq 2\|((W_1 \ldots W_{i-1}) \otimes ((W_{i+1} \ldots W_\ell X)^\top)^\top (W_1 \ldots W_{j-1}) \otimes (W_{j+1} \ldots W_\ell X)^\top)\|_2 \\
&\quad + 2\|((W_{j+1} \ldots W_{i-1})^\top \otimes (W_{i+1} \ldots W_\ell X))(I_{n_j} \otimes ((W_1 \ldots W_\ell X - Y)^\top W_1 \ldots W_{j-1}))\|_2 \\
&\leq 2\|W_1\|_{\mathrm{F}}^{2\ell-2}\|X\|_2^2 + 2\|W_1\|_{\mathrm{F}}^{\ell-2}\|X\|_2 \sqrt{f(W)}.
\end{aligned}
$$

For the last inequality, we used that for matrices $A$ and $B$

- $\|A \otimes B\|_2 = \|A\|_2 \|B\|_2$.

- $\|A\|_2 = \|A^\top\|_2$

- $\|AB\|_2 \leq \|A\|_2 \|B\|_2$.

- $\|A\|_2 \leq \|A\|_{\mathrm{F}}$.

For $j = 1$ and $1 < i < \ell$, we have

$$
\begin{aligned}
\frac{\partial^2 f}{\partial \mathrm{vec}(W_i)\mathrm{vec}(W_1)} &= 2((W_1 \ldots W_{i-1}) \otimes (W_{i+1} \ldots W_\ell X)^\top)^\top (I_c \otimes (W_2 \ldots W_\ell X)^\top) \\
&\quad + 2((W_2 \ldots W_{i-1})^\top \otimes (W_{i+1} \ldots W_\ell X))(I_{n_j} \otimes (W_1 \ldots W_\ell X - Y)^\top),
\end{aligned}
$$

thus

$$
\left\| \frac{\partial^2 f}{\partial \mathrm{vec}(W_i)\mathrm{vec}(W_1)} \right\|_2 \leq 2\|W_1\|_{\mathrm{F}}^{2\ell-2}\|X\|_2^2 + 2\|W_1\|_{\mathrm{F}}^{\ell-2}\|X\|_2 \sqrt{f(W)}.
$$

For $j = 1$ and $i = \ell$, it holds

$$
\begin{aligned}
\frac{\partial^2 f}{\partial \mathrm{vec}(W_i)\mathrm{vec}(W_1)} &= 2((W_1 \ldots W_{\ell-1}) \otimes X)(I_c \otimes (W_2 \ldots W_\ell X)^\top) \\
&\quad + 2((W_2 \ldots W_{\ell-1})^\top \otimes X)(I_{n_1} \otimes ((W_1 \ldots W_\ell X - Y)^\top)),
\end{aligned}
$$

thus again

$$
\left\| \frac{\partial^2 f}{\partial \mathrm{vec}(W_\ell)\mathrm{vec}(W_1)} \right\|_2 \leq 2\|W_1\|_{\mathrm{F}}^{2\ell-2}\|X\|_2^2 + 2\|W_1\|_{\mathrm{F}}^{\ell-2}\|X\|_2 \sqrt{f(W)}.
$$

For the case that $1 < j < \ell$ and $i = \ell$, we have

$$
\begin{aligned}
\frac{\partial^2 f}{\partial \mathrm{vec}(W_i)\mathrm{vec}(W_j)} &= 2((W_1 \ldots W_{\ell-1}) \otimes X)((W_1 \ldots W_{j-1}) \otimes (W_{j+1} \ldots W_\ell X)^\top) \\
&\quad + 2((W_{j+1} \ldots W_{\ell-1})^\top \otimes X)(I_{n_j} \otimes ((W_1 \ldots W_\ell X - Y)^\top W_1 \ldots W_{j-1})).
\end{aligned}
$$

Again, we have

$$
\left\| \frac{\partial^2 f}{\partial \mathrm{vec}(W_\ell)\mathrm{vec}(W_j)} \right\|_2 \leq 2\|W_1\|_{\mathrm{F}}^{2\ell-2}\|X\|_2^2 + 2\|W_1\|_{\mathrm{F}}^{\ell-2}\|X\|_2 \sqrt{f(W)}.
$$

Similarly, we have for the diagonal blocks that

$$
\left\| \frac{\partial^2 f}{\partial \mathrm{vec}(W_i)\mathrm{vec}(W_j)} \right\|_2 \leq 2\|W_1\|_{\mathrm{F}}^{2\ell-2}\|X\|_2^2.
$$

In summary, since we have $(\ell^2 - \ell)$-many off-diagonal blocks and $\ell$-many diagonal blocks in the Hessian, its norm is bounded as

$$\|\nabla^2 f(W)\|_2 \leq 2\ell^2 \|W_1\|_F^{2\ell-2} \|X\|_2^2 + 2(\ell^2 - \ell)\|W_1\|_F^{\ell-2}\|X\|_2 \sqrt{f(W)}. \tag{6}$$

If $\|W_1\|_F^\ell \|X\|_2 < (1 - \frac{1}{\ell})\sqrt{f(W)}$, then the first term of the previous bound is smaller than the second, thus we have

$$\|\nabla^2 f(W)\|_2 \leq 4(\ell^2 - \ell)\|W_1\|_F^{\ell-2}\|X\|_2 \sqrt{f(W)}.$$

It also holds $\|W_1\|_F^{\ell-2} \leq (1 - \frac{1}{\ell})^{\frac{\ell-2}{\ell}} \|X\|_2^{-\frac{\ell-2}{\ell}} \sqrt{f(W)}^{\frac{\ell-2}{\ell}}$. In total, we have

$$\|\nabla^2 f(W)\|_2 \leq H_0^{\text{warm-up}} + H_1^{\text{warm-up}}(f(W) - f^*),$$

where

$$H_0^{\text{warm-up}} := 4(\ell^2 - \ell)\left(1 - \frac{1}{\ell}\right)^{\frac{\ell-2}{\ell}} \|X\|_2^{\frac{2}{\ell}}(1 + (f^*)^{\frac{\ell-1}{\ell}}), \qquad H_1^{\text{warm-up}} := 4(\ell^2 - \ell)\left(1 - \frac{1}{\ell}\right)^{\frac{\ell-2}{\ell}} \|X\|_2^{\frac{2}{\ell}}. \tag{7}$$

For the latter, we used the bound

$$f(W)^{\frac{\ell-1}{\ell}} = (f(W) - f^* + f^*)^{\frac{\ell-1}{\ell}} \leq (f(W) - f^*)^{\frac{\ell-1}{\ell}} + (f^*)^{\frac{\ell-1}{\ell}} \leq 1 + (f(W) - f^*) + (f^*)^{\frac{\ell-1}{\ell}}.$$

This concludes the result of (i).

Now, we move to the global bound. For that, we need to obtain a general lower bound for the loss value. It holds

$$\|W_1 \ldots W_\ell X\|_F^2 = \text{Tr}(X^\top W_\ell^\top \ldots W_2^\top W_1^\top W_1 W_2 \ldots W_\ell X) \geq \lambda_{\min}(XX^\top)\text{Tr}(W_\ell^\top \ldots W_2^\top W_1^\top W_1 W_2 \ldots W_\ell). \tag{8}$$

In order to deal with the last term, we use the strong balancedness assumption:

$$W_\ell^\top \ldots W_4^\top W_3^\top W_2^\top W_1^\top W_1 W_2 W_3 W_4 \ldots W_\ell = W_\ell^\top \ldots W_4^\top W_3^\top W_2^\top W_2 W_2^\top W_2 W_3 W_4 \ldots W_\ell =$$
$$W_\ell^\top \ldots W_4^\top W_3^\top W_3 W_3^\top W_3 W_3^\top W_3 W_4 \ldots W_\ell = W_\ell^\top \ldots W_4^\top W_4 W_4^\top W_3^\top W_3 W_4 W_4^\top W_4 \ldots W_\ell =$$
$$W_\ell^\top \ldots W_5 W_5^\top W_4^\top W_3^\top W_3 W_4 W_5 W_5^\top \ldots W_\ell$$

and the process continuous until we reach the expression

$$(W_\ell^\top W_\ell)W_\ell^\top W_{\ell-1}^\top \ldots W_6^\top W_5^\top W_4^\top W_3^\top W_3 W_4 W_5 W_6 \ldots W_{\ell-1}W_\ell(W_\ell^\top W_\ell).$$

We can now do the same process starting from $W_3$ and so on. Repeating this process $\ell/2$ times if $\ell$ is even and $(\ell - 1)/2$ if $\ell$ is odd, we arrive to the expression

$$\underbrace{(W_\ell^\top W_\ell) \ldots (W_\ell^\top W_\ell)}_{\ell-\text{times}} = (W_\ell^\top W_\ell)^\ell.$$

Since the eigenvalues of $(W_\ell^\top W_\ell)^\ell$ are $\ell$ powers of the eigenvalues of $W_\ell^\top W_\ell$, we can use the generalized mean inequality and derive

$$\frac{\text{Tr}((W_\ell^\top W_\ell)^\ell)}{d} \geq \frac{\text{Tr}((W_\ell^\top W_\ell))^\ell}{d^\ell} = \frac{\|W_\ell\|_F^{2\ell}}{d^\ell} = \frac{\|W_1\|_F^{2\ell}}{d^\ell},$$

thus

$$\text{Tr}((W_\ell^\top W_\ell)^\ell) \geq \frac{\|W_1\|_F^{2\ell}}{d^{\ell-1}}. \tag{9}$$

Notice that we made use of the weak balancedness assumption $\|W_\ell\|_F = \|W_1\|_F$.

Combining inequalities (8) and (9), we get

$$\|W_1 \ldots W_\ell X\|_F \geq \sqrt{\lambda_{\min}(XX^\top)}\frac{\|W_1\|_F^\ell}{d^{\frac{\ell-1}{2}}}. \tag{10}$$

Combining this with the triangle inequality $\|W_1 \dots W_\ell X\|_F \leq \|Y\|_F + \sqrt{f(W)}$, we get the bounds

$$\|W_1\|_F^{2\ell-2} = \left(\|W_1\|_F^\ell\right)^{\frac{2\ell-2}{\ell}} \leq K^{\frac{2\ell-2}{\ell}}(\|Y\|_F + \sqrt{f(W)})^{\frac{2\ell-2}{\ell}}$$

and

$$\|W_1\|_F^{\ell-2} = \left(\|W_1\|_F^\ell\right)^{\frac{\ell-2}{\ell}} \leq K^{\frac{\ell-2}{\ell}}\|Y\|_F + \sqrt{f(W)}^{\frac{\ell-2}{\ell}},$$

where

$$K := \frac{d^{\frac{\ell-1}{2}}}{\sqrt{\lambda_{\min}(XX^\top)}}. \tag{11}$$

Plugging these in inequality (6), we get

$$\left\|\nabla^2 f(W)\right\|_2 \leq 2\ell^2 K^{\frac{2\ell-2}{\ell}}(\|Y\|_F + \sqrt{f(W)})^{\frac{2\ell-2}{\ell}}\|X\|_2^2 + 2(\ell^2-\ell)K^{\frac{\ell-2}{\ell}}(\|Y\|_F + \sqrt{f(W)})^{\frac{\ell-2}{\ell}}\|X\|_2\sqrt{f(W)}. \tag{12}$$

We now separate $\|Y\|_F$ and $\sqrt{f(W)}$. For the first exponent $p_1 := \frac{2\ell-2}{\ell} \geq 1$, we use Jensen's inequality:

$$(\|Y\|_F + \sqrt{f})^{p_1} \leq 2^{p_1-1}\left(\|Y\|_F^{p_1} + f^{\frac{p_1}{2}}\right) = 2^{p_1-1}\left(\|Y\|_F^{\frac{2\ell-2}{\ell}} + f^{\frac{\ell-1}{\ell}}\right).$$

For the second exponent $p_2 := \frac{\ell-2}{\ell} \in [0,1)$, we use sub-additivity $(a+b)^{p_2} \leq a^{p_2} + b^{p_2}$. We bound this further by $2^{p_2}(a^{p_2} + b^{p_2})$ (since $2^{p_2} \geq 1$). Thus:

$$(\|Y\|_F + \sqrt{f})^{p_2}\sqrt{f} \leq \left(\|Y\|_F^{p_2} + f^{\frac{p_2}{2}}\right)\sqrt{f} \leq 2^{p_2}\left(\|Y\|_F^{\frac{\ell-2}{\ell}}\sqrt{f} + f^{\frac{\ell-1}{\ell}}\right).$$

We absorbe the factors of $2^{p_1-1}$ and $2^{p_2}$ into $K$ to forming $2K$.

Finally we linearize the remaining nonlinearity $\sqrt{f(W)}$. Using the standard Young's inequality valid for any $\beta > 0$, we have

$$\sqrt{f(W)} \leq \frac{f(W)}{2\beta} + \frac{\beta}{2} = \frac{f(W) - f^*}{2\beta} + \left(\frac{f^*}{2\beta} + \frac{\beta}{2}\right).$$

As above, we also have $f(W)^{\frac{\ell-1}{\ell}} \leq 1 + (f(W) - f^*) + (f^*)^{\frac{\ell-1}{\ell}}$. Applying these to (12) and choosing $\beta = 1$ gives the bound $\|\nabla^2 f(W)\|_2 \leq H_0 + H_1(f(W) - f^*)$ with

$$H_0 := A_0\|X\|_2^2 + A_1\|X\|_2\left(\frac{f^*}{2} + \frac{1}{2}\right) + (A_2\|X\|_2^2 + A_3\|X\|_2)(1 + (f^*)^{\frac{\ell-1}{\ell}}),$$

$$H_1 := (A_1 + A_3)\|X\|_2 + A_2\|X\|_2^2. \tag{13}$$

The constants $A_0, A_1, A_2, A_3$ are defined as

$$A_0 := 2\ell^2(2K)^{\frac{2\ell-2}{\ell}}\|Y\|_F^{\frac{2\ell-2}{\ell}}, \qquad\qquad A_2 := 2\ell^2(2K)^{\frac{2\ell-2}{\ell}}, \tag{14}$$

$$A_1 := 2(\ell^2-\ell)(2K)^{\frac{\ell-2}{\ell}}\|Y\|_F^{\frac{\ell-2}{\ell}}, \qquad A_3 := 2(\ell^2-\ell)(2K)^{\frac{\ell-2}{\ell}} \tag{15}$$

and $K$ as in Eq. (11).

$\square$

**Proposition D.1.** *Let $f$ be defined as*

$$f(W) \equiv f(W_1, ..., W_\ell) = \|Y - \underbrace{W_1\phi_1(W_2\phi(W_3 \dots \phi_{\ell-1}(W_\ell X) \dots))}_{F}\|_F^2$$

where $\phi_i$ is leaky-ReLU activation function with slopes $1$ and $b_i$, i.e., $\phi_i(x) = \max\{b_i x, x\}$, $0 < b_i \leq 1$. Assume that over the course of GD:

- $\lambda_{\min}(W_i^\top W_i) \geq h_i > 0$, for $i = 1, \ldots, \ell - 1$.
- The layers $W_i$ are weakly balanced, i.e., $\|W_1\|_{\mathrm{F}} = \ldots = \|W_\ell\|_{\mathrm{F}}$.

Then,

i) If it holds $\|W_1\|_F^\ell \|X\|_2 < (1 - \frac{1}{\ell})\sqrt{f(W)}$, then we have $\|\nabla^2 f(W)\|_2 \leq H_0^{\text{warm-up}} + H_1^{\text{warm-up}}(f(W) - f^*)$, with $H_0^{\text{warm-up}}$ and $H_1^{\text{warm-up}}$ defined as in Eq. (23).

ii) It holds $\|\nabla^2 f(W)\|_2 \leq H_0^{\text{warm-up}} + H_1^{\text{warm-up}}(f(W) - f^*)$ for all $W$, with $H_0$ and $H_1$ defined as in Eq. (27).

*Proof.* The proof is divided into two parts, similarly to the proof of Proposition 3.2: the first obtains an upper bound for the norm of the Hessian, while the second obtains a lower bound on the loss value.

The first part in the proof of Proposition 3.2 was easy, as one has ready formulas for the Hessian. In this case, the situation is more involved and we come up with a more general process to estimate the spectral norm of the Hessian based on the gradient finite differences.

**Upper bound for the Hessian norm:** To simplify the notation, we set

$$
\begin{aligned}
Z_\ell &= W_\ell X \\
A_{\ell-1} &= \phi_{\ell-1}(Z_\ell) \\
Z_{\ell-1} &= W_{\ell-1} A_{\ell-1} \\
&\;\vdots \\
Z_2 &= W_2 A_2 \\
A_1 &= \phi_1(Z_2) \\
Z_1 &= W_1 A_1 = F.
\end{aligned}
$$

By the backpropagation algorithm for the gradient, we have that the gradient of $f$ can be computed as

$$
\frac{\partial f}{\partial W_i} = \delta_i A_i^\top
$$

where $\delta_i$ is defined recursively as

$$
\begin{aligned}
\delta_1 &= -2(Y - F) \\
\delta_2 &= W_1^\top \delta_1 \odot \phi_1'(Z_2) \\
&\;\vdots \\
\delta_i &= W_{i-1}^\top \delta_{i-1} \odot \phi_{i-1}'(Z_i).
\end{aligned}
$$

We need to upper bound the difference of the gradient defined in two distinct, sufficiently close points $W = (W_1, \ldots, W_\ell)$ and $\bar{W} = (\bar{W}_1, \ldots, \bar{W}_\ell)$. We also define

$$
\mathrm{dist}(W, \bar{W}) := \sqrt{\sum_{i=1}^\ell \|W_i - \bar{W}_i\|_{\mathrm{F}}^2}.
$$

It holds that

$$
\|\nabla f(W) - \nabla f(\bar{W})\|_{\mathrm{F}} \leq \sum_{i=1}^\ell \left\| \frac{\partial f}{\partial W_i}(W) - \frac{\partial f}{\partial W_i}(\bar{W}) \right\|_{\mathrm{F}}.
$$

We have

$$\left\| \frac{\partial f}{\partial W_i}(W) - \frac{\partial f}{\partial W_i}(\bar{W}) \right\|_{\mathrm{F}} = \|\delta_i A_i^{\top} - \bar{\delta}_i \bar{A}_i^{\top}\|_{\mathrm{F}} \leq \|\delta_i\|_{\mathrm{F}}\|A_i - \bar{A}_i\|_{\mathrm{F}} + \|\bar{A}_i\|_{\mathrm{F}}\|\delta_i - \bar{\delta}_i\|_{\mathrm{F}}. \tag{16}$$

Here we use a bar to denote the sequences of matrices related to the point $\bar{W}$. We deal with the four sequences appearing in this upper bound one by one, starting from $\bar{A}_i$. We can equivalently deal with $\bar{A}_i$ as the only difference will be to substitute $\bar{W}$ in place of $W$.

We have

$$A_i = \phi_i(W_{i+1}A_{i+1}), \text{ for } i = 1, \ldots, \ell - 2,$$

thus

$$\|A_i\|_{\mathrm{F}} = \|\phi_i(W_{i+1}A_{i+1})\|_{\mathrm{F}} \leq \|W_{i+1}A_{i+1}\|_{\mathrm{F}} = \|W_1\|_{\mathrm{F}}\|A_{i+1}\|_{\mathrm{F}}.$$

The inequality follows from the fact that $\phi_i$ is leaky-ReLU, thus $|\phi_i(x)| \leq |x|$ and the last equality by the weakly balanced assumption, i.e. that $\|W_i\|_{\mathrm{F}} = \|W_1\|_{\mathrm{F}}$.

This implies that

$$\|A_i\|_{\mathrm{F}} \leq \|W_1\|^{\ell-1-i}\|A_{\ell-1}\| = \|W_1\|^{\ell-1-i}\|\phi_{\ell-1}(W_\ell X)\|_{\mathrm{F}} \leq \|W_1\|_{\mathrm{F}}^{\ell-i}\|X\|_2. \tag{17}$$

Similarly, it holds

$$\|\bar{A}_i\|_{\mathrm{F}} \leq \|\bar{W}_1\|_{\mathrm{F}}^{\ell-i}\|X\|_2. \tag{18}$$

Now, we deal with $A_i - \bar{A}_i$:

$$\|A_i - \bar{A}_i\|_{\mathrm{F}} = \|\phi_i(W_{i+1}A_{i+1}) - \phi_i(\bar{W}_{i+1}\bar{A}_{i+1})\|_{\mathrm{F}} \leq \|W_{i+1}A_{i+1} - \bar{W}_{i+1}\bar{A}_{i+1}\|_{\mathrm{F}} \leq$$
$$\|A_{i+1}\|_{\mathrm{F}}\|W_{i+1} - \bar{W}_{i+1}\|_{\mathrm{F}} + \|\bar{W}_{i+1}\|_{\mathrm{F}}\|A_{i+1} - \bar{A}_{i+1}\|_{\mathrm{F}} \leq$$
$$\|A_{i+1}\|_{\mathrm{F}}\mathrm{dist}(W,\bar{W}) + \|\bar{W}_1\|_{\mathrm{F}}\|A_{i+1} - \bar{A}_{i+1}\|_{\mathrm{F}}.$$

By an induction argument, we can get the bound

$$\|A_i - \bar{A}_i\|_{\mathrm{F}} \leq \left( \sum_{k=i+1}^{\ell-1} \|A_k\|\|\bar{W}_1\|^{k-i-1} \right) \mathrm{dist}(W,\bar{W}) + \|\bar{W}_1\|_{\mathrm{F}}^{\ell-i-1}\|A_{\ell-1} - \bar{A}_{\ell-1}\|_{\mathrm{F}}$$

and by inequality (17), we have

$$\|A_i - \bar{A}_i\|_{\mathrm{F}} \leq \left( \sum_{k=i+1}^{\ell-1} \|W_1\|_{\mathrm{F}}^{\ell-k}\|\bar{W}_1\|^{k-i-1} \right) \mathrm{dist}(W,\bar{W})\|X\|_2 + \|\bar{W}_1\|_{\mathrm{F}}^{\ell-i-1}\|W_\ell - \bar{W}_\ell\|_{\mathrm{F}}\|X\|_2$$
$$\leq \left( \sum_{k=i+1}^{\ell-1} \|W_1\|_{\mathrm{F}}^{\ell-k}\|\bar{W}_1\|^{k-i-1} \right) \mathrm{dist}(W,\bar{W})\|X\|_2 + \|\bar{W}_1\|_{\mathrm{F}}^{\ell-i-1}\mathrm{dist}(W,\bar{W})\|X\|_2. \tag{19}$$

Now we move to $\delta_i$. It holds

$$\|\delta_i\|_{\mathrm{F}} = \|W_{i-1}^{\top}\delta_{i-1} \odot \phi_{i-1}'(Z_i)\|_{\mathrm{F}} \leq \|W_{i-1}\|_{\mathrm{F}}\|\delta_{i-1}\|_{\mathrm{F}} = \|W_1\|_{\mathrm{F}}\|\delta_{i-1}\|_{\mathrm{F}}.$$

This implies that

$$\|\delta_i\|_{\mathrm{F}} \leq \|W_1\|_{\mathrm{F}}^{i-1}\|\delta_1\|_{\mathrm{F}} = 2\|W_1\|_{\mathrm{F}}^{i-1}\sqrt{f(W)}$$

and similarly

$$\|\bar{\delta}_i\|_{\mathrm{F}} \leq 2\|\bar{W}_1\|_{\mathrm{F}}^{i-1}\sqrt{f(\bar{W})}. \tag{20}$$

For the sequence $\delta_i - \bar{\delta}_i$, we have

$$\|\delta_i - \bar{\delta}_i\|_{\mathrm{F}} = \|W_{i-1}^{\top}\delta_{i-1} \odot \phi_{i-1}'(Z_i) - \bar{W}_{i-1}^{\top}\bar{\delta}_{i-1} \odot \phi_{i-1}'(\bar{Z}_i)\|_{\mathrm{F}}$$

and since all entries of $Z_i$ are non-zero and $\bar{Z}_i$ is taken sufficiently close to $Z_i$, these two points feature the same activation pattern, thus $\phi'_{i-1}(Z_i) = \phi'_{i-1}(\bar{Z}_i)$. This gives

$$\|\delta_i - \bar{\delta}_i\|_F \leq \|W_{i-1}^\top \delta_{i-1} - \bar{W}_{i-1}^\top \bar{\delta}_{i-1}\|_F \leq \|W_{i-1}\|_F\|\delta_{i-1} - \bar{\delta}_{i-1}\|_F + \|\bar{\delta}_{i-1}\|_F\|W_{i+1} - \bar{W}_{i+1}\|_F$$
$$\leq \|W_1\|_F\|\delta_{i-1} - \bar{\delta}_{i-1}\|_F + \|\bar{\delta}_{i-1}\|_F\mathrm{dist}(W, \bar{W}).$$

By induction, we have

$$\|\delta_i - \bar{\delta}_i\|_F \leq \sum_{k=i-1}^{1} \|\bar{\delta}_k\|_F\|W_1\|_F^{i-1-k}\mathrm{dist}(W, \bar{W}) + \|W_1\|_F^{i-1}\|\delta_1 - \bar{\delta}_1\|_F$$

$$\leq 2\sqrt{f(\bar{W})} \sum_{k=i-1}^{1} \|\bar{W}_1\|_F^{k-1}\|W_1\|_F^{i-1-k}\mathrm{dist}(W, \bar{W}) + \|W_1\|_F^{i-1}\|\delta_1 - \bar{\delta}_1\|_F.$$

The second inequality in the previous derivation follows by inequality (20).

For $\|\delta_1 - \bar{\delta}_1\|_F$, we have

$$\|\delta_1 - \bar{\delta}_1\|_F = 2\|W_1 A_1 - \bar{W}_1 \bar{A}_1\|_F \leq 2\|W_1\|_F\|A_1 - \bar{A}_1\|_F + 2\|\bar{A}_1\|_F\|W_1 - \bar{W}_1\|_F \leq$$

$$2\|W_1\|_F\left(\left(\sum_{k=2}^{\ell-1}\|W_1\|_F^{\ell-k}\|\bar{W}_1\|^{k-2}\right) + \|\bar{W}_1\|_F^{\ell-2}\right)\mathrm{dist}(W, \bar{W})\|X\|_2 + 2\|\bar{W}_1\|_F^{\ell-1}\mathrm{dist}(W, \bar{W})\|X\|_2 =$$

$$2\left(\|W_1\|_F\left(\left(\sum_{k=2}^{\ell-1}\|W_1\|_F^{\ell-k}\|\bar{W}_1\|^{k-2}\right) + \|\bar{W}_1\|_F^{\ell-2}\right) + \|\bar{W}_1\|_F^{\ell-1}\right)\mathrm{dist}(W, \bar{W})\|X\|_2.$$

Thus,

$$\|\delta_i - \bar{\delta}_i\|_F \leq 2\sqrt{f(\bar{W})} \sum_{k=i-1}^{1} \|\bar{W}_1\|_F^{k-1}\|W_1\|_F^{i-1-k}\mathrm{dist}(W, \bar{W}) +$$

$$2\|W_1\|_F^{i-1}\left(\|W_1\|_F\left(\left(\sum_{k=2}^{\ell-1}\|W_1\|_F^{\ell-k}\|\bar{W}_1\|^{k-2}\right) + \|\bar{W}_1\|_F^{\ell-2}\right) + \|\bar{W}_1\|_F^{\ell-1}\right)\mathrm{dist}(W, \bar{W})\|X\|_2. \qquad (21)$$

Combining inequalities (16),(18),(19),(20) and (21), we get

$$\left\|\frac{\partial f}{\partial W_i}(W) - \frac{\partial f}{\partial W_i}(\bar{W})\right\|_F \leq$$

$$2\|\bar{W}_1\|_F^{i-1}\sqrt{f(\bar{W})}\left(\left(\sum_{k=i+1}^{\ell-1}\|W_1\|_F^{\ell-k}\|\bar{W}_1\|^{k-i-1}\right) + \|\bar{W}_1\|_F^{\ell-i-1}\right)\mathrm{dist}(W, \bar{W})\|X\|_2 +$$

$$2\|\bar{W}_1\|_F^{\ell-i}\|X\|_2\sqrt{f(\bar{W})} \sum_{k=i-1}^{1} \|\bar{W}_1\|_F^{k-1}\|W_1\|_F^{i-1-k}\mathrm{dist}(W, \bar{W}) +$$

$$2\|\bar{W}_1\|_F^{\ell-i}\|W_1\|_F^{i-1}\left(\|W_1\|_F\left(\left(\sum_{k=2}^{\ell-1}\|W_1\|_F^{\ell-k}\|\bar{W}_1\|^{k-2}\right) + \|\bar{W}_1\|_F^{\ell-2}\right) + \|\bar{W}_1\|_F^{\ell-1}\right)\mathrm{dist}(W, \bar{W})\|X\|_2^2,$$

thus

$$\frac{\left\|\frac{\partial f}{\partial W_i}(W) - \frac{\partial f}{\partial W_i}(\bar{W})\right\|_{\mathrm{F}}}{\mathrm{dist}(W, \bar{W})} \leq$$

$$2\|\bar{W}_1\|_{\mathrm{F}}^{i-1}\sqrt{f(\bar{W})}\left(\left(\sum_{k=i+1}^{\ell-1}\|W_1\|_{\mathrm{F}}^{\ell-k}\|\bar{W}_1\|^{k-i-1}\right) + \|\bar{W}_1\|_{\mathrm{F}}^{\ell-i-1}\right)\|X\|_2 +$$

$$2\|\bar{W}_1\|_{\mathrm{F}}^{\ell-i}\|X\|_2\sqrt{f(\bar{W})}\sum_{k=i-1}^{1}\|\bar{W}_1\|_{\mathrm{F}}^{k-1}\|W_1\|_{\mathrm{F}}^{i-1-k} +$$

$$2\|\bar{W}_1\|_{\mathrm{F}}^{\ell-i}\|W_1\|_{\mathrm{F}}^{i-1}\left(\|W_1\|_{\mathrm{F}}\left(\left(\sum_{k=2}^{\ell-1}\|W_1\|_{\mathrm{F}}^{\ell-k}\|\bar{W}_1\|^{k-2}\right) + \|\bar{W}_1\|_{\mathrm{F}}^{\ell-2}\right) + \|\bar{W}_1\|_{\mathrm{F}}^{\ell-1}\right)\|X\|_2^2$$

and taking the limit as $\bar{W} \longrightarrow W$, we get

$$\lim_{\bar{W}\to W}\frac{\left\|\frac{\partial f}{\partial W_i}(W) - \frac{\partial f}{\partial W_i}(\bar{W})\right\|_{\mathrm{F}}}{\mathrm{dist}(W, \bar{W})} \leq$$
$$2(\ell-i)\|X\|_2\|W_1\|_{\mathrm{F}}^{\ell-2}\sqrt{f(W)} + 2(i-1)\|X\|_2\|W_1\|_{\mathrm{F}}^{\ell-2}\sqrt{f(W)} + 2(\ell-1)\|W_1\|_{\mathrm{F}}^{2\ell-2}\|X\|_2^2 =$$
$$2(\ell-1)\|X\|_2\sqrt{f(W)}\|W_1\|_{\mathrm{F}}^{\ell-2} + 2\ell\|W_1\|_{\mathrm{F}}^{2\ell-2}\|X\|_2^2.$$

This is because, when $\bar{W} \longrightarrow W$, it holds $\bar{W}_1 \longrightarrow W_1$.

For the total gradient difference, we have

$$\lim_{\bar{W}\to W}\frac{\left\|\nabla f(W) - \nabla f(\bar{W})\right\|_{\mathrm{F}}}{\mathrm{dist}(W, \bar{W})} \leq \sum_{i=1}^{\ell}\lim_{\bar{W}\to W}\frac{\left\|\frac{\partial f}{\partial W_i}(W) - \frac{\partial f}{\partial W_i}(\bar{W})\right\|_{\mathrm{F}}}{\mathrm{dist}(W, \bar{W})}$$
$$\leq 2\ell(\ell-1)\|W_1\|_{\mathrm{F}}^{\ell-2}\|X\|_2\sqrt{f(W)} + 2\ell^2\|W_1\|_{\mathrm{F}}^{2\ell-2}\|X\|_2^2.$$

It holds

$$\|\nabla^2 f(W)\|_2 = \lim_{\bar{W}\to W}\frac{\left\|\nabla f(W) - \nabla f(\bar{W})\right\|_{\mathrm{F}}}{\mathrm{dist}(W, \bar{W})},$$

thus

$$\|\nabla^2 f(W)\|_2 \leq 2\ell(\ell-1)\|W_1\|_{\mathrm{F}}^{\ell-2}\|X\|_2\sqrt{f(W)} + 2\ell^2\|W_1\|_{\mathrm{F}}^{2\ell-2}\|X\|_2^2. \tag{22}$$

Notice that this is the same upper bound as the one provided in (6).

Thus, if $\|W\|_F^{\ell}\|X\|_2 \leq (1 - \frac{1}{\ell})\sqrt{f(W)}$, then we get $\|\nabla^2 f(W)\| \leq H_0^{\text{warm-up}} + H_1^{\text{warm-up}}(f(W) - f^*)$ with exactly the same constants as in the previous proposition, i.e.

$$H_0^{\text{warm-up}} := 4(\ell^2 - \ell)\left(1 - \frac{1}{\ell}\right)^{\frac{\ell-2}{\ell}}\|X\|_2^{\frac{2}{\ell}}(1 + (f^*)^{\frac{\ell-1}{\ell}}), \qquad H_1^{\text{warm-up}} := 4(\ell^2 - \ell)\left(1 - \frac{1}{\ell}\right)^{\frac{\ell-2}{\ell}}\|X\|_2^{\frac{2}{\ell}}. \tag{23}$$

We now move to a global bound and for that we need a global lower bound for the loss value. For $i = 1, \ldots, \ell - 2$, we have

$$\|W_i A_i\|_F^2 \geq \lambda_{\min}(W_i^T W_i)\|A_i\|_F^2 \geq h_i\|\phi_i(W_{i+1}A_{i+1})\|_F^2 \geq h_i b_i^2\|W_{i+1}A_{i+1}\|_F^2$$

and by induction,

$$
\begin{aligned}
\|W_1 A_1\|_F^2 &\geq \left(\Pi_{i=1}^{\ell-2} h_i b_i^2\right) \|W_{\ell-1} A_{\ell-1}\|_F^2 \\
&\geq \left(\Pi_{i=1}^{\ell-2} h_i b_i^2\right) \lambda_{\min}(W_{\ell-1}^T W_{\ell-1}) \|A_{\ell-1}\|_F^2 \\
&= \left(\Pi_{i=1}^{\ell-2} h_i b_i^2\right) \lambda_{\min}(W_{\ell-1}^T W_{\ell-1}) \|\phi_{\ell-1}(W_\ell X)\|_F^2 \\
&\geq \left(\Pi_{i=1}^{\ell-2} h_i b_i^2\right) h_{\ell-1} b_{\ell-1}^2 \lambda_{\min}(XX^T) \|W_\ell\|_F^2 \\
&= \left(\Pi_{i=1}^{\ell-2} h_i b_i^2\right) h_{\ell-1} b_{\ell-1}^2 \lambda_{\min}(XX^T) \|W_1\|_F^2 \\
&= \left(\Pi_{i=1}^{\ell-1} h_i b_i^2\right) \lambda_{\min}(XX^T) \|W_1\|_F^2.
\end{aligned}
\tag{24}
$$

We have repeatedly used the assumption that $\lambda_{\min}(W_i W_i^T) \geq h_i$ and that

$$
\|\phi_i(S)\|_F^2 \geq b_i^2 \|S\|_F^2,
$$

for any matrix $S$.

To derive inequality (24), we also used the weak balancedness assumption, that is, all $\|W_i\|_F$ are equal.

Defining

$$
K_{gd} := \left(\lambda_{\min}(XX^\top) \prod_{i=1}^{\ell-1} h_i b_i^2\right)^{-\frac{1}{2}}
\tag{25}
$$

we get

$$
\|W_1\|_F \leq K_{gd} \|W_1 A_1\|_F.
$$

By the triangle inequality, $\|W_1 A_1\|_F \leq \|Y\|_F + \sqrt{f(W)}$. Thus:

$$
\|W_1\|_F \leq K_{gd}\left(\|Y\|_F + \sqrt{f(W)}\right).
\tag{26}
$$

Now we substitute this into the sharp Hessian bound (22).

**Term 1:**

$$
2\ell^2 \|W_1\|_F^{2\ell-2} \|X\|_2^2 \leq 2\ell^2 K_{gd}^{2\ell-2}\left(\|Y\|_F + \sqrt{f}\right)^{2\ell-2} \|X\|_2^2 \leq 2\ell^2 K_{gd}^{2\ell-2} 2^{2\ell-3}\left(\|Y\|_F^{2\ell-2} + f^{\ell-1}\right)\|X\|_2^2.
$$

(Using Jensen's inequality $(a+b)^p \leq 2^{p-1}(a^p + b^p)$ with $p = 2\ell - 2$).

**Term 2:**

$$
\begin{aligned}
2(\ell^2 - \ell) \|W_1\|_F^{\ell-2} \|X\|_2 \sqrt{f} &\leq 2(\ell^2 - \ell) K_{gd}^{\ell-2}\left(\|Y\|_F + \sqrt{f}\right)^{\ell-2} \|X\|_2 \sqrt{f} \\
&\leq 2(\ell^2 - \ell) K_{gd}^{\ell-2} 2^{\ell-3}\left(\|Y\|_F^{\ell-2} + f^{\frac{\ell-2}{2}}\right)\|X\|_2 \sqrt{f} \\
&= 2(\ell^2 - \ell) K_{gd}^{\ell-2} 2^{\ell-3}\left(\|Y\|_F^{\ell-2} \sqrt{f} + f^{\frac{\ell-1}{2}}\right)\|X\|_2.
\end{aligned}
$$

We now use the inequality $u \leq 1 + u^p$, which holds for all $u \geq 0$ and $p \geq 1$. In our case, it translates to

$$
\sqrt{f} \leq 1 + f^{\ell-1}
$$

and

$$
f^{\frac{\ell-1}{2}} \leq 1 + f^{\ell-1}.
$$

Term 2 can be bounded as

$$
2(\ell^2 - \ell) K_{gd}^{\ell-2} 2^{\ell-3}\left(\|Y\|_F^{\ell-2} \sqrt{f} + f^{\frac{\ell-1}{2}}\right)\|X\|_2 \leq 2(\ell^2 - \ell) K_{gd}^{\ell-2} 2^{\ell-3}\left(1 + \|Y\|_F^{\ell-2}\right)\|X\|_2\left(1 + f^{\ell-1}\right) =
$$

$$
2(\ell^2 - \ell) K_{gd}^{\ell-2} 2^{\ell-3}\left(1 + \|Y\|_F^{\ell-2}\right)\|X\|_2 + 2(\ell^2 - \ell) K_{gd}^{\ell-2} 2^{\ell-3}\left(1 + \|Y\|_F^{\ell-2}\right)\|X\|_2 f^{\ell-1}.
$$

Summing with the bound of Term 1 yields the inequality $\|\nabla^2 f(W)\|_2 \le H_0 + H_1(f(W) - f^*)$, with

$$H_0 := 2\ell^2 K_{gd}^{2\ell-2} 2^{2\ell-3} \|Y\|_F^{2\ell-2} \|X\|_2^2 + 2(\ell^2 - \ell) K_{gd}^{\ell-2} 2^{\ell-3}(1 + \|Y\|_F^{\ell-2}) \|X\|_2,$$
$$H_1 := 2\ell^2 K_{gd}^{2\ell-2} 2^{2\ell-3} \|X\|_2^2 + 2(\ell^2 - \ell) K_{gd}^{\ell-2} 2^{\ell-3}(1 + \|Y\|_F^{\ell-2}) \|X\|_2, \tag{27}$$

where $K_{gd}$ is defined as in Eq. (25).

$\square$

**Proposition 3.3.** *Consider a 2-layer non-linear model with cross-entropy loss and L2 regularization:*

$$f(W) \equiv f(W_1, W_2) =$$
$$- Y \log(P)^\top - (\mathbb{1} - Y) \log(\mathbb{1} - P)^\top$$
$$+ \frac{\lambda_1}{2} \|W_1\|_F^2 + \frac{\lambda_2}{2} \|W_2\|_F^2,$$

*where $Y \in \mathbb{R}^{1\times m}$ are true labels, and $P = \sigma(W_1 \phi(W_2 X))$ is the output of the model with the activation function $\phi$ such that $|\phi(s)| \le C_1|s|, |\phi'(s)| \le C_2$ and $|\phi''(s)| \le C_3$ for all $s \in \mathbb{R}$, sigmoid function $\sigma$, and weight matrices $W_1 \in \mathbb{R}^{1\times n_1}, W_2 \in \mathbb{R}^{n_1 \times d}$. Then, it holds*

*(i) If $\|W_1\|_F, \|W_2\|_F$ are sufficiently small compared to the loss value in the sense of Eq. (29), then $(H_0, H_1)$-smoothness holds with constants defined as in Eq. (30).*

*(ii) For all $W$, $(H_0, H_1)$-smoothness holds for constants $H_0$ and $H_1$ defined as in Eq. (31).*

*Proof.* We start by calculating the gradients and Hessians of $f$. The Hessian of the regularization part is just $(\lambda_1 + \lambda_2)I$. We denote the main part of the loss as

$$\bar{f}(W) = -Y \log(P)^\top - (\mathbb{1} - Y) \log(\mathbb{1} - P)^\top.$$

Again, it holds

$$\|\nabla^2 f(W)\|_2 \le \|\nabla^2 \bar{f}(W)\|_2 + (\lambda_1 + \lambda_2).$$

Some useful notation is

$$A := W_2 X$$
$$H := \phi(A)$$
$$Z := W_1 H$$
$$P := \sigma(Z).$$

The gradient of $\bar{L}$ with respect to $\text{vec}(W_1)$ is

$$\frac{\partial \bar{f}}{\partial Z} \cdot \frac{\partial Z}{\partial \text{vec}(W_1)}.$$

It holds

$$\frac{\partial \bar{f}}{\partial P} = -Y \odot \frac{1}{P} + (\mathbb{1} - Y) \odot \frac{1}{\mathbb{1} - P}$$

where $1/\text{vector}$ is used to denote entry-wise inversion.

We also have

$$\frac{\partial P}{\partial Z} = \sigma'(Z) = P \odot (\mathbb{1} - P).$$

Thus,

$$\frac{\partial \bar{f}}{\partial Z} = \frac{\partial \bar{f}}{\partial P} \odot \frac{\partial P}{\partial Z} = P - Y.$$

We denote the vectorized form of this term by $R$ since it plays the role of a residual. Since $P - Y$ is a row vector, its vectorized form is just its transpose, however, we will often keep the standard form $R = \text{vec}(P - Y)$ to ensure compatibility with previous calculations.

It holds

$$\frac{\partial \bar{f}}{\partial \text{vec}(W_1)} = \frac{\partial \bar{f}}{\partial Z} \frac{\partial Z}{\partial \text{vec}(W_1)} = R^\top H^\top = R^\top \phi(W_2 X)^\top.$$

This is a row vector, thus we transpose it to bring it to column form:

$$\frac{\partial \bar{f}}{\partial \text{vec}(W_1)} = HR = \text{vec}((P - Y)H^\top) = \text{vec}((P - Y)\phi(W_2 X)^\top)$$

For the partial derivative with respect to $\text{vec}(W_2)$, we have

$$\frac{\partial \bar{f}}{\partial \text{vec}(W_2)} = \frac{\partial \bar{f}}{\partial Z} \cdot \frac{\partial Z}{\partial \text{vec}(W_2)} = R^\top \frac{\partial Z}{\partial \text{vec}(W_2)}$$

and

$$\frac{\partial R}{\partial \text{vec}(W_2)} = -(I_m \otimes W_1)\frac{\partial \text{vec}(\phi(W_2 X))}{\partial \text{vec}(W_2)} = -(I_m \otimes W_1)\frac{\partial \text{vec}(\phi(W_2 X))}{\partial \text{vec}(W_2 X)} \frac{\partial \text{vec}(W_2 X)}{\partial \text{vec}(W_2)}$$

$\frac{\partial \text{vec}(\phi(W_2 X))}{\partial \text{vec}(W_2)}$ is the diagonal matrix $\text{diag}(\text{vec}(\phi'(W_2 X)))$.

Since $\text{vec}(W_2 X) = (X^\top \otimes I_{n_1})\text{vec}(W_2)$, the gradient $\frac{\partial \text{vec}(W_2 X)}{\partial \text{vec}(W_2)}$ is

$$\frac{\partial \text{vec}(W_2 X)}{\partial \text{vec}(W_2)} = X^\top \otimes I_{n_1}.$$

Putting it all together, we have

$$\frac{\partial f}{\partial \text{vec}(W_2)} = R^\top (I_m \otimes W_1)\text{diag}(\text{vec}(\phi'(W_2 X)))(X^\top \otimes I_{n_1}).$$

Writing that again as column vector yields

$$(X \otimes I_{n_1})\text{diag}(\text{vec}(\phi'(W_2 X)))(I_m \otimes W_1^\top)R.$$

After some modifications, we can write

$$\begin{aligned}
\text{diag}(\text{vec}(\phi'(W_2 X)))(I_m \otimes W_1^\top)R &= \\
\text{diag}(\text{vec}(\phi'(W_2 X)))\text{vec}(W_1^\top(P - Y)) &= \\
\text{vec}(W_1^\top(P - Y) \odot \phi'(W_2 X)). &
\end{aligned}$$

where $\odot$ is the Hadamard product.

This means that we can write the previous gradient as

$$\text{vec}(((W_1^\top(P - Y)) \odot \phi'(W_2 X))X^\top).$$

We now move to the calculation of the Hessian.

For the first block, we have

$$\frac{\partial^2 \bar{f}}{\partial \text{vec}(W_1)\text{vec}(W_1)^\top} = \phi(W_2 X)\frac{\partial R}{\partial \text{vec}(W_1)^\top}$$

$$= \phi(W_2 X)\frac{\partial \text{vec}(P - Y)}{\partial \text{vec}(W_1)^\top}$$

$$= \phi(W_2 X)\text{diag}(P \odot (\mathbb{1} - P))\phi(W_2 X)^\top.$$

For the off-diagonal blocks, it suffices to compute one of them, as they are symmetric.

We use the product rule (see (Magnus, 1985), Theorem 9)

$$\frac{\partial \text{vec}(A(W)B(W))}{\partial \text{vec}(W)^\top} = (B(W)^\top \otimes I)\frac{\partial \text{vec}(A(W))}{\partial \text{vec}(W)^\top} + (I \otimes A(W))\frac{\partial \text{vec}(B(W))}{\partial \text{vec}(W)^\top}.$$

We have

$$\frac{\partial}{\partial \text{vec}(W_2)^\top}\frac{\partial \bar{f}}{\partial \text{vec}(W_1)} = (\phi(W_2 X) \otimes I_1)\frac{\partial \text{vec}(P - Y)}{\partial \text{vec}(W_2)^\top}$$

$$+ (I_{n_1} \otimes (P - Y))\frac{\partial \text{vec}(\phi(W_2 X)^\top)}{\partial \text{vec}(W_2)^\top}.$$

In order to proceed, we need to write $\text{vec}(\phi(W_2 X)^\top)$ in terms of $\text{vec}(\phi(W_2 X))$, and this can be done formally using the so-called commutation matrix:

$$\text{vec}(\phi(W_2 X)^\top) = K_{n_1 m}\text{vec}(\phi(W_2 X)).$$

For the first partial derivative in the sum, we have

$$\frac{\partial \text{vec}(P - Y)}{\partial \text{vec}(W_2)^\top} = \frac{\partial \text{vec}(P)}{\partial \text{vec}(Z)}\frac{\partial \text{vec}(Z)}{\partial \text{vec}(W_2)^\top}$$

$$= \text{diag}(P \odot (\mathbb{1} - P))\frac{\partial \text{vec}(W_1 \phi(W_2 X))}{\partial \text{vec}(W_2)^\top}$$

$$= \text{diag}(P \odot (\mathbb{1} - P))(I_m \otimes W_1)\frac{\partial \text{vec}(\phi(W_2 X))}{\partial \text{vec}(W_2)^\top}$$

$$= \text{diag}(P \odot (\mathbb{1} - P))(I_m \otimes W_1)\text{diag}(\text{vec}(\phi'(W_2 X)))\frac{\partial \text{vec}(W_2 X)}{\partial \text{vec}(W_2)^\top}$$

$$= \text{diag}(P \odot (\mathbb{1} - P))(I_m \otimes W_1)\text{diag}(\text{vec}(\phi'(W_2 X)))(X^\top \otimes I_{n_1}).$$

As it is evident in the previous calculation

$$\frac{\partial \text{vec}(\phi(W_2 X))}{\partial \text{vec}(W_2)^\top} = \text{diag}(\text{vec}(\phi'(W_2 X)))(X^\top \otimes I_{n_1}).$$

Putting it all together, we get

$$\frac{\partial^2 \bar{f}}{\partial \text{vec}(W_1)\text{vec}(W_2)^\top} = \phi(W_2 X)\text{diag}(P \odot (\mathbb{1} - P)(I_m \otimes W_1)\text{diag}(\text{vec}(\phi'(W_2 X)))(X^\top \otimes I_{n_1})$$

$$+ (I_{n_1} \otimes (P - Y))K_{n_1 m}\text{diag}(\text{vec}(\phi'(W_2 X)))(X^\top \otimes I_{n_1})$$

$$= (\phi(W_2 X)\text{diag}(P \odot (\mathbb{1} - P))(I_m \otimes W_1)$$

$$+ (I_{n_1} \otimes (P - Y)K_{n_1 m}))\text{diag}(\text{vec}(\phi'(W_2 X)))(X^\top \otimes I_{n_1}).$$

We conclude with the calculation of the last block. To differentiate $\text{vec}(((W_1^\top R) \odot \phi'(W_2 X))X^\top)$, we can use the product rule for the Hadamard product, see (Magnus, 1985) (Theorem 10):

$$\frac{\partial \mathrm{vec}((W_1^\top R) \odot \phi'(W_2 X))}{\partial \mathrm{vec}(W_2)^\top} = \mathrm{diag}(\mathrm{vec}(\phi'(W_2 X)) \frac{\partial \mathrm{vec}(W_1^\top R)}{\partial \mathrm{vec}(W_2)^\top} + \mathrm{diag}(\mathrm{vec}(W_1^\top R)) \frac{\partial \phi'(W_2 X)}{\partial \mathrm{vec}(W_2)^\top}.$$

For the first term of the last sum, we have by previous calculations that

$$\frac{\partial \mathrm{vec}(W_1^\top R)}{\partial \mathrm{vec}(W_2)^\top} = (I_m \otimes W_1^\top)\mathrm{diag}(P \odot (\mathbb{1} - P))(I_m \otimes W_1)\mathrm{diag}(\mathrm{vec}(\phi'(W_2 X)))(X^\top \otimes I_{n_1}).$$

For the second term of the last sum, we have

$$\frac{\partial \phi'(W_2 X)}{\partial \mathrm{vec}(W_2)^\top} = \mathrm{diag}(\mathrm{vec}(\phi''(W_2 X)))(X^\top \otimes I_{n_1}).$$

In total, we have

$$\begin{aligned}
\frac{\partial^2 \bar{f}}{\partial \mathrm{vec}(W_2)\mathrm{vec}(W_2)^\top} =& (X \otimes I_{n_1})\mathrm{diag}(\mathrm{vec}(\phi'(W_2 X)))(I_m \otimes W_1^\top)\mathrm{diag}(P \odot (\mathbb{1} - P)) \\
& (I_m \otimes W_1)\mathrm{diag}(\mathrm{vec}(\phi'(W_2 X)))(X^\top \otimes I_{n_1}) \\
& + (X \otimes I_{n_1})\mathrm{diag}(\mathrm{vec}(W_1^\top R))\mathrm{diag}(\mathrm{vec}(\phi''(W_2 X)))(X^\top \otimes I_{n_1}).
\end{aligned}$$

This completes the calculation of all four blocks of the Hessian of $\bar{f}$.

To upper bound $\|\nabla^2 \bar{f}(W)\|_2$, we can write

$$\|\nabla^2 \bar{f}(W)\|_2 \le \left\| \frac{\partial^2 \bar{f}}{\partial \mathrm{vec}(W_1)\mathrm{vec}(W_1)^\top} \right\|_2 + 2 \left\| \frac{\partial^2 \bar{f}}{\partial \mathrm{vec}(W_1)\mathrm{vec}(W_2)^\top} \right\|_2 + \left\| \frac{\partial^2 \bar{f}}{\partial \mathrm{vec}(W_2)\mathrm{vec}(W_2)^\top} \right\|_2.$$

It holds

$$\begin{aligned}
\left\| \frac{\partial^2 \bar{f}}{\partial \mathrm{vec}(W_1)\mathrm{vec}(W_1)^\top} \right\|_2 &\le \|\mathrm{diag}(P \odot (\mathbb{1} - P))\|_2 \|\phi(W_2 X)\phi(W_2 X)^\top\|_2 \\
&\le \|\mathrm{diag}(P \odot (\mathbb{1} - P))\|_2 \|\phi(W_2 X)\phi(W_2 X)^\top\|_\mathrm{F} \le C_1^2 \|W_2\|_\mathrm{F}^2 \|X\|_2^2,
\end{aligned}$$

since all entries of $P \odot (\mathbb{1} - P)$ are upper bounded by 1 in absolute value.

For the off-diagonal blocks, it holds

$$\begin{aligned}
\left\| \frac{\partial^2 \bar{f}}{\partial \mathrm{vec}(W_1)\mathrm{vec}(W_2)^\top} \right\|_2 &\le (\|\phi(W_2 X)\|_2 \|W_1\|_2 + \|P - Y\|_2)C_2 \|X\|_2 \\
&\le C_2(C_1 \|W_1\|_\mathrm{F} \|W_2\|_\mathrm{F} \|X\|_2 + \|P - Y\|_\mathrm{F})\|X\|_2 \\
&\le C_2(C_1 \|W_1\|_\mathrm{F} \|W_2\|_\mathrm{F} \|X\|_2 + \sqrt{2f(W)})\|X\|_2
\end{aligned}$$

and

$$\begin{aligned}
\left\| \frac{\partial^2 \bar{f}}{\partial \mathrm{vec}(W_2)\mathrm{vec}(W_2)^\top} \right\|_2 &\le \|X\|_2^2 C_2^2 \|W_1^\top\|_2 \|W_1\|_2 + \|X\|_2^2 C_3 \|W_1^\top(P - Y)\|_2 \\
&\le \|X\|_2^2 C_2^2 \|W_1\|_\mathrm{F}^2 + \|X\|_2^2 C_3 \|W_1\|_\mathrm{F} \|P - Y\|_\mathrm{F} \\
&\le \|X\|_2^2 C_2^2 \|W_1\|_\mathrm{F}^2 + \|X\|_2^2 C_3 \|W_1\|_\mathrm{F} \sqrt{2f(W)}
\end{aligned}$$

In the previous inequalities we used the standard inequality $\|P - Y\|_F \le \sqrt{2\bar{f}(W)} \le \sqrt{2f(W)}$.

We define the structural constants:

$$A_1 := C_2^2 \|X\|_2^2,$$
$$A_2 := C_1^2 \|X\|_2^2,$$
$$A_{12} := 2C_1 C_2 \|X\|_2^2,$$
$$B_{\text{mix}} := \sqrt{2} C_3 \|X\|_2^2,$$
$$B_{\text{res}} := 2\sqrt{2} C_2 \|X\|_2. \tag{28}$$

Using the explicit block-wise bounds, we get that the Hessian spectral norm is bounded by:

$$\left\| \nabla^2 f(W) \right\|_2 \le (\lambda_1 + \lambda_2) + T_{\text{quad}}(W) + T_{\text{mixed}}(W) + T_{\text{res}}(W),$$

where:

$$T_{\text{quad}}(W) := A_1 \|W_1\|_F^2 + A_2 \|W_2\|_F^2 + A_{12} \|W_1\|_F \|W_2\|_F,$$
$$T_{\text{mixed}}(W) := B_{\text{mix}} \|W_1\|_F \sqrt{\bar{f}(W)},$$
$$T_{\text{res}}(W) := B_{\text{res}} \sqrt{\bar{f}(W)}.$$

For (i), the condition

$$A_1 \|W_1\|^2 + A_2 \|W_2\|^2 + A_{12} \|W_1\| \|W_2\| \le B_{\text{mix}} \|W_1\| \sqrt{f} + B_{\text{res}} \sqrt{f} \tag{29}$$

implies $T_{\text{quad}}(W) \le T_{\text{mixed}}(W) + T_{\text{res}}(W)$. Since weight norms are non-negative, we can also drop the terms involving $\|W_2\|$ to find a bound for $\|W_1\|$:

$$A_1 \|W_1\|_F^2 - (B_{\text{mix}} \sqrt{f}) \|W_1\|_F - (B_{\text{res}} \sqrt{f}) \le 0.$$

Solving this quadratic inequality for $\|W_1\|_F$ (finding the positive root) gives

$$\|W_1\|_F \le \frac{B_{\text{mix}}}{A_1} \sqrt{f} + \sqrt{\frac{B_{\text{res}}}{A_1}} f^{1/4}.$$

Substituting this back into $T_{\text{mixed}} = B_{\text{mix}} \|W_1\|_F \sqrt{f}$ yields

$$T_{\text{mixed}} \le \frac{B_{\text{mix}}^2}{A_1} f + B_{\text{mix}} \sqrt{\frac{B_{\text{res}}}{A_1}} f^{3/4}.$$

The total bound is $(\lambda_1 + \lambda_2) + 2(T_{\text{mixed}} + T_{\text{res}})$. We linearize $\sqrt{f}$ and $f^{3/4}$ (using $\sqrt{f} \le (f - f^*) + f^* + 1$ and $f^{3/4} \le \frac{3}{4}(f - f^*) + \frac{3}{4} f^* + \frac{1}{4}$) and group coefficients to obtain

$$H_0^{\text{warm-up}} := (\lambda_1 + \lambda_2) + 2 \left( B_{\text{res}} \left( \frac{f^*}{2} + \frac{1}{2} \right) + \frac{B_{\text{mix}}^2}{A_1} f^* + B_{\text{mix}} \sqrt{\frac{B_{\text{res}}}{A_1}} \left( \frac{3}{4} f^* + \frac{1}{4} \right) \right),$$

$$H_1^{\text{warm-up}} := 2 \left( B_{\text{res}} + \frac{B_{\text{mix}}^2}{A_1} + \frac{3}{4} B_{\text{mix}} \sqrt{\frac{B_{\text{res}}}{A_1}} \right). \tag{30}$$

We proceed now to the global bound (ii). It holds

$$A_{12} \|W_1\| \|W_2\| \le \frac{A_{12}}{2} \|W_1\|^2 + \frac{A_{12}}{2} \|W_2\|^2.$$

and

$$T_{\text{mixed}} = B_{\text{mix}} \|W_1\| \sqrt{f} \le \frac{B_{\text{mix}}}{2} \|W_1\|^2 + \frac{B_{\text{mix}}}{2} f.$$

By the definition of the function $f$, we can bound the weight norms as $\|W_i\|_F^2 \leq \frac{2f}{\lambda_i}$. This implies

$$T_{\text{quad}} \leq \left( \frac{2A_1 + A_{12}}{\lambda_1} + \frac{2A_2 + A_{12}}{\lambda_2} \right) f.$$

$$T_{\text{mixed}} \leq \left( \frac{B_{\text{mix}}}{\lambda_1} + \frac{B_{\text{mix}}}{2} \right) f.$$

For the $T_{\text{res}}$, we bound it as

$$T_{\text{res}} \leq B_{\text{res}} \sqrt{f(W)} \leq B_{\text{res}}((f(W) - f^*) + f^* + 1).$$

In total, we have

$$\|\nabla^2 f(W)\| \leq H_0 + H_1(f(W) - f^*)$$

with

$$H_0 := (\lambda_1 + \lambda_2) + B_{\text{res}}(f^* + 1),$$

$$H_1 := B_{\text{res}} + \left( \frac{2A_1 + A_{12} + B_{\text{mix}}}{\lambda_1} + \frac{2A_2 + A_{12}}{\lambda_2} + \frac{B_{\text{mix}}}{2} \right). \tag{31}$$

$\square$

# E. Missing proofs from Section 3.1.2

First, we describe the setup in more detail. The input data is encoded into a single matrix $Z_0 \in \mathbb{R}^{(d+1)\times(n+1)}$. This matrix contains $n$ training tokens and one query token. The training tokens cover the first $n$ columns of the matrix, while the query token the last one. The label of the query's feature is initialized at $0$.

$$Z_0 = \begin{bmatrix} x_1 & x_2 & \cdots & x_n & x_{\text{query}} \\ y_1 & y_2 & \cdots & y_n & 0 \end{bmatrix} \in \mathbb{R}^{(d+1)\times(n+1)}.$$

The model's objective is to predict the true value for this entry. The model is defined by

$$Z_1 = Z_0 + \frac{1}{n} P Z_0 M \cdot \phi(Z_0^T Q Z_0),$$

where the trainable parameters are $P$ and $Q$ and $M = \text{diag}(1, 1, \ldots, 1, 0)$ is a fixed mask and $\phi$ is a general activation applied to the attention scores. The output of the model is

$$\hat{y} = [Z_1]_{(d+1),(n+1)}$$

and the cost function for one task with true target $y_{\text{true}}$ is

$$f(P, Q) = \ell(\hat{y}, y_{\text{true}}),$$

where $\ell$ is some loss function (MSE for continuous variables or cross-entropy for binary ones). The most interesting property of transformers is their ability to learn in-context, i.e., minimize an in-context cost function defined below.

**Definition E.1.** *Let $D_x$ be a distribution over an input space $X$, $H$ a set of functions $X \to Y$, and $D_H$ a distribution over functions in $H$. Let $\ell : Y \times Y \to \mathbb{R}$ be a loss function, $S = \{(x_1, y_1, \ldots, x_n, y_n) : x_i \in X, y_i \in Y\}$ be the set of finite-length sequences of $(x, y)$ pairs and $F_\theta = \{f_\theta : S \times X \to Y, \theta \in \Theta\}$ be a class of functions parameterized by $\theta$ in some set $\Theta$. For $n > 0$, we say that a model $f : S \times X \to Y$ is trained on in-context examples of functions in $H$ under loss $\ell$ w.r.t. $(D_x, D_H)$ if $f = f_\theta$, where $\theta$ minimizes minimizes*

$$\mathbb{E}_{j=(x_1, h(x_1), \ldots, x_n, h(x_n), x_{query})} [\ell(f_\theta(j), h(x_{query}))]$$

*where $x_i, x_{query}$ are chosen i.i.d. from $D_x$ and $h \sim D_H$ is independent. $j$ represents a prompt.*

**Proposition E.1.** *Consider the aforementioned $1$-layer transformer model with a regularized loss:*

$$f(P, Q) = (\hat{y} - y_{true})^2 + \frac{\lambda_P}{2}\|P\|_F^2 + \frac{\lambda_Q}{2}\|Q\|_F^2.$$

*Assume that the input $Z$ is bounded, and the activation function and its derivatives are globally bounded:*

- $\|Z\|_2 \leq C_Z$
- $\|\phi(x)\| \leq C_1\|x\|$
- $\|\phi'(x)\| \leq C_2$
- $\|\phi''(x)\| \leq C_3$.

*Then, it holds*

*(i) If $K_1\|Q\|_F^2 + K_2\|P\|_F\|Q\|_F + K_4\|P\|_2^F \leq \sqrt{2}K_5\|P\|_F\sqrt{f(P,Q)} + \sqrt{2}K_3\sqrt{f(P,Q)}$, we have*

$$\|\nabla^2 f(P,Q)\|_2 \leq H_0^{warm\text{-}up} + H_1^{warm\text{-}up}(f(P,Q) - f^*),$$

*where $H_0^{warm\text{-}up}$ and $H_1^{warm\text{-}up}$ are defined as in Eq. (34).*

*(ii) For all $(P, Q)$ it holds*

$$\|\nabla^2 f(P,Q)\|_2 \leq H_0 + H_1(f(P,Q) - f^*),$$

*where $H_0$ and $H_1$ are defined as in Eq. (35).*

*Proof.* To compute the Hessian, we first vectorize the parameters and the model. For simplicity, we set $k = d + 1$ and $m = n + 1$.

- **Parameters:** $p = \text{vec}(P) \in \mathbb{R}^{k^2 \times 1}$ and $q = \text{vec}(Q) \in \mathbb{R}^{k^2 \times 1}$.
- **Constants:** Let $V = ZM \in \mathbb{R}^{k \times m}$ and $E = e_k e_m^T \in \mathbb{R}^{k \times m}$.
- **Intermediates:** $S(Q) = Z^T QZ$ and $A(Q) = \phi(S(Q))$.

Using the trace identity $e_k^T X e_m = \text{Tr}(e_m e_k^T X) = \text{Tr}(E^T X)$, we rewrite $\hat{y}$:

$$\hat{y} = \frac{1}{n}\text{Tr}(E^T PVA(Q)) = \frac{1}{n}\text{Tr}((VA(Q)E^T)P).$$

Using $\text{Tr}(\mathbf{A}^T\mathbf{B}) = \text{vec}(\mathbf{A})^T\text{vec}(\mathbf{B})$, we get:

$$\hat{y} = \frac{1}{n}\text{vec}((VA(Q)E^T)^T)^T\text{vec}(P) = \frac{1}{n}\text{vec}(EA(Q)^T V^T)^T p.$$

Let $b(q) = \text{vec}(EA(Q)^T V^T)$. The prediction is linear in $p$:

$$\hat{y}(p, q) = \frac{1}{n}b(q)^T p.$$

We now expand $b(q)$ using the Kronecker product ($\otimes$) and commutation matrix ($K$):

1. $s(q) = \text{vec}(S(Q)) = \text{vec}(Z^T QZ) = (Z^T \otimes Z^T)q$. Let $J_S = (Z^T \otimes Z^T)$.
2. $a(q) = \text{vec}(A(Q)) = \phi(s(q)) = \phi(J_S q)$.
3. $b(q) = \text{vec}(EA(Q)^T V^T) = (V \otimes E)\text{vec}(A(Q)^T)$.
4. $\text{vec}(A(Q)^T) = K_{m,m}\text{vec}(A(Q)) = K_{m,m}a(q)$.

Combining these, we define the constant matrix $J_A$:

$$b(q) = (V \otimes E)K_{m,m}a(q) = \underbrace{(V \otimes E)K_{m,m}}_{J_A}\phi(J_S q).$$

Our final vectorized prediction is:

$$\hat{y}(p, q) = \frac{1}{n}(J_A\phi(J_S q))^T p.$$

We define the loss $f$ as either the Mean Squared Error (MSE) or Cross-Entropy (CE). The gradient and Hessian of the loss are given by the chain rule:

$$\nabla f = r\nabla\hat{y} \quad \text{and} \quad H_f = \gamma_1(\nabla\hat{y})(\nabla\hat{y})^T + \gamma_2 r H_{\hat{y}},$$

where $r$ is the residual and $\gamma_1, \gamma_2$ are scalar coefficients depending on the loss.

- **MSE:** $f = (\hat{y} - y)^2$. Then $r = 2(\hat{y} - y)$, $\gamma_1 = 2$, $\gamma_2 = 1$. (Or equivalently absorbing the 2 into $r$, giving coefficients 2 and 2).

- **CE:** $f = \text{BCE}(\sigma(\hat{y}), y)$. Then $r = \sigma(\hat{y}) - y$, $\gamma_1 = \sigma(\hat{y})(1 - \sigma(\hat{y})) \le 1/4$, $\gamma_2 = 1$.

In both cases, the coefficients are upper bounded by 2. Thus, we perform the calculation using the upper bound coefficients (2, 2) which covers both cases:

$$(\nabla\hat{y})(\nabla\hat{y})^T + 2r H_{\hat{y}}.$$

Finally, we define the diagonal matrices of derivatives:

$$D'(q) = \text{diag}(\text{vec}(\phi'(J_S q))) \quad \text{and} \quad D''(q) = \text{diag}(\text{vec}(\phi''(J_S q))).$$

We first compute $\nabla\hat{y}$.

$$\nabla_p\hat{y} = \frac{\partial\hat{y}}{\partial p} = \frac{\partial}{\partial p}\left(\frac{1}{n}b(q)^T p\right) = \frac{1}{n}b(q) = \frac{1}{n}J_A\phi(J_S q).$$

We also have

$$\nabla_q\hat{y} = \frac{\partial\hat{y}}{\partial q} = \frac{\partial}{\partial q}\left(\frac{1}{n}p^T b(q)\right) = \frac{1}{n}\left(\frac{\partial b(q)}{\partial q}\right)^T p.$$

We need the Jacobian of $b(q)$, $\frac{\partial b(q)}{\partial q^T}$.

$$\frac{\partial b(q)}{\partial q^T} = \frac{\partial(J_A\phi(J_S q))}{\partial q^T} = J_A\frac{\partial(\phi(J_S q))}{\partial q^T} = J_A D'(q)\frac{\partial(J_S q)}{\partial q^T} = J_A D'(q)J_S$$

Plugging this back in (and transposing):

$$\nabla_q\hat{y} = \frac{1}{n}(J_A D'(q)J_S)^T p = \frac{1}{n}J_S^T D'(q)^T J_A^T p = \frac{1}{n}J_S^T D'(q)J_A^T p.$$

We will compute the four blocks of $H_f$ by first finding the four blocks of $H_{\hat{y}}$.

$H_{\hat{y},pp}$**:** We differentiate $\nabla_p\hat{y}$ w.r.t $p^T$:

$$H_{\hat{y},pp} = \frac{\partial}{\partial p^T}(\nabla_p\hat{y}) = \frac{\partial}{\partial p^T}\left(\frac{1}{n}b(q)\right) = \mathbf{0}.$$

since $b(q)$ does not depend on $p$.

$H_{\hat{y},pq}$: We differentiate $\nabla_p \hat{y}$ w.r.t $q^T$:

$$H_{\hat{y},pq} = \frac{\partial}{\partial q^T}(\nabla_p \hat{y}) = \frac{\partial}{\partial q^T}\left(\frac{1}{n}b(q)\right) = \frac{1}{n}\frac{\partial b(q)}{\partial q^T}.$$

We have already computed $\frac{\partial b(q)}{\partial q^T}$:

$$H_{\hat{y},pq} = \frac{1}{n}J_A D'(q)J_S.$$

$H_{\hat{y},qp}$: This block is the transpose of the previous one:

$$H_{\hat{y},qp} = H_{\hat{y},pq}^T = \left(\frac{1}{n}J_A D'(q)J_S\right)^T = \frac{1}{n}J_S^T D'(q)J_A^T.$$

$H_{\hat{y},qq}$: We differentiate $\nabla_q \hat{y}$ w.r.t $q^T$:

$$H_{\hat{y},qq} = \frac{\partial}{\partial q^T}(\nabla_q \hat{y}) = \frac{\partial}{\partial q^T}\left(\frac{1}{n}J_S^T D'(q)J_A^T p\right).$$

Let $v = J_A^T p$ be a constant vector. We need to find $\frac{1}{n}J_S^T\left(\frac{\partial(D'(q)v)}{\partial q^T}\right)$. Using the rule for differentiating a diagonal matrix $\frac{\partial(\mathrm{diag}(X)v)}{\partial y^T} = \mathrm{diag}(v)\frac{\partial X}{\partial y^T}$, we have:

$$\frac{\partial(D'(q)v)}{\partial q^T} = \frac{\partial(\mathrm{diag}(\mathrm{vec}(\phi'(J_S q)))v)}{\partial q^T} = \mathrm{diag}(v)\frac{\partial(\mathrm{vec}(\phi'(J_S q)))}{\partial q^T}.$$

The derivative of the vector $\mathrm{vec}(\phi'(J_S q)$ is $D''(q)J_S$.

$$\frac{\partial(D'(q)v)}{\partial q^T} = \mathrm{diag}(v)D''(q)J_S = \mathrm{diag}(J_A^T p)D''(q)J_S.$$

Plugging this back into the expression for $H_{\hat{y},qq}$:

$$H_{\hat{y},qq} = \frac{1}{n}J_S^T \mathrm{diag}(J_A^T p)D''(q)J_S.$$

We now put together the four blocks of $H_f \approx 2(\nabla \hat{y})(\nabla \hat{y})^T + 2r H_{\hat{y}}$.

$$H_{pp} = 2(\nabla_p \hat{y})(\nabla_p \hat{y})^T + 2r H_{\hat{y},pp} = 2\left(\frac{1}{n}b(q)\right)\left(\frac{1}{n}b(q)\right)^T + \mathbf{0}.$$

$$H_{pp} = \frac{2}{n^2}(J_A\phi(J_S q))(J_A\phi(J_S q))^T$$

$$H_{qq} = 2(\nabla_q \hat{y})(\nabla_q \hat{y})^T + 2r H_{\hat{y},qq}$$

$$H_{qq} = \frac{2}{n^2}\left(J_S^T D'(q)J_A^T p\right)\left(J_S^T D'(q)J_A^T p\right)^T + \frac{2r}{n}\left[J_S^T \mathrm{diag}(J_A^T p)D''(q)J_S\right]$$

$$H_{pq} = 2(\nabla_p \hat{y})(\nabla_q \hat{y})^T + 2r H_{\hat{y},pq}$$

$$H_{pq} = 2\left(\frac{1}{n}J_A\phi(J_S q)\right)\left(\frac{1}{n}J_S^T D'(q)J_A^T p\right)^T + 2r\left(\frac{1}{n}J_A D'(q)J_S\right)$$

$$H_{pq} = \frac{2}{n^2}(J_A\phi(J_S q))(p^T J_A D'(q)J_S) + \frac{2r}{n}(J_A D'(q)J_S)$$

$$H_{qp} = 2(\nabla_q \hat{y})(\nabla_p \hat{y})^T + 2r H_{\hat{y},qp} = H_{pq}^T$$

$$H_{qp} = \frac{2}{n^2}(J_S^T D'(q) J_A^T p)(J_A \phi(J_S q))^T + \frac{2r}{n}(J_S^T D'(q) J_A^T).$$

Now we pass in bounding the regulatized loss. We use the notation $f$ for the regulatized and $\bar{f}$ unregularized loss. The Hessian of the regularized loss $f$ is

$$\nabla^2 f(p,q) = \nabla^2 \bar{f}(p,q) + \begin{bmatrix} \lambda_P I & 0 \\ 0 & \lambda_Q I \end{bmatrix}.$$

Using the triangle inequality for the spectral norm ($\|\cdot\|_2$):

$$\|\nabla^2 f(p,q)\|_2 \le \|\nabla^2 \bar{f}(p,q)\|_2 + \left\| \begin{bmatrix} \lambda_P I & 0 \\ 0 & \lambda_Q I \end{bmatrix} \right\|_2 \le \|\nabla^2 \bar{f}(p,q)\|_2 + (\lambda_P + \lambda_Q).$$

We proceed by computing an upper bound for $\|\nabla^2 \bar{f}(p,q)\|_2$.

From our previous calculations, we have the four blocks:

$$
\begin{aligned}
H_{pp} &= \frac{2}{n^2}(J_A \phi(J_S q))(J_A \phi(J_S q))^T \\
H_{qq} &= \frac{2}{n^2}(J_S^T D'(q) J_A^T p)(J_S^T D'(q) J_A^T p)^T + \frac{2r}{n}\left[ J_S^T \operatorname{diag}(J_A^T p) D''(q) J_S \right] \\
H_{pq} &= \frac{2}{n^2}(J_A \phi(J_S q))(p^T J_A D'(q) J_S) + \frac{2r}{n}(J_A D'(q) J_S) \\
H_{qp} &= H_{pq}^T
\end{aligned}
$$

We now upper bound the spectral norm of the full Hessian by the sum of the norms of its blocks:

$$\|\nabla^2 \bar{f}(p,q)\|_2 \le \|H_{pp}\|_2 + 2\|H_{pq}\|_2 + \|H_{qq}\|_2.$$

Since $\|Z\|_2 \le C_Z$, it holds $\|J_S\|_2 \le C_Z^2$ and $\|J_A\|_2 \le C_Z$.

**Bounding $H_{pp}$:** $H_{pp}$ is a rank-1 matrix.

$$\|H_{pp}\|_2 = \frac{2}{n^2}\|J_A \phi(J_S q)\|_2^2 \le \frac{2}{n^2}\|J_A\|_2^2 \|\phi(J_S q)\|_2^2 \le \frac{2}{n^2}C_Z^6 C_1^2 \|q\|_2^2.$$

**Bounding $H_{pq}$:** We use the triangle inequality.

$$
\begin{aligned}
\|H_{pq}\|_2 &\le \left\| \frac{2}{n^2}(J_A \phi(J_S q))(p^T J_A D'(q) J_S) \right\|_2 + \left\| \frac{2r}{n}(J_A D'(q) J_S) \right\|_2 \\
&\le \frac{2}{n^2}\|J_A \phi(J_S q)\|_2 \cdot \|p^T J_A D'(q) J_S\|_2 + \frac{2|r|}{n}\|J_A D'(q) J_S\|_2 \\
&\le \frac{2}{n^2}(C_Z^3 C_1 \|q\|_2) \cdot (\|p\|_2 \|J_A\|_2 \|D'(q)\|_2 \|J_S\|_2) + \frac{2|r|}{n}(\|J_A\|_2 \|D'(q)\|_2 \|J_S\|_2) \\
&\le \frac{2}{n^2}(C_Z^3 C_1 \|q\|_2) \cdot (\|p\|_2 C_2 C_Z^3) + \frac{2|r|}{n}C_2 C_Z^3 \\
&= \frac{2}{n^2}C_Z^6 C_1 C_2 \|p\|_2 \|q\|_2 + \frac{2}{n}C_2 C_Z^3 |r|.
\end{aligned}
$$

**Bounding $H_{qq}$:**   We use the triangle inequality.

$$\|H_{qq}\|_2 \leq \frac{2}{n^2}\|J_S^T D'(q)J_A^T p\|_2^2 + \frac{2|r|}{n}\|J_S^T \mathrm{diag}(J_A^T p)D''(q)J_S\|_2$$

$$\leq \frac{2}{n^2}(\|J_S\|_2\|D'(q)\|_2\|J_A\|_2\|p\|_2)^2 + \frac{2|r|}{n}(\|J_S\|_2^2\|\mathrm{diag}(J_A^T p)\|_2\|D''(q)\|_2)$$

$$\leq \frac{2}{n^2}(C_2 C_Z^3\|p\|_2)^2 + \frac{2|r|}{n}((C_Z^2)^2 \cdot (\|J_A\|_2\|p\|_2) \cdot C_3)$$

$$= \frac{2}{n^2}C_2^2 C_Z^6\|p\|_2^2 + \frac{2}{n}C_Z^5 C_3|r|\|p\|_2.$$

**Overall Bound for $\bar{f}$:**

$$\|\nabla^2 \bar{f}\|_2 \leq \|H_{pp}\|_2 + 2\|H_{pq}\|_2 + \|H_{qq}\|_2$$

$$\leq \underbrace{\frac{2}{n^2}C_Z^6 C_1^2\|q\|_2^2}_{K_1} + \underbrace{\frac{2}{n^2}C_Z^6 C_1 C_2\|p\|_2\|q\|_2}_{K_2} + \underbrace{\frac{2}{n}C_2 C_Z^3|r|}_{K_3} + \underbrace{\frac{2}{n^2}C_2^2 C_Z^6\|p\|_2^2}_{K_4} + \underbrace{\frac{2}{n}C_Z^5 C_3|r|\|p\|_2}_{K_5}. \tag{32}$$

We bound the residual magnitude $|r|$ by the loss. For MSE, $|r| = \sqrt{\bar{f}}$. For Cross-Entropy, $|r| \leq \sqrt{2\bar{f}}$. To unify notation, let $C_r = 1$ for MSE and $C_r = \sqrt{2}$ for CE. The Hessian spectral norm satisfies:

$$\|\nabla^2 f(p,q)\|_2 \leq (\lambda_P + \lambda_Q) + T_{\mathrm{quad}}(p,q) + T_{\mathrm{mixed}}(p,q) + T_{\mathrm{res}}(p,q), \tag{33}$$

where:

$$T_{\mathrm{quad}} := K_1\|q\|_2^2 + K_2\|p\|_2\|q\|_2 + K_4\|p\|_2^2,$$

$$T_{\mathrm{mixed}} := K_5 C_r\|p\|_2\sqrt{\bar{f}},$$

$$T_{\mathrm{res}} := K_3 C_r\sqrt{\bar{f}}.$$

The local condition for (i) is equivalent to $T_{\mathrm{quad}} \leq T_{\mathrm{mixed}} + T_{\mathrm{res}}$. Thus,

$$\|\nabla^2 f(p,q)\|_2 \leq (\lambda_P + \lambda_Q) + 2(T_{\mathrm{mixed}}(p,q) + T_{\mathrm{res}}(p,q)).$$

Since the terms involving $\|q\|_2$ are non-negative ($K_1, K_2 \geq 0$), we can drop them to obtain a necessary condition for $\|p\|_2$:

$$K_4\|p\|_2^2 \leq K_5 C_r\sqrt{\bar{f}}\|p\|_2 + K_3 C_r\sqrt{\bar{f}}.$$

Rewriting this as $K_4\|p\|_2^2 - (K_5 C_r\sqrt{\bar{f}})\|p\|_2 - (K_3 C_r\sqrt{\bar{f}}) \leq 0$, we solve for the positive root to bound $\|p\|_2$:

$$\|p\|_2 \leq \frac{K_5 C_r}{K_4}\sqrt{\bar{f}} + \sqrt{\frac{K_3 C_r}{K_4}}\bar{f}^{1/4}.$$

We substitute this bound directly into the mixed term $T_{\mathrm{mixed}} = K_5 C_r\|p\|_2\sqrt{\bar{f}}$:

$$T_{\mathrm{mixed}} \leq K_5 C_r\left(\frac{K_5 C_r}{K_4}\bar{f} + \sqrt{\frac{K_3 C_r}{K_4}}\bar{f}^{3/4}\right) = \frac{(K_5 C_r)^2}{K_4}\bar{f} + C_{\mathrm{cross}}\bar{f}^{3/4}, \quad \text{where } C_{\mathrm{cross}} = K_5 C_r\sqrt{\frac{K_3 C_r}{K_4}}.$$

Using $\bar{f}(W) \leq f(W)$, we replace all instances of $\bar{f}$ with $f$. To obtain the affine bound $H_0^{\mathrm{warm\text{-}up}} + H_1^{\mathrm{warm\text{-}up}}(f(W) - f^*)$, we apply the following linearizations valid for $f \geq 0$ and $\beta > 0$:

1. **Linear term:** $f = (f - f^*) + f^*$.

2. **Cross term ($f^{3/4}$):** By Young's inequality ($ab \leq a^p/p + b^q/q$ with $p = 4/3, q = 4$), we have $u^{3/4} \cdot 1 \leq \frac{3}{4}u + \frac{1}{4}$. Thus:

$$f^{3/4} \leq \frac{3}{4}f + \frac{1}{4} = \frac{3}{4}(f - f^*) + \left(\frac{3}{4}f^* + \frac{1}{4}\right).$$

3. **Residual term ($\sqrt{f}$):** Using the robust smoothness inequality $\sqrt{u} \leq \frac{u}{2\beta} + \frac{\beta}{2}$:

$$\sqrt{f} \leq \frac{f}{2\beta} + \frac{\beta}{2} = \frac{f - f^*}{2\beta} + \left(\frac{f^*}{2\beta} + \frac{\beta}{2}\right).$$

Substituting these into the general curvature bound results to

$$\|\nabla^2 f\|_2 \leq (\lambda_P + \lambda_Q) + 2\frac{(K_5 C_r)^2}{K_4}\left((f - f^*) + f^*\right) + 2C_{\text{cross}}\left(\frac{3}{4}(f - f^*) + \left(\frac{3}{4}f^* + \frac{1}{4}\right)\right) + 2K_3 C_r\left(\frac{f - f^*}{2\beta} + \left(\frac{f^*}{2\beta} + \frac{\beta}{2}\right)\right).$$

We now collect the coefficients of $(f(W) - f^*)$ into $H_0^{\text{warm-up}}$ and the constant terms into $H_1^{\text{warm-up}}$ and set $\beta = 1$:

$$H_0^{\text{warm-up}} = (\lambda_P + \lambda_Q) + 2\left(\frac{(K_5 C_r)^2}{K_4}f^* + C_{\text{cross}}\left(\frac{3}{4}f^* + \frac{1}{4}\right) + K_3 C_r\left(\frac{f^*}{2} + \frac{1}{2}\right)\right)$$

$$H_1^{\text{warm-up}} = 2\left(\frac{(K_5 C_r)^2}{K_4} + \frac{3}{4}C_{\text{cross}} + \frac{K_3 C_r}{2}\right). \tag{34}$$

For the global bound (ii), we begin again with the exact decomposition of the Hessian bound:

$$\|\nabla^2 f(W)\|_2 \leq (\lambda_P + \lambda_Q) + T_{\text{quad}}(W) + T_{\text{mixed}}(W) + T_{\text{res}}(W).$$

Recall the definitions:

$$T_{\text{quad}} = K_1\|q\|_2^2 + K_2\|p\|_2\|q\|_2 + K_4\|p\|_2^2,$$

$$T_{\text{mixed}} = K_5 C_r \|p\|_2 \sqrt{\bar{f}},$$

$$T_{\text{res}} = K_3 C_r \sqrt{\bar{f}}.$$

We first apply the AM-GM inequality $ab \leq \frac{1}{2}(a^2 + b^2)$ to separate the coupled terms.

For $T_{\text{quad}}$:

$$T_{\text{quad}} = K_1\|q\|_2^2 + K_2(\|p\|_2\|q\|_2) + K_4\|p\|_2^2$$

$$\leq K_1\|q\|_2^2 + \frac{K_2}{2}(\|p\|_2^2 + \|q\|_2^2) + K_4\|p\|_2^2$$

$$= \left(K_4 + \frac{K_2}{2}\right)\|p\|_2^2 + \left(K_1 + \frac{K_2}{2}\right)\|q\|_2^2.$$

For $T_{\text{mixed}}$:

$$T_{\text{mixed}} = K_5 C_r(\|p\|_2\sqrt{\bar{f}}) \leq K_5 C_r\left(\frac{1}{2}\|p\|_2^2 + \frac{1}{2}\bar{f}\right) = \frac{K_5 C_r}{2}\|p\|_2^2 + \frac{K_5 C_r}{2}\bar{f}.$$

We now use the regularization bounds derived from the total loss definition $f(W) \geq \frac{\lambda_P}{2}\|p\|_2^2 + \frac{\lambda_Q}{2}\|q\|_2^2$:

$$\|p\|_2^2 \leq \frac{2f(W)}{\lambda_P}, \qquad \|q\|_2^2 \leq \frac{2f(W)}{\lambda_Q}.$$

Substituting these into the disentangled terms (and using $\bar{f} \leq f$):

For $T_{\text{quad}}$:

$$T_{\text{quad}} \leq \left( \frac{2}{\lambda_P} \left( K_4 + \frac{K_2}{2} \right) + \frac{2}{\lambda_Q} \left( K_1 + \frac{K_2}{2} \right) \right) f(W).$$

For $T_{\text{mixed}}$:

$$T_{\text{mixed}} \leq \left( \frac{2}{\lambda_P} \left( \frac{K_5 C_r}{2} \right) \right) f(W) + \frac{K_5 C_r}{2} f(W) = \left( \frac{K_5 C_r}{\lambda_P} + \frac{K_5 C_r}{2} \right) f(W).$$

Let us define a consolidated coefficient $C_{\text{linear}}$ for all terms that are strictly proportional to $f(W)$:

$$C_{\text{linear}} := \underbrace{\frac{2K_4 + K_2 + K_5 C_r}{\lambda_P}}_{\text{Terms from } \|p\|^2} + \underbrace{\frac{2K_1 + K_2}{\lambda_Q}}_{\text{Terms from } \|q\|^2} + \underbrace{\frac{K_5 C_r}{2}}_{\text{From } \bar{f}}.$$

For $T_{\text{res}} = K_3 C_r \sqrt{\bar{f}}$, we use $\sqrt{\bar{f}} \leq \sqrt{f}$ and the linearization $\sqrt{f} \leq \frac{f - f^*}{2\beta} + \left( \frac{f^*}{2\beta} + \frac{\beta}{2} \right)$:

$$T_{\text{res}} \leq K_3 C_r \left[ \frac{f - f^*}{2\beta} + \left( \frac{f^*}{2\beta} + \frac{\beta}{2} \right) \right].$$

Combining everything into the form $H_0 + H_1 (f - f^*)$:

$$\|\nabla^2 f\|_2 \leq (\lambda_P + \lambda_Q) + C_{\text{linear}} f(W) + T_{\text{res}}.$$

We rewrite $C_{\text{linear}} f(W)$ as $C_{\text{linear}} (f - f^*) + C_{\text{linear}} f^*$.

Setting $\beta = 1$ and $C_r = \sqrt{2}$, we arrive at the following expressions for the constants $H_0$ and $H_1$:

$$H_0 = (\lambda_P + \lambda_Q) + \sqrt{2} K_3 \left( \frac{f^*}{2} + \frac{1}{2} \right) + f^* \left( \frac{2K_4 + K_2 + \sqrt{2} K_5}{\lambda_P} + \frac{2K_1 + K_2}{\lambda_Q} + \frac{\sqrt{2} K_5}{2} \right)$$

$$H_1 = \frac{\sqrt{2} K_3}{2} + C_{\text{linear}} = \frac{\sqrt{2} K_3}{2} + \left( \frac{2K_4 + K_2 + \sqrt{2} K_5}{\lambda_P} + \frac{2K_1 + K_2}{\lambda_Q} + \frac{\sqrt{2} K_5}{2} \right), \tag{35}$$

where

$$K_1 = \frac{2}{n} C_Z^6 C_1^2$$

$$K_2 = \frac{2}{n^2} C_Z^6 C_1 C_2$$

$$K_3 = \frac{2}{n} C_2 C_Z^3$$

$$K_4 = \frac{2}{n^2} C_2^2 C_Z^6$$

$$K_5 = \frac{2}{n} C_Z^5 C_3.$$

$\square$

We close the discussion of this section with our main result, i.e. proving an $(H_0, H_1)$-smoothness condition for the *in-context* loss function over a distribution of tasks. Before we get to its proof, we present some useful lemmas from the theory of random matrices.

**Lemma 1.** *A sub-Gaussian random variable is also sub-exponential.*

**Lemma 2.** *For a random matrix $Z$ whose entries are independent and sub-exponential (with zero mean), its spectral norm $\|Z\|_2$ exhibits sub-exponential tails. This means the probability of the norm being large decays very quickly.*

$$\mathbb{P}(\|Z\|_2 > t) \leq C \exp(-ct) \quad \text{for large } t,$$

*for constants $C, c$.*

**Lemma 3** (Finite Moments)**.** *A random variable $X$ with sub-exponential tails (like $\|Z_j\|_2$) has finite moments:*

$$\mathbb{E}[|X|^k] < \infty \quad \text{for all } k \geq 1.$$

For a rigorous treatment of these topics, see (Vershynin, 2026).

Now, we are ready for the main proof. This is the formal version of Proposition E.1.

**Proposition E.2.** *Consider the transformer model described in Section 3.1.2, for in-context learning in continuous variables with MSE loss, or in binary variables with CE loss. $\phi$ is assumed to satisfy the conditions of Proposition 3.3 and $f_j(P, Q)$ is the L2 regularized loss corresponding to the j-th prompt. Consider the regularized in-context loss function*

$$f(P, Q) = \mathbb{E}_j[f_j(P, Q)], \text{ where } f_j(P, Q) = (\hat{y}_j - y_{j,true})^2 + \frac{\lambda_P}{2}\|P\|_F^2 + \frac{\lambda_Q}{2}\|Q\|_F^2.$$

*Assume that $D_x$ is sub-gaussian and the distribution of $y = h(x)$ is sub-exponential. Assume also that the variance $V_f(\theta) := Var_j(f_j(\theta)$ is finite and upper bounded by a constant $V$. Then, it holds*

*(i) If*

$$K_{1,j}\|Q\|_F^2 + K_{2,j}\|P\|_F\|Q\|_F + K_{4,j}\|P\|_F^2 \leq \sqrt{2}K_{5,j}\|P\|_F\sqrt{f(P,Q)} + \sqrt{2}K_{3,j}\sqrt{f(P,Q)},$$

*for all $j$, where $K_{1,j}, K_{2,j}, K_{3,j}, K_{4,j}, K_{5,j}$ are the analogues of the constants $K_1, K_2, K_3, K_4, K_5$ from Eq. (32) substituting $Z$ with the task-dependent data matrix $Z_j$, then*

$$\|\nabla^2 f(P,Q)\|_2 \leq H_0^{warm\text{-}up} + H_1^{warm\text{-}up}(f(P,Q) - f^*)),$$

*with constants $H_0^{warm\text{-}up}$ and $H_1^{warm\text{-}up}$ defined as in equation (37).*

*(ii) For any $(P, Q)$, we have*

$$\left\|\nabla^2 f(P,Q)\right\|_2 \leq H_0 + H_1(f(P,Q) - f^*).$$

*The constants $H_0$ and $H_1$ are defined as in equation (38) and explicitly incorporate the loss variance $\sqrt{V_f(\theta)}$.*

*Proof.* For simplicity, we denote $\theta \equiv (P, Q)$.

We start with Jensen's Inequality for the spectral norm:

$$\left\|\nabla^2 f(\theta)\right\|_2 = \left\|\mathbb{E}_j[\nabla^2 f_j(\theta)]\right\|_2 \leq \mathbb{E}_j\left[\left\|\nabla^2 f_j(\theta)\right\|_2\right].$$

If it holds $K_{1,j}\|q\|_2^2 + K_{2,j}\|p\|_2\|q\|_2 + K_{4,j}\|p\|_2^2 \leq K_{5,j}C_r\|p\|_2\sqrt{f} + K_{3,j}C_r\sqrt{f}$ for all tasks $j$, the single-task bound from Proposition E.1 applies pointwise:

$$\left\|\nabla^2 f_j(\theta)\right\|_2 \leq C_{0,j}^{\text{warm-up}} + C_{1,j}^{\text{warm-up}}\sqrt{f_j(\theta)}. \tag{36}$$

for some $C_{0,j}^{\text{warm-up}}$ and $C_{1,j}^{\text{warm-up}}$ random variables depending on the data of task $j$. Taking the expectation:

$$\mathbb{E}_j\left[\left\|\nabla^2 f_j(\theta)\right\|_2\right] \leq \mathbb{E}_j[C_{0,j}^{\text{warm-up}}] + \mathbb{E}_j\left[C_{1,j}^{\text{warm-up}}\sqrt{f_j(\theta)}\right].$$

We define the mean constant $\bar{C}_0 = \mathbb{E}_j[C_{0,j}^{\text{warm-up}}]$. For the second term, we apply the Cauchy-Schwarz inequality for expectations:

$$\mathbb{E}_j\left[C_{1,j}^{\text{warm-up}}\sqrt{f_j(\theta)}\right] \leq \sqrt{\mathbb{E}_j[(C_{1,j}^{\text{warm-up}})^2]} \cdot \sqrt{\mathbb{E}_j[(\sqrt{f_j(\theta)})^2]}.$$

Note that $\mathbb{E}_j[(\sqrt{f_j})^2] = \mathbb{E}_j[f_j] = f(\theta)$. Let $\bar{C}_1^{\text{rms}} = \sqrt{\mathbb{E}_j[(C_{1,j}^{\text{warm-up}})^2]}$. These moments are finite because $C_{1,j}^{\text{warm-up}}$ is polynomial in $\|Z_j\|_2$, which has sub-exponential tails. Thus, we have:

$$\left\| \nabla^2 f(\theta) \right\|_2 \leq \bar{C}_0 + \bar{C}_1^{\text{rms}} \sqrt{f(\theta)}.$$

To obtain the affine form, we use the linearization inequality $\sqrt{u} \leq \frac{u - u^*}{2\beta} + \left( \frac{u^*}{2\beta} + \frac{\beta}{2} \right)$ with $u = f(\theta)$:

$$\left\| \nabla^2 f(\theta) \right\|_2 \leq \bar{C}_0 + \bar{C}_1^{\text{rms}} \left[ \frac{f(\theta) - f^*}{2\beta} + \frac{f^*}{2\beta} + \frac{\beta}{2} \right].$$

Collecting terms and setting $\beta = 1$ yields the final constants:

$$H_0^{\text{warm-up}} = \bar{C}_0 + \bar{C}_1^{\text{rms}} \left( \frac{f^*}{2} + \frac{1}{2} \right), \quad H_1^{\text{warm-up}} = \frac{\bar{C}_1^{\text{rms}}}{2}. \tag{37}$$

For the global bound, the single-task result holds unconditionally:

$$\left\| \nabla^2 f_j(\theta) \right\|_2 \leq H_{0,j} + H_{1,j} f_j(\theta).$$

Taking the expectation, we get

$$\mathbb{E}_j \left[ \left\| \nabla^2 f_j(\theta) \right\|_2 \right] \leq \mathbb{E}_j[H_{0,j}] + \mathbb{E}_j \left[ H_{1,j} f_j(\theta) \right].$$

Let $\bar{H}_0 = \mathbb{E}_j[H_{0,j}]$. For the second term, we apply Cauchy-Schwarz:

$$\mathbb{E}_j \left[ H_{1,j} f_j(\theta) \right] \leq \sqrt{\mathbb{E}_j[(H_{1,j})^2]} \cdot \sqrt{\mathbb{E}_j[f_j(\theta)^2]}.$$

Using the definition of variance $\mathbb{E}[X^2] = (\mathbb{E}[X])^2 + \text{Var}(X)$, we have $\mathbb{E}[f_j^2] = f(\theta)^2 + V_f(\theta)$. Let $\bar{H}_1^{\text{rms}} = \sqrt{\mathbb{E}_j[(H_{1,j})^2]}$. Using $\sqrt{a + b} \leq \sqrt{a} + \sqrt{b}$, we have

$$\sqrt{f(\theta)^2 + V_f(\theta)} \leq f(\theta) + \sqrt{V_f(\theta)}.$$

Thus:

$$\left\| \nabla^2 f(\theta) \right\|_2 \leq \bar{H}_0 + \bar{H}_1^{\text{rms}} \left( f(\theta) + \sqrt{V_f(\theta)} \right).$$

We write $f(\theta) = (f(\theta) - f^*) + f^*$. Absorbing the variance term into the constant and upper bounding it by $V$:

$$H_0 = \bar{H}_0 + \bar{H}_1^{\text{rms}} \left( f^* + \sqrt{V} \right), \quad H_1 = \bar{H}_1^{\text{rms}}. \tag{38}$$

The expressions $\bar{H}_0$ and $\bar{H}_1^{\text{rms}}$ are finite by Lemma 3. $\qquad\square$

## F. Neural Networks are in general not $(L_0, L_1)$-smooth

In this section, we demonstrate that neural networks still violate the $(L_0, L_1)$-smoothness, even in the presence of L2 regularization or weight balancedness. We start with an example of a simple 2-layer neural network with L2 regularization when $(L_0, L_1)$-smoothness is violated for $L_0, L_1 \geq 0$.

**Proposition F.1.** *We consider a simple 2-layer neural network with MSE loss*

$$f(u, v) = \frac{1}{2}(u\sigma(v))^2 + \frac{\lambda_1}{2} u^2 + \frac{\lambda_2}{2} v^2,$$

*such that $\sigma(0) = 0, \sigma'(0) \neq 0$[3]. Then $(L_0, L_1)$-smoothness does not hold for any $L_0, L_1 \geq 0$.*

---

[3]These assumptions are satisfied for several activation functions such as `tanh`, `GELU`, `SiLU`.

*Proof.* For this example, the gradient and the Hessian are

$$\nabla f(u,v) = \begin{bmatrix} u\sigma^2(v) + \lambda_1 u \\ u^2\sigma(v)\sigma'(v) + \lambda_2 v \end{bmatrix}, \ \nabla^2 f(u,v) = \begin{bmatrix} \sigma^2(v) + \lambda_1 & 2u\sigma(v)\sigma'(v) \\ 2u\sigma(v)\sigma'(v) & u^2((\sigma'(v))^2 + \sigma(v)\sigma''(v)) + \lambda_2. \end{bmatrix}$$

Let us evaluate them at the point $(u,0)$. Note that $\sigma(0) = 0, \sigma'(0) \neq 0$ by the assumption of the proposition. We obtain

$$\nabla f(u,v) = \begin{bmatrix} \lambda_1 u \\ 0 \end{bmatrix}, \ \nabla^2 f(u,v) = \begin{bmatrix} \lambda_1 & 0 \\ 0 & u^2(\sigma'(0))^2 + \lambda_2. \end{bmatrix}$$

Therefore, we obtain that

$$\|\nabla^2 f(u,0)\|_2 = \max\{\lambda_1, u^2(\sigma'(0))^2 + \lambda_2\}, \quad \|\nabla f(u,0)\| = \lambda_1|u|.$$

Thus, if $(L_0, L_1)$-smoothness was true for this function, then there were constants $L_0, L_1 \geq 0$ such that

$$\|\nabla^2 f(u,0)\|_2 = \max\{\lambda_1, u^2(\sigma'(0))^2 + \lambda_2\} \leq L_0 + L_1\lambda_1|u|. \tag{39}$$

Let $u \geq \frac{\sqrt{\lambda_1}}{|\sigma'(0)|}$. Then $\|\nabla^2 f(u,0)\|_2 = u^2(\sigma'(0))^2 + \lambda_2$. Therefore, dividing both sides of (39) by $u$ we obtain

$$u(\sigma'(0))^2 \leq \frac{L_0}{u} + L_1\lambda_1.$$

Taking $u \to +\infty$, we get that LHS goes to $+\infty$ while RHS goes to a constant. Therefore, $(L_0, L_1)$-smoothness is violated for any $L_0, L_1 \geq 0$. □

Next, we demonstrate that $(L_0, L_1)$-smoothness is violated under a balancedness condition as well.

**Proposition F.2.** *We consider a* 2*-layer neural network with MSE loss*

$$f(W_1, W_2) = \|Y - W_1\phi(W_2 X)\|_F^2$$

*and leaky-ReLU or linear activation function, i.e. $\phi(x) = \max\{x, bx\}$, with $0 < b \leq 1$. Then, $(L_0, L_1)$-smoothness does not hold under weak balancedness for any $L_0, L_1 \geq 0$.*

*Proof.* We start by computing the gradients and Hessian of $f$.

For all computations, we work with a vectorized version of $f$:

$$f(W_1, W_2) = \|\text{vec}(Y) - \text{vec}(W_1\phi(W_2 X))\|_F^2.$$

Let us denote

$$R := \text{vec}(Y) - \text{vec}(W_1\phi(W_2 X)) = \text{vec}(Y) - (\phi(W_2 X)^\top \otimes I_c)\text{vec}(W_1).$$

For the second inequality, we used a classic property between vectorization and the Kronecker product.

The derivative with respect to $\text{vec}(W_1)$ is

$$\frac{\partial f}{\partial \text{vec}(W_1)} = \frac{\partial f}{\partial R} \cdot \frac{\partial R}{\partial \text{vec}(W_1)} = -2R^\top(\phi(W_2 X)^\top \otimes I_c) = -2R^\top(\phi(W_2 X)^\top \otimes I_c).$$

Transposing in order to bring the vector in column form, we get

$$\frac{\partial f}{\partial \text{vec}(W_1)} = 2(\phi(W_2 X) \otimes I_c)R = -2\text{vec}((Y - W_1\phi(W_2 X))\phi(W_2 X)^\top). \tag{40}$$

The gradient with respect to $W_2$ is similarly

$$\frac{\partial f}{\partial \text{vec}(W_2)} = \frac{\partial f}{\partial R} \cdot \frac{\partial R}{\partial \text{vec}(W_2)}.$$

$\frac{\partial f}{\partial R}$ is again $2R^\top$. In order to deal with $\frac{\partial R}{\partial \text{vec}(W_2)}$, we write

$$R = \text{vec}(Y) - (I_m \otimes W_1)\text{vec}(\phi(W_2 X)).$$

Thus,

$$\frac{\partial R}{\partial \text{vec}(W_2)} = -(I_m \otimes W_1)\frac{\partial \text{vec}(\phi(W_2 X))}{\partial \text{vec}(W_2)} = -(I_m \otimes W_1)\frac{\partial \text{vec}(\phi(W_2 X))}{\partial \text{vec}(W_2 X)}\frac{\partial \text{vec}(W_2 X)}{\partial \text{vec}(W_2)}$$

$\frac{\partial \text{vec}(\phi(W_2 X))}{\partial \text{vec}(W_2)}$ is the diagonal matrix $\text{diag}(\text{vec}(\phi'(W_2 X))$.

Since $\text{vec}(W_2 X) = (X^\top \otimes I_{n_1})\text{vec}(W_2)$, the gradient $\frac{\partial \text{vec}(W_2 X)}{\partial \text{vec}(W_2)}$ is

$$\frac{\partial \text{vec}(W_2 X)}{\partial \text{vec}(W_2)} = X^\top \otimes I_{n_1}.$$

Putting it all together, we have

$$\frac{\partial f}{\partial \text{vec}(W_2)} = -2R^\top(I_m \otimes W_1)\text{diag}(\text{vec}(\phi'(W_2 X))(X^\top \otimes I_{n_1}). \tag{41}$$

Writing that again as column vector yields

$$-2(X \otimes I_{n_1})\text{diag}(\text{vec}(\phi'(W_2 X)))(I_m \otimes W_1^\top)R.$$

After some modifications, we can write

$$\text{diag}(\text{vec}(\phi'(W_2 X)))(I_m \otimes W_1^\top)R =$$
$$\text{diag}(\text{vec}(\phi'(W_2 X)))\text{vec}(W_1^\top(Y - W_1\phi(W_2 X)))) =$$
$$\text{vec}((W_1^\top(Y - W_1\phi(W_2 X)) \odot \phi'(W_2 X)).$$

where $\odot$ is the Hadamard product.

This means that we can write the previous gradient as

$$-2\text{vec}(((W_1^\top(Y - W_1\phi(W_2 X))) \odot \phi'(W_2 X))X^\top).$$

For the first block of the Hessian, we differentiate $\frac{\partial f}{\partial \text{vec}(W_1)}$ with respect to $\partial \text{vec}(W_1)^\top$. Since

$$\frac{\partial f}{\partial \text{vec}(W_1)} = -2\text{vec}((Y - W_1\phi(W_2 X))\phi(W_2 X)^\top) = -2(\phi(W_2 X) \otimes I_c)\text{vec}(Y - W_1\phi(W_2 X)),$$

we have

$$\frac{\partial^2 f}{\partial \text{vec}(W_1)\text{vec}(W_1)^\top} = -2(\phi(W_2 X) \otimes I_c)\frac{\partial \text{vec}(Y - W_1\phi(W_2 X))}{\partial \text{vec}(W_1)^\top}$$
$$= 2(\phi(W_2 X) \otimes I_c)(\phi(W_2 X)^\top \otimes I_c)\frac{\partial \text{vec}(W_1)}{\partial \text{vec}(W_1)^\top}$$
$$= 2(\phi(W_2 X)\phi(W_2 X)^\top \otimes I_c).$$

For the off-diagonal blocks, it suffices to compute only one, as they are symmetric to each other. We use the product rule (see (Magnus, 1985), Theorem 9)

$$\frac{\partial \text{vec}(A(W)B(W))}{\partial \text{vec}(W)^\top} = (B(W)^\top \otimes I)\frac{\partial \text{vec}(A(W))}{\partial \text{vec}(W)^\top} + (I \otimes A(W))\frac{\partial \text{vec}(B(W))}{\partial \text{vec}(W)^\top}.$$

We have

$$\frac{\partial}{\partial \text{vec}(W_2)^\top} \frac{\partial f}{\partial \text{vec}(W_1)} = -2(\phi(W_2X) \otimes I_c)\frac{\partial \text{vec}(Y - W_1\phi(W_2X))}{\partial \text{vec}(W_2)^\top}$$
$$- 2(I_{n_1} \otimes (Y - W_1\phi(W_2X)))\frac{\partial \text{vec}(\phi(W_2X)^\top)}{\partial \text{vec}(W_2)^\top}.$$

In order to proceed, we need to write $\text{vec}(\phi(W_2X)^\top)$ in terms of $\text{vec}(\phi(W_2X))$, and this can be done formally using the so-called commutation matrix:

$$\text{vec}(\phi(W_2X)^\top) = K_{n_1m}\text{vec}(\phi(W_2X)).$$

For the first partial derivative in the sum, we have

$$\frac{\partial \text{vec}(Y - W_1\phi(W_2X))}{\partial \text{vec}(W_2)^\top} = -\frac{\partial \text{vec}(W_1\phi(W_2X))}{\partial \text{vec}(W_2)^\top} = -(I_m \otimes W_1)\frac{\partial \text{vec}(\phi(W_2X))}{\partial \text{vec}(W_2)^\top}$$
$$= -(I_m \otimes W_1)\text{diag}(\text{vec}(\phi'(W_2X)))\frac{\partial \text{vec}(W_2X)}{\partial \text{vec}(W_2)^\top}$$
$$= -(I_m \otimes W_1)\text{diag}(\text{vec}(\phi'(W_2X)))(X^\top \otimes I_{n_1}).$$

As it is evident in the previous calculation

$$\frac{\partial \text{vec}(\phi(W_2X))}{\partial \text{vec}(W_2)^\top} = \text{diag}(\text{vec}(\phi'(W_2X)))(X^\top \otimes I_{n_1}).$$

Putting it all together, we get

$$\frac{\partial^2 f}{\partial \text{vec}(W_1)\text{vec}(W_2)^\top} =$$
$$2(\phi(W_2X) \otimes W_1)\text{diag}(\text{vec}(\phi'(W_2X)))(X^\top \otimes I_{n_1})$$
$$-2(I_{n_1} \otimes (Y - W_1\phi(W_2X)))K_{n_1m}\text{diag}(\text{vec}(\phi'(W_2X)))(X^\top \otimes I_{n_1}) =$$
$$2(\phi(W_2X) \otimes W_1 + (I_{n_1} \otimes (W_1\phi(W_2X) - Y))K_{n_1m})\text{diag}(\text{vec}(\phi'(W_2X)))(X^\top \otimes I_{n_1}).$$

We also have

$$\frac{\partial^2 f}{\partial \text{vec}(W_2)\text{vec}(W_1)^\top} = \left(\frac{\partial^2 f}{\partial \text{vec}(W_1)\text{vec}(W_2)^\top}\right)^\top.$$

For the second derivative of $L$ with respect to $W_2$, we remind that

$$\frac{\partial f}{\partial \text{vec}(W_2)} = -2(X \otimes I_{n_1})\text{diag}(\text{vec}(\phi'(W_2X)))(I_m \otimes W_1^\top)R.$$

Differentiating that with respect to $\text{vec}(W_2)^\top$ involves a product rule, as $W_2$ appears in $\text{diag}(\text{vec}(\phi'(W_2X)))$ and in $R$. It is more convenient to bring $\frac{\partial f}{\partial \text{vec}(W_2)}$ back in fully vectorized form as:

$$\frac{\partial f}{\partial \text{vec}(W_2)} = -2\text{vec}(((W_1^\top(Y - W_1\phi(W_2X))) \odot \phi'(W_2X))X^\top).$$

We have

$$-2\frac{\partial \text{vec}(((W_1^\top(Y - W_1\phi(W_2X))) \odot \phi'(W_2X))X^\top)}{\partial \text{vec}(W_2)^\top} =$$
$$-2(X \otimes I_{n_1})\left(\frac{\partial \text{vec}(W_1^\top(Y - W_1\phi(W_2X)) \odot \phi'(W_2X))}{\partial \text{vec}(W_2)^\top}\right).$$

Now we can use the product rule for the Hadamard product, see (Magnus, 1985) (Theorem 10):

$$\frac{\partial \mathrm{vec}((W_1^\top(Y - W_1\phi(W_2X))) \odot \phi'(W_2X))}{\partial \mathrm{vec}(W_2)^\top} =$$

$$\mathrm{diag}(\mathrm{vec}(\phi'(W_2X))\frac{\partial \mathrm{vec}(W_1^\top(Y - W_1\phi(W_2X)))}{\partial \mathrm{vec}(W_2)^\top} + \mathrm{diag}(\mathrm{vec}(W_1^\top(Y - W_1\phi(W_2X))))\frac{\partial \phi'(W_2X)}{\partial \mathrm{vec}(W_2)^\top}.$$

For the first term of the last sum, we have by previous calculations that

$$\frac{\partial \mathrm{vec}(W_1^\top(Y - W_1\phi(W_2X)))}{\partial \mathrm{vec}(W_2)^\top} = -(I_m \otimes W_1^\top W_1)\mathrm{diag}(\mathrm{vec}(\phi'(W_2X)))(X^\top \otimes I_{n_1}).$$

For the second term of the last sum, we have

$$\frac{\partial \phi'(W_2X)}{\partial \mathrm{vec}(W_2)^\top} = \mathrm{diag}(\mathrm{vec}(\phi''(W_2X)))(X^\top \otimes I_{n_1}).$$

In total, we have

$$\frac{\partial^2 f}{\partial \mathrm{vec}(W_2)\mathrm{vec}(W_2)^\top} =$$
$$2(X \otimes I_{n_1})\mathrm{diag}(\mathrm{vec}(\phi'(W_2X)))(I_m \otimes W_1^\top W_1)\mathrm{diag}(\mathrm{vec}(\phi'(W_2X)))(X^\top \otimes I_{n_1})$$
$$-2(X \otimes I_{n_1})\mathrm{diag}(\mathrm{vec}(W_1^\top(Y - W_1\phi(W_2X))))\mathrm{diag}(\mathrm{vec}(\phi''(W_2X)))(X^\top \otimes I_{n_1}). \tag{42}$$

This completes the calculation of gradients and Hessian of $f$.

Now, we can simply consider $X = \begin{bmatrix} 1 & 0 & 0 \\ 0 & 1 & 0 \\ 0 & 0 & 1 \end{bmatrix}$ and $Y = \begin{bmatrix} 1 & 0 & 0 \\ 0 & 2 & 0 \\ 0 & 0 & 3 \end{bmatrix}$. Take also $W_1 = \begin{bmatrix} t & 0 \\ 0 & 0 \\ 0 & 0 \end{bmatrix}$ and $W_2 = \begin{bmatrix} \frac{1}{t} & 0 & 0 \\ \sqrt{t^2 - 1/t^2} & 0 & 0 \end{bmatrix}$, for $t > 1$ (notice that the entries of $W_2$ are positive, thus it is not affected by leaky-ReLU). It holds $\|W_1\|_F = t = \|W_2\|_F$, thus we indeed satisfy the weak balancedness condition. It also holds

$$Y - W_1W_2X = \begin{bmatrix} 0 & 0 & 0 \\ 0 & 2 & 0 \\ 0 & 0 & 3 \end{bmatrix}.$$

We can use that to compute

$$W_1^T(Y - W_1W_2X) = \begin{bmatrix} \frac{1}{t} & 0 & 0 \\ 0 & 0 & 0 \end{bmatrix}\begin{bmatrix} 0 & 0 & 0 \\ 0 & 2 & 0 \\ 0 & 0 & 3 \end{bmatrix} = \begin{bmatrix} 0 & 0 & 0 \\ 0 & 0 & 0 \end{bmatrix}$$

and

$$(Y - W_1W_2X)X^TW_2^T = \begin{bmatrix} 0 & 0 & 0 \\ 0 & 2 & 0 \\ 0 & 0 & 3 \end{bmatrix}\begin{bmatrix} \frac{1}{t} & \sqrt{t^2 - 1/t^2} \\ 0 & 0 \\ 0 & 0 \end{bmatrix} = \begin{bmatrix} 0 & 0 \\ 0 & 0 \\ 0 & 0 \end{bmatrix}.$$

It holds $\nabla_{W_1} f(W_1, W_2) = (Y - W_1W_2X)X^TW_2^T$ and $\nabla_{W_2} f(W_1, W_2) = W_1^T(Y - W_1W_2X)$, thus we have $\|\nabla f\| = 0$, while the Frobenius norm of the Hessian (thus also its spectral norm) goes to infinity at $t$ goes to infinity by Eq. (42), since

$$W_1^TW_1 = \begin{bmatrix} t^2 & 0 \\ 0 & 0 \end{bmatrix}.$$

$\square$

# G. Useful Lemmas

To obtain convergence results, we need to bound the smoothness between two arbitrary points $w, y \in \mathbb{R}^d$. This can be done using $(H_0, H_1)$-smoothness. If we parametrize $w(t) := x + t(y - w)$, then from the new smoothness assumption, the smoothness constant along the segment $[x, w(t)]$ can be bounded by a function of $\chi(t) := \int_0^t (H_0 + H_1(f(w(\theta)) - f^*)) d\theta$; see the next derivations for detailed proof. In particular, the bound on the smoothness constant along $[x, y]$ is related to $\chi(1)$. Our proof techniques are inspired by the results in Li et al. (2023). However, due to a more general smoothness inequality, our derivations involve a more careful analysis, because we need to deal with additional terms that do not appear in the case of $(L_0, L_1)$-smoothness. This highlights the difficulty of obtaining convergence guarantees, in particular, a gradient upper bound (Lemma 6) and quadratic upper bound (Lemma 7).

We start with a restatement of Lemma A.3 in Li et al. (2023), which can be seen as a generalization of Grönwall's inequality.

The next lemma provides sufficient conditions when the smoothness constant along the segment $[x, y]$ can be bounded by some constant, which uses information at $w$ only. The concurrent work Liu et al. (2025) provides similar derivations based on Li et al. (2023).

**Lemma 4** (Lemma A.3 in (Li et al., 2023)). *Let $\alpha: [a, b] \to [0, \infty)$ and $\beta: [0, \infty) \to (0, \infty)$ be two continuous functions. Suppose $\alpha'(t) \leq \beta(\alpha(t))$ almost everywhere over $(a, b)$. We have for all $K \in [a, b]$,*

$$\int_{\alpha(a)}^{\alpha(t)} \frac{1}{\beta(u)} du \leq t - a.$$

**Lemma 5.** *Let $f$ be $(H_0, H_1)$-smooth, $\chi(t) := \int_0^t (H_0 + H_1(f(w(\theta)) - f^*)) d\theta$, and define $c_1 := H_0 + H_1(c_2 + f(w) - f^*)$ for some $c_2 > 0$. Then it holds*

$$\|\nabla f(w)\| \cdot \|y - w\| + \frac{c_1}{2} \|y - w\|^2 \leq c_2 \implies \chi(1) \leq c_1.$$

*Proof.* Let $w(t) = x + t(y - w)$. Then, by Taylor's theorem for a gradient, we have

$$
\begin{aligned}
\|\nabla f(w(t)) - \nabla f(w)\| &= \left\| \int_0^t \nabla^2 f(w(\theta))(w(t) - w) d\theta \right\| \\
&\overset{(i)}{\leq} \int_0^t \|\nabla^2 f(w(\theta))\| d\theta \cdot \|w(t) - w\| \\
&\overset{(ii)}{\leq} \|w(t) - w\| \int_0^t (H_0 + H_1(f(w(\theta)) - f^*)) d\theta = \|w(t) - w\| \chi(t),
\end{aligned}
\tag{43}
$$

where $(i)$ follows from the definition of a spectral norm, $(ii)$ — from Definition 3.1, and $\chi(t) := \int_0^t (H_0 + H_1(f(w(\theta)) - f^*)) d\theta$. Eventually, we want to bound $\|\nabla f(y) - \nabla f(w)\|$ for any $y = w(1)$ and $w$. Therefore, our goal is to bound $\chi(1)$. Differentiating $\chi(t)$ we have

$$
\begin{aligned}
\chi'(t) &= (H_0 + H_1(f(w(t)) - f^*)) \\
&= H_0 + H_1(f(w) - f^*) + H_1(f(w(t)) - f(w)).
\end{aligned}
\tag{44}
$$

Now we need to bound the difference $f(w(t)) - f(w)$ using Taylor's theorem for the function

$$
\begin{aligned}
f(w(t)) - f(w) &= \int_0^t \langle \nabla f(w(\theta)), w(t) - x \rangle d\theta \\
&\overset{(iii)}{\leq} \langle \nabla f(w), w(t) - w \rangle + \|w(t) - w\| \int_0^t \|\nabla f(w(\theta)) - \nabla f(w)\| d\theta \\
&\overset{(iv)}{\leq} \langle \nabla f(w), w(t) - w \rangle + \|w(t) - w\| \int_0^t \chi(\theta) \|w(\theta) - w\| d\theta \\
&\overset{(v)}{\leq} \langle \nabla f(w), w(t) - w \rangle + \|w(t) - w\| \chi(t) \int_0^t \theta \|y - w\| d\theta \\
&\overset{(vi)}{\leq} \langle \nabla f(w), w(t) - w \rangle + \frac{1}{2} \|y - w\|^2 \chi(t),
\end{aligned}
\tag{45}
$$
$$
\tag{46}
$$

where $(iii)$ follows from Cauchy-Shawrz inequality, $(iv)$ — from (43), $(v)$ — from monotonicity of $\chi(t)$ by definition, $(vi)$ — from the fact that $t \in [0, 1]$. Using Cauchy-Schwarz inequality again, we obtain

$$f(w(t)) - f(w) \le \|\nabla f(w)\|\cdot\|w(t) - w\|+\frac{1}{2}\|y - w\|^2\chi(t) \le \|\nabla f(w)\|\cdot\|y - w\|$$
$$+ \frac{1}{2}\|y - w\|^2\chi(t), \qquad (47)$$

Therefore, we obtain from (44) and (47) that

$$\chi'(t) \le H_0 + H_1(f(w) - f^*) + H_1\|\nabla f(w)\|\cdot\|y - w\|+\frac{1}{2}H_1\|y - w\|^2\chi(t).$$

Now we use Lemma 4 with $\alpha(t) = \chi(t), \beta(t) := H_0 + H_1(f(w) - f^*) + H_1\|\nabla f(w)\|\cdot\|y - w\|+\frac{t}{2}H_1\|y - w\|^2, a = 0, b = 1$. We obtain

$$\int_0^{\chi(1)} \frac{1}{\beta(u)}du \le 1.$$

Note that since $\beta(t)$ is monotonically increasing in $t$, then for any $c_1 > 0$ such that $\beta(c_1) \le c_1$ we have

$$\int_0^{\chi(1)} \frac{1}{\beta(u)}du \le 1 \le \frac{c_1}{\beta(c_1)} \le \int_0^{c_1} \frac{1}{\beta(u)}du.$$

This implies that $\chi(1) \le c_1$ by monotonicity of the integral. Note that $\beta(c_1) \le c_1$ can be rewritten as

$$\beta(c_1) \le c_1$$
$$H_0 + H_1(f(w) - f^*) + H_1\|\nabla f(w)\|\cdot\|y - w\|+\frac{c_1}{2}H_1\|y - w\|^2 \le H_0 + H_1(c_2 + f(w) - f^*)$$
$$\|\nabla f(w)\|\cdot\|y - w\|+\frac{c_1}{2}\|y - w\|^2 \le c_2.$$

Therefore, $\chi(1) \le c_1$ if $\beta(c_1) \le c_1$ which is equivalent to

$$\|\nabla f(w)\|\cdot\|y - w\|+\frac{c_1}{2}\|y - w\|^2\le c_2.$$

$\square$

Our next lemma provides a gradient bound which can be seen as a generalization of the classic $\|\nabla f(w)\|^2\le 2L(f(w)-f^*)$ inequality, which holds in the $L$-smooth regime. The proof of Lemma 6 requires a careful choice of constants $c_1$ and $c_2$ from Lemma 5.

**Lemma 6.** *Let $f$ be $(H_0, H_1)$-smooth. Then we have*

$$\|\nabla f(w)\|^2\le \frac{9}{4}(H_0 + 3H_1(f(w) - f^*))(f(w) - f^*).$$

*Proof.* Let $c_2 := 2(f(w) - f^*)$, then $c_1 = H_0 + H_1(c_2 + f(w) - f^*) = H_0 + 3H_1(f(w) - f^*)$. From Lemma 5 we obtain that $\chi(1) \le c_1$ if

$$\|\nabla f(w)\|\cdot\|y - w\|+\frac{c_1}{2}\|y - w\|^2\le 2(f(w) - f^*) \Longleftrightarrow$$
$$\|\nabla f(w)\|\cdot\|y - w\|+\frac{1}{2}(H_0 + 3H_1(f(w) - f^*))\|y - w\|^2\le 2(f(w) - f^*).$$

This is a quadratic polynomial of $\|y - w\|$ which has one negative and one positive solution. Therefore, to satisfy the constraint above, we should have

$$\|y - w\|\le \frac{-\|\nabla f(w)\|+\sqrt{\|\nabla f(w)\|^2+4c_1(f(w) - f^*)}}{c_1} =: r.$$

In other words, we have $\chi(1) \leq c_1$ if points $y$ and $w$ satisfy the constraint above: $\|y - w\| \leq r$.

We choose $y := w - \eta \frac{\nabla f(w)}{\|\nabla f(w)\|}$, with $\eta \leq r$. Then, we have $\|y - w\| \leq r$, thus it holds $\chi(1) \leq c_1$.

From (46), the bound $\chi(1) \leq c_1$ implies

$$f(y) - f(w) \leq \langle \nabla f(w), y - w \rangle + \frac{c_1}{2}\|y - w\|^2.$$

Since $f(y) \geq f^*$ we can continue the inequality above as follows

$$f^* - f(w) \leq \langle \nabla f(w), y - w \rangle + \frac{c_1}{2}\|y - w\|^2.$$

Substituting $y = w - \eta \frac{\nabla f(w)}{\|\nabla f(w)\|}$, we get

$$-(f(w) - f^*) \leq -\eta\|\nabla f(w)\| + \frac{c_1}{2}\eta^2 \iff \frac{c_1}{2}\eta^2 - \eta\|\nabla f(w)\| + (f(w) - f^*) \geq 0. \tag{48}$$

(48) is a quadratic polynomial of $\eta$, which attains the minimum at $\eta = \frac{\|\nabla f(w)\|}{c_1}$. We consider two cases:

1. $\frac{\|\nabla f(w)\|}{c_1} \leq r$ : Plugging this bound into (48) implies

$$\frac{c_1}{2}\frac{\|\nabla f(w)\|^2}{c_1^2} - \|\nabla f(w)\|\frac{\|\nabla f(w)\|}{c_1} + (f(w) - f^*) \geq 0 \Leftrightarrow \|\nabla f(w)\|^2 \leq 2c_1(f(w) - f^*).$$

2. $\frac{\|\nabla f(w)\|}{c_1} > r$ : In this case, the minimum of the polynomial in (48) for $\eta = [0, r]$ is attained at $\eta = r$. Thus, from (48) we obtain

$$\frac{c_1}{2}r^2 - r\|\nabla f(w)\| + (f(w) - f^*) \geq 0 \Leftrightarrow \|\nabla f(w)\| \leq \frac{c_1}{2}r + \frac{f(w) - f^*}{r}.$$

We plug in the definition of $r$ in the bound above and obtain

$$\|\nabla f(w)\| \leq \frac{-\|\nabla f(w)\| + \sqrt{\|\nabla f(w)\|^2 + 4c_1(f(w) - f^*)}}{2}$$

$$+ \frac{c_1(f(w) - f^*)}{-\|\nabla f(w)\| + \sqrt{\|\nabla f(w)\|^2 + 4c_1(f(w) - f^*)}},$$

$$2\|\nabla f(w)\|(-\|\nabla f(w)\| + \sqrt{\|\nabla f(w)\|^2 + 4c_1(f(w) - f^*)})$$
$$\leq (-\|\nabla f(w)\| + \sqrt{\|\nabla f(w)\|^2 + 4c_1(f(w) - f^*)})^2 2c_1(f(w) - f^*),$$

$$-2\|\nabla f(w)\|^2 + 2\|\nabla f(w)\|\sqrt{\|\nabla f(w)\|^2 + 4c_1(f(w) - f^*)} \leq \|\nabla f(w)\|^2 + \|\nabla f(w)\|^2$$
$$+ 4c_1(f(w) - f^*) - 2\|\nabla f(w)\|\sqrt{\|\nabla f(w)\|^2 + 4c_1(f(w) - f^*)} + 2c_1(f(w) - f^*)$$

$$4\|\nabla f(w)\|\sqrt{\|\nabla f(w)\|^2 + 4c_1(f(w) - f^*)} \leq 4\|\nabla f(w)\|^2 + 6c_1(f(w) - f^*)$$

$$4\|\nabla f(w)\|^2(\|\nabla f(w)\|^2 + 4c_1(f(w) - f^*)) \leq 4\|\nabla f(w)\|^4 + 9c_1^2(f(w) - f^*)^2$$
$$+ 12c_1\|\nabla f(w)\|^2(f(w) - f^*),$$

$$4c_1\|\nabla f(w)\|^2(f(w) - f^*) \leq 9c_1^2(f(w) - f^*)^2,$$

$$\|\nabla f(w)\|^2 \leq \frac{9c_1}{4}(f(w) - f^*).$$

We obtain that, in both cases, we have the following bound

$$\|\nabla f(w)\|^2 \leq \max\left\{\frac{9}{4}, 2\right\}(H_0 + 3H_1(f(w) - f^*))(f(w) - f^*)$$

$$\leq \frac{9}{4}(H_0 + 3H_1(f(w) - f^*))(f(w) - f^*).$$

$\square$

Finally, we provide a quadratic upper bound on the change of the function value. This inequality will be useful later to demonstrate that GD returns iterates with decreasing function value.

**Lemma 7.** *Let $f$ be $(H_0, H_1)$-smooth. Then for any $y, w \in \mathbb{R}^d$ such that $\|y - w\| \leq \frac{1}{6\sqrt{H_1}}$ we have*

$$f(y) \leq f(w) + \langle \nabla f(w), y - w \rangle + \frac{2H_0 + 2H_1(f(w) - f^*)}{2} \|y - w\|^2.$$

*Proof.* From Lemma 5 we know that $\chi(1) \leq c_1$ if

$$\|\nabla f(w)\| \cdot \|y - w\| + \frac{c_1}{2} \|y - w\|^2 \leq c_2. \tag{49}$$

If we choose $y$ and $w$ such that

$$\|y - w\| \leq \min \left\{ \frac{c_2}{2\|\nabla f(w)\|}, \sqrt{\frac{c_2}{c_1}} \right\} =: r,$$

then (49) is satisfied. Indeed, this is due to

$$\|\nabla f(w)\| \cdot \|y - w\| \leq \|\nabla f(w)\| \frac{c_2}{2\|\nabla f(w)\|} = \frac{c_2}{2},$$

and

$$\frac{c_1}{2} \|y - w\|^2 \leq \frac{c_1}{2} \left( \sqrt{\frac{c_2}{c_1}} \right)^2 = \frac{c_2}{2}.$$

Therefore,

$$\|\nabla f(w)\| \cdot \|y - w\| + \frac{c_1}{2} \|y - w\|^2 \leq c_2.$$

Let $a_1 := \frac{9}{4}(H_0 + 3H_1(f(w) - f^*))(f(w) - f^*)$ and $a_2 = c_2(H_0 + H_1(c_2 + f(w) - f^*)) = c_1 c_2$. Then from Lemma 6 we have $\|\nabla f(w)\|^2 \leq a_1$. Therefore, we obtain that

$$r \geq \frac{2c_2}{\sqrt{a_1} + 2\sqrt{c_1 c_2}} = \frac{2c_2}{\sqrt{a_1} + 2\sqrt{a_2}}.$$

Let

$$c_2 = \max \left\{ f(w) - f^*, \frac{H_0}{3H_1} \right\}.$$

1. If $H_0 \leq 3H_1(f(w) - f^*)$, then $\frac{H_0}{3H_1} \leq f(w) - f^*$. Therefore, $c_2 = f(w) - f^*$. This implies that $a_1 \leq \frac{27}{2} H_1(f(w) - f^*)^2$, and

$$a_2 = c_2(H_0 + H_1(c_2 + f(w) - f^*)) = (f(w) - f^*)(H_0 + 2H_1(f(w) - f^*))$$
$$\leq 5H_1(f(w) - f^*)^2.$$

Moreover, in this case,

$$c_1 = H_0 + H_1(c_2 + f(w) - f^*) = H_0 + 2H_1(f(w) - f^*) \leq 5H_1(f(w) - f^*).$$

Therefore, in this case, we obtain

$$r \geq \frac{2(f(w) - f^*)}{\frac{3}{2}\sqrt{6H_1(f(w) - f^*)} + 2 \cdot \sqrt{5H_1(f(w) - f^*}} = \frac{2}{\frac{3}{2}\sqrt{6H_1} + 2\sqrt{5H_1}} \geq \frac{0.245}{\sqrt{H_1}}$$
$$\geq \frac{1}{5\sqrt{H_1}}.$$

2. If $H_0 > 3H_1(f(w) - f^*)$, then $\frac{H_0}{3H_1} > f(w) - f^*$. Therefore, $c_2 = \frac{H_0}{3H_1}$. This implies that

$$a_1 = \frac{9}{4}(H_0 + 3H_1(f(w) - f^*))(f(w) - f^*) \leq \frac{9}{2}H_0(f(w) - f^*) \leq \frac{3H_0^2}{2H_1},$$

$$a_2 = c_2(H_0 + H_1(c_2 + f(w) - f^*)) = \frac{H_0}{3H_1}(H_0 + H_1(H_0/3H_1 + f(w) - f^*))$$

$$\leq \frac{H_0}{3H_1}(H_0 + H_1 \cdot 2H_0/3H_1) = \frac{5H_0^2}{9H_1},$$

and

$$c_1 = H_0 + H_1(c_2 + f(w) - f^*) \leq H_0 + H_1\left(\frac{H_0}{3H_1} + \frac{H_0}{3H_1}\right) = \frac{5}{3}H_0.$$

Therefore, in this case, we obtain

$$r \geq \frac{2\frac{H_0}{3H_1}}{\frac{H_0}{2}\sqrt{\frac{6}{H_1}} + \frac{2H_0}{3}\sqrt{\frac{5}{H_1}}} = \frac{2/3}{\sqrt{H_1}(\frac{\sqrt{6}}{2} + \frac{2\sqrt{5}}{3})} \geq \frac{0.193}{\sqrt{H_1}} \geq \frac{1}{6\sqrt{H_1}}.$$

Combining both cases, we obtain that if

$$\|y - w\| \leq \min\left\{\frac{1}{5\sqrt{H_1}}, \frac{1}{6\sqrt{H_1}}\right\} = \frac{1}{6\sqrt{H_1}} \leq r,$$

we obtain $\chi(1) \leq c_1 \leq \max\{H_0 + 2H_1(f(w) - f^*), 5H_0/3\} \leq \frac{5H_0}{3} + 2H_1(f(w) - f^*) \leq 2H_0 + 2H_1(f(w) - f^*)$. From (46), this implies

$$f(y) - f(w) \leq \langle \nabla f(w), y - w \rangle + \frac{2H_0 + 2H_1(f(w) - f^*)}{2}\|y - w\|^2.$$

$\square$

# H. Missing Proofs for Section 4

## H.1. Convergence for General Non-Convex Functions

**Theorem H.1.** *Let $f$ be $(H_0, H_1)$-smooth. Then the iterates of GD $w_{k+1} = w_k - \eta_k \nabla f(w_k)$ where $\eta_k = \frac{1}{10H_0 + 20H_1(f(w_k) - f^*)}$ satisfy*

$$\min_{k < K}\|\nabla f(w_k)\|^2 \leq \frac{20(H_0 + 2H_1(f(w_0) - f^*))(f(w_0) - f^*)}{K} \frac{1}{1 + \frac{10H_1(f(w_0) - f^*)(K-1)(K-2)}{K^2(10H_0 + 20H_1(f(w_0) - f^*))}}.$$

*If $K \geq 6$, then the rate can be simplified*

$$\min_{k < K}\|\nabla f(w_k)\|^2 \leq \frac{20(H_0 + 2H_1(f(w_0) - f^*))(f(w_0) - f^*)}{K} \frac{1}{1 + \frac{H_1(f(w_0) - f^*)}{(2H_0 + 4H_1(f(w_0) - f^*))}}.$$

*Proof.* Note that $\|w_{k+1} - w_k\| = \eta_k\|\nabla f(w_k)\|$. Now we use Lemma 6 to obtain

$$\eta_k\|\nabla f(w_k)\| \leq \eta_k\frac{3}{2}\sqrt{(H_0 + 3H_1(f(w_k) - f^*))(f(w_k) - f^*)}.$$

1. If $H_0 \leq 3H_1(f(w_k) - f^*)$, then

$$\frac{3}{2}\eta_k\sqrt{6H_1(f(w_k) - f^*)^2} \leq \frac{3}{2}\eta_k\sqrt{6H_1}(f(w_k) - f^*). \tag{50}$$

We need to upper bound the above by $\frac{1}{6\sqrt{H_1}}$ to be able to use Lemma 7. We satisfy (50) by the choice of the step-size $\eta_k$

$$\eta_k\|\nabla f(w_k)\|\leq \frac{3}{2}\eta_k\sqrt{6H_1}(f(w_k) - f^*) \leq \frac{1}{6\sqrt{H_1}} \Leftrightarrow \eta_k \leq \frac{1}{\frac{15}{2}\sqrt{6H_1}(f(w_k) - f^*)},$$

where the last inequality is satisfied since

$$\eta_k = \frac{1}{10H_0 + 20H_1(f(w_k) - f^*)} \leq \frac{1}{20H_1(f(w_k) - f^*)} \leq \frac{1}{\frac{15}{2}\sqrt{6H_1}(f(w_k) - f^*)}.$$

2. If $H_0 > 3H_1(f(w_k) - f^*)$, then

$$\eta_k\|\nabla f(w_k)\|\leq \frac{3}{2}\eta_k\sqrt{2H_0(f(w_k) - f^*)} \leq \frac{3}{2}\eta_k\sqrt{2H_0 \cdot \frac{H_0}{3H_1}} = \eta_k\frac{\sqrt{3}H_0}{\sqrt{2H_1}}. \tag{51}$$

We need to upper bound the above by $\frac{1}{6\sqrt{H_1}}$ to be able to use Lemma 7. We satisfy (51) by the choice of the step-size $\eta_k$

$$\eta_k\|\nabla f(w_k)\|\leq \eta_k\frac{\sqrt{3}H_0}{\sqrt{2H_1}} \leq \frac{1}{6\sqrt{H_1}} \Leftrightarrow \eta_k \leq \frac{\sqrt{6}}{18H_0},$$

where the last inequality is satisfied since

$$\eta_k = \frac{1}{10H_0 + 20H_1(f(w_k) - f^*)} \leq \frac{1}{10H_0} \leq \frac{\sqrt{6}}{18H_0}.$$

Therefore, the choice of the step-size allows to use Lemma 7 since the restriction $\|w_{k+1} - w_k\|\leq \frac{1}{6\sqrt{H_1}}$ is satisfied. Therefore, we have

$$
\begin{aligned}
f(w_{k+1}) &\overset{(i)}{\leq} f(w_k) + \langle\nabla f(w_k), w_{k+1} - w_k\rangle + \frac{2H_0 + 2H_1(f(w_k) - f^*)}{2}\|w_{k+1} - w_k\|^2 \\
&\overset{(ii)}{\leq} f(w_k) - \eta_k\|\nabla f(w_k)\|^2 + (H_0 + H_1(f(w_k) - f^*))\eta_k^2\|\nabla f(w_k)\|^2 \\
&= f(w_k) - \eta_k\|\nabla f(w_k)\|^2(1 - \eta_k(H_0 + H_1(f(w_k) - f^*))) \\
&\overset{(iii)}{\leq} f(w_k) - \frac{\eta_k}{2}\|\nabla f(w_k)\|^2,
\end{aligned}
\tag{52}
$$

where $(i)$ follows from Lemma 7, $(ii)$ — from Lemma 6, $(iii)$ — from the choice of the step-size $\eta_k \leq \frac{1}{10H_0+20H_1(f(w_k)-f^*)}$. This implies that GD achieves a monotone decrease of the function value. By the choice of the step-size $\eta_k = \frac{1}{10H_0+20H_1(f(w_k)-f^*)}$, we obtain that $\eta_k$ is increasing with $k$. Rearranging the last inequality we obtain $\|\nabla f(w_k)\|^2\leq \frac{2}{\eta_k}(f(w_k) - f(w_{k+1}))$. Summing this inequality over iterations $\{0,\ldots, K - 1\}$ we obtain

$$\frac{1}{K}\sum_{k=0}^{K-1}\|\nabla f(w_k)\|^2 \le \frac{1}{K}\sum_{k=0}^{K-1}\frac{2}{\eta_k}(f(w_k)-f(w_{k+1}))$$

$$= \frac{1}{K}\sum_{k=0}^{K-1}(20H_0 + 40H_1(f(w_k)-f^*))(f(w_k)-f(w_{k+1}))$$

$$= \frac{20H_0}{K}\sum_{k=0}^{K-1}f(w_k)-f(w_{k+1})$$

$$+ \frac{40H_1}{K}\sum_{k=0}^{K-1}(f(w_k)-f^*)^2 - (f(w_k)-f^*)(f(w_{k+1})-f^*)$$

$$\overset{(iv)}{\le} \frac{20H_0}{K}\sum_{k=0}^{K-1}f(w_k)-f(w_{k+1})$$

$$+ \frac{40H_1}{K}\sum_{k=0}^{K-1}(f(w_k)-f^*)^2 - (f(w_{k+1})-f^*)^2$$

$$\le \frac{20H_0(f(w_0)-f^*)}{K} + \frac{40H_1(f(w_0)-f^*)^2}{K}.$$

The current rate is the same as with a constant step-size $\eta = \frac{1}{10H_0+20H_1(f(w_0)-f^*)}$, i.e., we do not show improvement. Now our goal is to obtain a tighter rate for GD using the fact that the sequence $\{\eta_k\}$ is increasing. By (52), we obtain

$$f(w_k) \le f(w_0) - \sum_{j=0}^{k-1}\frac{\eta_j}{2}\|\nabla f(w_j)\|^2 \Rightarrow f(w_k)-f^* \le (f(w_0)-f^*) - \sum_{j=0}^{k-1}\frac{\eta_j}{2}\|\nabla f(w_j)\|^2.$$

Therefore,

$$\frac{1}{\sum_{k=0}^{K-1}\eta_k}\sum_{k=0}^{K-1}\eta_k\|\nabla f(w_k)\|^2 \le \frac{2(f(w_0)-f^*)}{\sum_{k=0}^{K-1}\eta_k}.$$

To provide a tighter bound, we should take into account that the step-sizes are increasing since $f(w_k)-f^*$ is decreasing. Remember that $\eta_k = \frac{1}{10H_0+20H_1(f(w_k)-f^*)}$, then

$$\sum_{k=0}^{K-1}\eta_k = \sum_{k=0}^{K-1}\frac{1}{10H_0 + 20H_1(f(w_k)-f^*)}$$

$$\ge \sum_{k=0}^{K-1}\frac{1}{10H_0 + 20H_1\left(f(w_0)-f^* - \sum_{j=0}^{k-1}\frac{\eta_j}{2}\|\nabla f(w_j)\|^2\right)}.$$

Let us denote $\Lambda_k = \sum_{j=0}^{k-1}\eta_j\|\nabla f(w_j)\|^2$, then

$$\sum_{k=0}^{K-1}\eta_k \ge \sum_{k=0}^{K-1}\frac{1}{10H_0 + 20H_1(f(w_0)-f^*) - 10H_1\Lambda_k}.$$

Since the function $u \to g(u) := \frac{1}{10H_0+20H_1(f(w_0)-f^*)-10H_1u}$ is convex in the set $\{u \in \mathbb{R} \mid g(u) > 0\}$, then by Jensen's inequality we have

$$\frac{1}{K}\sum_{k=0}^{K-1}g(\Lambda_k) \ge g\left(\frac{1}{K}\sum_{k=0}^{K-1}\Lambda_k\right).$$

In our case, we obtain

$$\sum_{k=0}^{K-1} \eta_k \geq \sum_{k=0}^{K-1} g(\Lambda_k) \geq \frac{K}{10H_0 + 20H_1(f(w_0) - f^*) - \frac{10H_1}{K}\sum_{k=0}^{K-1}\Lambda_k}.$$

Now we estimate

$$\sum_{k=0}^{K-1} \Lambda_k = \sum_{k=0}^{K-1}\sum_{j=0}^{k-1} \eta_j\|\nabla f(w_j)\|^2 \geq \min_{k<K}\|\nabla f(w_k)\|^2 \sum_{k=0}^{K-1}\sum_{j=0}^{k-1} \eta_j \geq \min_{k<K}\|\nabla f(w_k)\|^2 \eta_0 \frac{(K-1)K}{2},$$

where we use the fact that $\eta_0 \leq \eta_k$ for all $k \geq 0$. This leads to the following bound

$$\min_{k<K}\|\nabla f(w_k)\|^2 \leq \frac{1}{\sum_{k=0}^{K-1}\eta_k}\sum_{k=0}^{K-1}\eta_k\|\nabla f(w_k)\|^2$$

$$\leq \frac{2(f(w_0) - f^*)}{\frac{K}{10H_0+20H_1(f(w_0)-f^*)-\frac{10H_1}{K}\eta_0\frac{(K-1)(K-2)}{2}\min_k\|\nabla f(w_k)\|^2}}$$

$$\leq \frac{2(10H_0 + 20H_1(f(w_0) - f^*))(f(w_0) - f^*)}{K}$$

$$- \frac{10H_1(f(w_0) - f^*)(K-1)(K-2)\eta_0\min_k\|\nabla f(w_k)\|^2}{K^2}.$$

Rearranging the terms, we obtain

$$\min_{k<K}\|\nabla f(w_k)\|^2 \leq \frac{20(H_0 + 2H_1(f(w_0) - f^*))(f(w_0) - f^*)}{K}\frac{1}{1 + \frac{10H_1(f(w_0)-f^*)(K-1)(K-2)}{K^2(10H_0+20H_1(f(w_0)-f^*))}}.$$

If $K \geq 6$, then $\frac{10(K-1)(K-2)}{K^2} \geq 5$, which leads to the simplified rate. $\qquad\square$

## H.2. Convergence under Aiming Condition

**Theorem 4.2.** *Assume that $f$ is $(H_0, H_1)$-smooth, and it satisfies the Aiming condition with constant $\theta$ around the set of global minimizers $\mathcal{S}$. Then the iterates of* GD *with adaptive step-size $\theta \cdot \eta_k$ satisfy*

$$f(w_K) - f^* \leq \varepsilon \quad \text{after at most}$$
$$\frac{40H_0\text{dist}(w_0, \mathcal{S})^2}{\theta^2\varepsilon} + \frac{40H_1\text{dist}(w_0, \mathcal{S})^2}{\theta^2} \quad \text{iterations.}$$

*Proof.* We start by (52)

$$f(w_{k+1}) \leq f(w_k) - \frac{\eta_k}{2}\|\nabla f(w_k)\|^2 = f(w_k) - \frac{\theta}{20H_0 + 40H_1(f(w_k) - f^*)}\|\nabla f(w_k)\|^2. \tag{53}$$

Next, we show that the distance to the set of global minimizers $\mathcal{S}$ of the function $f$ does not increase. Indeed, we have

$$\text{dist}(w_{k+1}, \mathcal{S})^2 \overset{(i)}{=} \|w_{k+1} - \pi_{\mathcal{S}}(w_k)\|^2$$

$$= \|w_k - \pi_{\mathcal{S}}(w_k)\|^2 - 2\eta_k\langle w_k - \pi_{\mathcal{S}}(w_k), \nabla f(w_k)\rangle + \eta_k^2\|\nabla f(w_k)\|^2$$

$$\overset{(ii)}{\leq} \text{dist}(w_k, \mathcal{S})^2 - 2\eta_k\theta(f(w_k) - f^*) + \eta_k^2\|\nabla f(w_k)\|^2$$

$$\overset{(iii)}{\leq} \text{dist}(w_k, \mathcal{S})^2 - 2\eta_k\theta(f(w_k) - f^*)$$

$$+ \frac{9\eta_k^2}{4}(H_0 + 3H_1(f(w_k) - f^*))(f(w_k) - f^*)$$

$$= \text{dist}(w_k, \mathcal{S})^2 - 2\eta_k(f(w_k) - f^*)\left(\theta - \frac{9}{8}\eta_k(H_0 + 3H_1(f(w_k) - f^*))\right),$$

where $(i)$ follows from the definition of the projection, $(ii)$ follows from the definition of the Aiming condition, $(iii)$ — from Lemma 6. Now we use the choice of the step-size $\eta_k = \frac{\theta}{10H_0 + 20H_1(f(w_k) - f^*)}$ to obtain

$$\text{dist}(w_{k+1}, \mathcal{S})^2 \leq \text{dist}(w_k, \mathcal{S})^2 - \eta_k \theta(f(w_k) - f^*). \tag{54}$$

Therefore, we have that $\text{dist}(w_{k+1}, \mathcal{S})^2 \leq \text{dist}(w_k, \mathcal{S})^2$ for any $k \geq 0$. Now we consider two cases:

- $f(w_k) - f^* \geq \frac{H_0}{2H_1}$ (large function value). In this case, we can lower bound the step-size as

$$\eta_k = \frac{\theta}{10H_0 + 20H_1(f(w_k) - f^*)} \geq \frac{\theta}{40H_1(f(w_k) - f^*)}.$$

Therefore, from (54), we obtain

$$\begin{aligned}
\text{dist}(w_{k+1}, \mathcal{S})^2 &\leq \text{dist}(w_k, \mathcal{S})^2 - \eta_k \theta(f(w_k) - f^*) \\
&\leq \text{dist}(w_k, \mathcal{S})^2 - \frac{\theta}{40H_1(f(w_k) - f^*)}\theta(f(w_k) - f^*) \\
&= \text{dist}(w_k, \mathcal{S})^2 - \frac{\theta^2}{40H_1}.
\end{aligned}$$

Since $\text{dist}(w_k, \mathcal{S})^2 \geq 0$, we can stay in this regime at most $T$ iterations, such that

$$0 \leq \text{dist}(w_T, \mathcal{S})^2 \leq \text{dist}(w_0, \mathcal{S})^2 - \frac{\theta^2}{40H_1}T \Rightarrow T := \frac{40H_1\text{dist}(w_0, \mathcal{S})^2}{\theta^2}.$$

- $f(w_k) - f^* \leq \frac{H_0}{2H_1}$ (small function value). In this case, we can lower bound the step-size as

$$\eta_k = \frac{\theta}{10H_0 + 20H_1(f(w_k) - f^*)} \geq \frac{\theta}{20H_0}.$$

Therefore, from (54), we obtain

$$\begin{aligned}
\text{dist}(w_{k+1}, \mathcal{S})^2 &\leq \text{dist}(w_k, \mathcal{S})^2 - \eta_k \theta(f(w_k) - f^*) \\
&\leq \text{dist}(w_k, \mathcal{S})^2 - \frac{\theta^2}{40H_0}(f(w_k) - f^*).
\end{aligned}$$

Rearranging the terms, we obtain

$$f(w_k) - f^* \leq \frac{40H_0}{\theta^2}(\text{dist}(w_k, \mathcal{S})^2 - \text{dist}(w_{k+1}, \mathcal{S})^2). \tag{55}$$

Averaging the inequalities (55) for $k \in \{T, \ldots, K\}$, we obtain

$$\begin{aligned}
\frac{1}{K - T + 1}\sum_{k=T}^{K}(f(w_k) - f^*) &\leq \frac{40H_0(\text{dist}(w_0, \mathcal{S})^2 - \text{dist}(w_{K+1}, \mathcal{S})^2)}{\theta^2(K - T + 1)} \\
&\leq \frac{40H_0\text{dist}(w_0, \mathcal{S})^2}{\theta^2(K - T + 1)}.
\end{aligned}$$

Since $f(w_k) - f^*$ is decreasing by (53), we have

$$f(w_K) - f^* \leq \frac{40H_0\text{dist}(w_0, \mathcal{S})^2}{\theta^2(K - T + 1)}.$$

To achieve $\varepsilon$ accuracy, we need the number of iterations $K$ to be

$$\begin{aligned}
f(w_K) - f^* \leq \frac{40H_0\text{dist}(w_0, \mathcal{S})^2}{\theta^2(K - T + 1)} \leq \varepsilon &\Rightarrow K \geq \frac{40H_0\text{dist}(w_0, \mathcal{S})^2}{\theta^2\varepsilon} + T \\
&= \frac{40H_0\text{dist}(w_0, \mathcal{S})^2}{\theta^2\varepsilon} + \frac{40H_1\text{dist}(w_0, \mathcal{S})^2}{\theta^2}.
\end{aligned}$$

$\square$

The next theorem demonstrates that when the function suboptimality is large, we should expect a linear decrease. This gives another intuition behind the improvement from the warm-up schedule. This result demonstrates that linear convergence can be expected even beyond the PL case.

**Theorem H.2.** *Assume that $f$ is $(H_0, H_1)$-smooth, and it satisfies the Aiming condition with constant $\theta$ around the set of global minimizers $\mathcal{S}$. Assume that $f(w_k) - f^* \geq \frac{H_0}{2H_1}$. Then the iterates of GD $w_{k+1} = w_k - \eta_k \nabla f(w_k)$ with a step-size $\eta_k = \frac{\theta}{10H_0 + 20H_1(f(w_k) - f^*)}$ satisfy*

$$f(w_{k+1}) - f^* \leq \left(1 - \frac{\theta^3}{80H_1 \text{dist}(w_0, \mathcal{S})^2}\right)(f(w_k) - f^*).$$

*Proof.* First, we use the previously derived decrease in the function value (53)

$$f(w_{k+1}) - f^* \leq f(w_k) - f^* - \frac{\theta}{20H_0 + 40H_1(f(w_k) - f^*)}\|\nabla f(w_k)\|^2,$$

and in the distance (54)

$$\text{dist}(w_{k+1}, \mathcal{S})^2 \leq \text{dist}(w_k, \mathcal{S})^2 - \eta_k \theta(f(w_k) - f^*).$$

In particular, $\text{dist}(w_k, \mathcal{S})^2 \leq \text{dist}(w_0, \mathcal{S})^2$. From the Aiming condition, we have

$$\begin{aligned} \theta(f(w_k) - f^*) \leq \langle \nabla f(w_k), w_k - \pi_{\mathcal{S}}(w_k) \rangle &\leq \|\nabla f(w_k)\| \cdot \text{dist}(w_k, \mathcal{S}) \\ &\leq \|\nabla f(w_k)\| \cdot \text{dist}(w_0, \mathcal{S}). \end{aligned} \tag{56}$$

Therefore, we obtain

$$\begin{aligned} f(w_{k+1}) - f^* &\leq f(w_k) - f^* - \frac{\theta}{20H_0 + 40H_1(f(w_k) - f^*)}\|\nabla f(w_k)\|^2 \\ &\overset{(i)}{\leq} f(w_k) - f^* - \frac{\theta}{80H_1(f(w_k) - f^*)}\|\nabla f(w_k)\|^2 \\ &\overset{(ii)}{\leq} f(w_k) - f^* - \frac{\theta}{80H_1(f(w_k) - f^*)} \frac{\theta^2(f(w_k) - f^*)^2}{\text{dist}(w_0, \mathcal{S})^2} \\ &= \left(1 - \frac{\theta^3}{80H_1 \text{dist}(w_0, \mathcal{S})^2}\right)(f(w_k) - f^*). \end{aligned}$$

where $(i)$ follows from the bound $f(w_k) - f^* \geq \frac{H_0}{2H_1}$, $(ii)$ – from (56). $\qquad\square$

## H.3. Convergence under Polyak-Łojasiewicz Condition

**Theorem 4.3.** *Assume that $f$ is $(H_0, H_1)$-smooth, and it satisfies $\mu$-PL condition. Then the iterates of GD with adaptive step-size $\eta_k$ satisfy*

$$f(w_K) - f^* \leq \varepsilon \quad \text{after at most}$$
$$\frac{40H_1}{\mu}(f(w_0) - f^*) + \frac{20H_0}{\mu}\log\frac{H_0}{2H_1\varepsilon} \quad \text{iterations.}$$

*Proof.* We start with the Eq. (52) and use $\mu$-PL inequality

$$\begin{aligned} f(w_{k+1}) &\leq f(w_k) - \frac{\eta_k}{2}\|\nabla f(w_k)\|^2 \\ &\leq f(w_k) - \mu\eta_k(f(w_k) - f^*) \\ &= f(w_k) - \frac{\mu(f(w_k) - f^*)}{10H_0 + 20H_1(f(w_k) - f^*)}. \end{aligned}$$

Now we consider two cases.

- $f(w_k) - f^* \geq \frac{H_0}{2H_1}$ (large function value). In this case, we have

$$f(w_{k+1}) \leq f(w_k) - \frac{\mu(f(w_k) - f^*)}{10H_0 + 20H_1(f(w_k) - f^*)}$$
$$\leq f(w_k) - \frac{\mu(f(w_k) - f^*)}{40H_1(f(w_k) - f^*)}$$
$$= f(w_k) - \frac{\mu}{40H_1}.$$

Since GD decreases the function value (see (52)), we have $f(w_t) - f^* \geq \frac{H_0}{2H_1}$ for all $K \in \{0, \ldots, k\}$. Therefore,

$$f(w_{k+1}) - f^* \leq f(w_0) - f^* - \frac{\mu}{40H_1}(k+1).$$

However, we cannot reduce the function value infinitely many times, since it is lower bounded. We can stay in this regime as long as $f(w_t) - f^* \geq \frac{H_0}{2H_1}$, therefore, GD stays in this regime for at most $k \leq \frac{40H_1}{\mu}\left(f(w_0) - f^* - \frac{H_0}{2H_1}\right) - 1 \leq \frac{40H_1}{\mu}(f(w_0) - f^*) - \frac{20H_0}{\mu}$ iterations. In other words, the cardinality of the set $\mathcal{T} := \{k \in \{0, \ldots, K-1\}: f(w_k) - f^* \geq \frac{H_0}{2H_1}\}$ is bounded by $T = \frac{40H_1}{\mu}(f(w_0) - f^*) - \frac{20H_0}{\mu}$.

- $f(w_k) - f^* \leq \frac{H_0}{2H_1}$ (small function value). In this case, we have

$$f(w_{k+1}) \leq f(w_k) - \frac{\mu(f(w_k) - f^*)}{10H_0 + 20H_1(f(w_k) - f^*)}$$
$$\leq f(w_k) - \frac{\mu(f(w_k) - f^*)}{20H_0}. \tag{57}$$

Since the function along the trajectory of GD does not increase (see (52)), we stay in this regime for the rest of the training. Therefore, summing up (57) for all iterations $k \in \{T, \ldots, K-1\}$ we obtain

$$f(w_K) - f^* \leq \left(1 - \frac{\mu}{20H_0}\right)(f(w_{K-1}) - f^*)$$
$$\leq \ldots$$
$$\leq \left(1 - \frac{\mu}{20H_0}\right)^{K-T}(f(w_T) - f^*).$$

Since $f(w_T) - f^* \leq f(w_0) - f^* - \frac{\mu T}{40H_1}$, we get the rate

$$f(w_K) - f^*$$
$$\leq \left(1 - \frac{\mu}{20H_0}\right)^{K-T}\left(f(w_0) - f^* - \frac{\mu}{40H_1}\left(\frac{40H_1}{\mu}(f(w_0) - f^*) - \frac{20H_0}{\mu}\right)\right)$$
$$= \left(1 - \frac{\mu}{20H_0}\right)^{K-T}\frac{H_0}{2H_1}.$$

To achieve $f(w_K) - f^* \leq \varepsilon$ we need to satisfy

$$f(w_K) - f^* \leq \left(1 - \frac{\mu}{20H_0}\right)^{K-T}\frac{H_0}{2H_1} \leq \varepsilon \Rightarrow K \geq T + \frac{20H_0}{\mu}\log\frac{H_0}{2H_1\varepsilon}$$
$$= \frac{40H_1}{\mu}(f(w_0) - f^*) + \frac{20H_0}{\mu}\log\frac{H_0}{2H_1\varepsilon}.$$

$\square$

## H.4. Convergence in the Stochastic Setting

**Theorem 4.4.** *Assume that the problem* (P) *satisfies the interpolation condition. Assume that each $f_i$ is $(H_0, H_1)$-smooth and satisfies the Aiming condition around the set of global minimizers $\mathcal{S}$. Then the iterates of SGD $w_{k+1} = w_k - \eta_k \nabla f_{S_k}(w_k)$ with batch $S_k \subseteq [n]$ and a step-size*

$$\eta_k = \frac{\theta}{10H_0 + 20H_1(f_{S_k}(w_k) - f_{S_k}^*)}$$

*satisfy*

$$\frac{1}{K+1} \sum_{k=0}^{K} \mathbb{E}\left[\min\left\{f(w_k) - f^*, \frac{H_0}{2nH_1}\right\}\right]$$
$$\leq \frac{20H_0 \text{dist}(w_0, \mathcal{S})^2}{\theta^2(K+1)}.$$

*Proof.* We show that the distance to the set of global minimizers $\mathcal{S}$ of the function $f$ does not increase. Indeed, we have

$$
\begin{aligned}
\text{dist}(w_{k+1}, \mathcal{S})^2 &= \|w_{k+1} - \pi_{\mathcal{S}}(w_{k+1})\|^2 \\
&\leq \|w_{k+1} - \pi_{\mathcal{S}}(w_k)\|^2 \\
&= \|w_k - \pi_{\mathcal{S}}(w_k)\|^2 - 2\eta_k\langle w_k - \pi_{\mathcal{S}}(w_k), \nabla f_{S_k}(w_k)\rangle + \eta_k^2\|\nabla f_{S_k}(w_k)\|^2 \\
&\overset{(i)}{\leq} \|w_k - \pi_{\mathcal{S}}(w_k)\|^2 - 2\theta\eta_k(f_{S_k}(w_k) - f_{S_k}^*) + \eta_k^2\|\nabla f_{S_k}(w_k)\|^2 \\
&\overset{(ii)}{\leq} \text{dist}(w_k, \mathcal{S})^2 - 2\theta\eta_k(f_{S_k}(w_k) - f_{S_k}^*) \\
&\qquad + \frac{9}{4}\eta_k^2(H_0 + 3H_1(f_{S_k}(w_k) - f_{S_k}^*))(f_{S_k}(w_k) - f_{S_k}^*) \\
&\overset{(iii)}{=} \text{dist}(w_k, \mathcal{S})^2 - 2\eta_k(f_{S_k}(w_k) - f_{S_k}(w^*)) \\
&\qquad + \frac{9}{4}\eta_k^2(H_0 + 3H_1(f_{S_k}(w_k) - f_{S_k}(w^*)))(f_{S_k}(w_k) - f_{S_k}(w^*)) \\
&= \text{dist}(w_k, \mathcal{S})^2 - 2\eta_k(f_{S_k}(w_k) - f_{S_k}(w^*))\left(\theta - \frac{9}{8}\eta_k(H_0 + 3H_1(f_{S_k}(w_k) - f_{S_k}(w^*)))\right)
\end{aligned}
$$

where $(i)$ follows from Definition 4.1, $(ii)$ — from Lemma 6, $(iii)$ — from the interpolation condition. Now we use the choice of the step-size

$$\eta_k = \frac{\theta}{10H_0 + 20H_1(f_{S_k}(w_k) - f_{S_k}^*)} = \frac{\theta}{10H_0 + 20H_1(f_{S_k}(w_k) - f_{S_k}(w^*))}$$

to obtain

$$\text{dist}(w_{k+1}, \mathcal{S})^2 \leq \text{dist}(w_k, \mathcal{S})^2 - \eta_k\theta(f_{S_k}(w_k) - f_{S_k}(w^*)). \tag{58}$$

Therefore, we have that $\text{dist}(w_{k+1}, \mathcal{S})^2 \leq \text{dist}(w_k, \mathcal{S})^2$ for any $k \geq 0$. Now we consider two cases:

- $f_{S_k}(w_k) - f_{S_k}(w^*) \geq \frac{H_0}{2H_1}$ (large function value). In this case, we can lower bound the step-size $\eta_k$ as

$$\eta_k = \frac{\theta}{10H_0 + 20H_1(f_{S_k}(w_k) - f_{S_k}(w^*))} \geq \frac{\theta}{40H_1(f_{S_k}(w_k) - f_{S_k}(w^*))}.$$

Therefore, from (58), we obtain

$$
\begin{aligned}
\text{dist}(w_{k+1}, \mathcal{S})^2 &\leq \text{dist}(w_k, \mathcal{S})^2 - \eta_k\theta(f_{S_k}(w_k) - f_{S_k}(w^*)) \\
&\leq \text{dist}(w_k, \mathcal{S})^2 - \frac{\theta^2}{40H_1(f_{S_k}(w_k) - f_{S_k}(w^*))}(f_{S_k}(w_k) - f_{S_k}(w^*)) \\
&= \text{dist}(w_k, \mathcal{S})^2 - \frac{\theta^2}{40H_1}. \tag{59}
\end{aligned}
$$

- $f_{S_k}(w_k) - f_{S_k}(w^*) \leq \frac{H_0}{2H_1}$ (small function value). In this case, we can lower bound the step-size $\eta_k$ as

$$\eta_k = \frac{\theta}{10H_0 + 20H_1(f_{S_k}(w_k) - f_{S_k}(w^*))} \geq \frac{\theta}{20H_0}.$$

Therefore, from (58), we obtain

$$\text{dist}(w_{k+1}, \mathcal{S})^2 \leq \text{dist}(w_k, \mathcal{S})^2 - \eta_k \theta(f_{S_k}(w_k) - f_{S_k}(w^*))$$

$$\leq \text{dist}(w_k, \mathcal{S})^2 - \frac{\theta^2}{20H_0}(f_{S_k}(w_k) - f_{S_k}(w^*)). \tag{60}$$

To combine descent inequalities (59) and (60), we introduce the even $E(w_k) := \left\{ f_{S_k}(w_k) - f_{S_k}(w^*) \geq \frac{H_0}{2H_1} \mid w_k \right\}$ for given $w_k$ and its indicator function $\mathbb{1}_{E(w_k)}$, i.e., for given $w_k$, $\mathbb{1}_{E(w_k)} = 1$ if $f_{S_k}(w_k) - f_{S_k}(w^*) \geq \frac{H_0}{2H_1}$, and $\mathbb{1}_{E(w_k)} = 0$ if $f_{S_k}(w_k) - f_{S_k}(w^*) < \frac{H_0}{2H_1}$. Then the descent in the general case can be written as

$$\text{dist}(w_{k+1}, \mathcal{S})^2 \leq \text{dist}(w_k, \mathcal{S})^2 - \mathbb{1}_{E(w_k)} \frac{\theta^2}{40H_1} - (1 - \mathbb{1}_{E(w_k)}) \frac{\theta^2}{20H_0}(f_{S_k}(w_k) - f_{S_k}(w^*)). \tag{61}$$

We denote $\mathbb{E}_k[\cdot]$ as $\mathbb{E}[\cdot \mid w_k]$ – the expectation conditioned on $w_k$. Thus, we have from (61) that

$$\mathbb{E}_k\left[\text{dist}(w_{k+1}, \mathcal{S})^2\right] \leq \text{dist}(w_k, \mathcal{S})^2 - \frac{\theta^2}{20H_0}\mathbb{E}_k\left[(1 - \mathbb{1}_{E(w_k)})(f_{S_k}(w_k) - f_{S_k}(w^*))\right]$$

$$- \mathbb{E}_k\left[\mathbb{1}_{E(w_k)}\right]\frac{\theta^2}{40H_1}$$

$$= \text{dist}(w_k, \mathcal{S})^2 - \frac{\theta^2}{20H_0}\mathbb{E}_k\left[(1 - \mathbb{1}_{E(w_k)})(f_{S_k}(w_k) - f_{S_k}(w^*))\right]$$

$$- p_k\frac{\theta^2}{40H_1}, \tag{62}$$

where $p_k := \mathbb{E}_k\left[\mathbb{1}_{E(w_k)}\right] = \mathbb{P}(E(w_k)) = \mathbb{P}(f_{S_k}(w_k) - f_{S_k}(w^*) \geq \frac{H_0}{2H_1})$. We emphasize that $p_k$ is a random variable. If $p_k > 0$, then there is at least one $i \in [n]$, so that $f_i(w_k) - f_i(w^*) \geq \frac{H_0}{2H_1}$ for given $w_k$. Thus, we have $p_k \geq \frac{1}{n}$. In the opposite case, we have $p_k = 0$, and $1 - \mathbb{1}_{E(w_k)} = 1$ for given $w_k$. Putting all together, we continue as follows

$$\mathbb{E}_k\left[\text{dist}(w_{k+1}, \mathcal{S})^2\right] \leq \text{dist}(w_k, \mathcal{S})^2 - \frac{\theta^2}{20H_0}\mathbb{1}_{\{p_k=0\}}(f(w_k) - f(w^*)) - \mathbb{1}_{\{p_k>0\}}p_k\frac{\theta^2}{40H_1}$$

$$\leq \text{dist}(w_k, \mathcal{S})^2 - \frac{\theta^2}{20H_0}\mathbb{1}_{\{p_k=0\}}(f(w_k) - f(w^*)) - \mathbb{1}_{\{p_k>0\}}\frac{\theta^2}{40nH_1}$$

$$\leq \text{dist}(w_k, \mathcal{S})^2 - \min\left\{\frac{\theta^2}{20H_0}(f(w_k) - f(w^*)), \frac{\theta^2}{40nH_1}\right\}.$$

Taking full expectation and rearranging terms, we obtain

$$\sum_{k=0}^{K}\mathbb{E}\left[\min\left\{\frac{\theta^2}{20H_0}(f(w_k) - f(w^*)), \frac{\theta^2}{40nH_1}\right\}\right] \leq \sum_{k=0}^{K+1}\mathbb{E}\left[\text{dist}(w_k, \mathcal{S})^2\right] - \mathbb{E}\left[\text{dist}(w_{k+1}, \mathcal{S})^2\right]$$

$$\leq \text{dist}(w_0, \mathcal{S})^2.$$

Dividing both sides by $\frac{\theta^2}{20H_0(K+1)}$, we obtain

$$\frac{1}{K+1}\sum_{k=0}^{K}\mathbb{E}\left[\min\left\{f(w_k) - f(w^*), \frac{H_0}{2nH_1}\right\}\right] \leq \frac{20H_0\text{dist}(w_0, \mathcal{S})^2}{\theta^2(K+1)}.$$

The rate above implies

$$\min_{k<K+1}\mathbb{E}\left[\min\left\{f(w_k) - f(w^*), \frac{H_0}{2nH_1}\right\}\right] \leq \frac{20H_0\text{dist}(w_0, \mathcal{S})^2}{\theta^2(K+1)}.$$

$\square$

## I. Missing Proofs for GD in the Convex Setting

In this case, we demonstrate the convergence to the minimizer $w^*$ of the convex function $f$.

*Proof.* The proof mainly follows the proof of Theorem 4.2 by setting $\theta = 1$ and $\mathcal{S} = \{w^*\}$. $\square$

## J. Lower Bounds

**Theorem 4.1.** *Let $f$ belong to the class $\mathcal{H}$ of $(H_0, H_1)$-smooth functions. Then it holds:*

1. *To satisfy $\|\nabla f(w_K)\| \leq \varepsilon$ for a general non-convex function $f$, GD with constant step-size initialized at $w_0$, needs at least*

$$K \geq \frac{H_1(f(w_0) - f^*)}{\log(f(w_0) - f^*) + 1} \frac{f(w_0) - f^* - 2\epsilon^2}{8\epsilon^2}$$

   *iterations.*

2. *To satisfy $f(w_K) - f^* \leq \varepsilon$ for convex function $f$, GD with constant step-size initialized at $w_0$, needs at least*

$$K \geq \frac{H_1(f(w_0) - f^*)}{\log(f(w_0) - f^*) + 1} \frac{f(w_0) - f^* - \epsilon}{4\epsilon}$$

   *iterations.*

3. *To satisfy $f(w_K) - f^* \leq \varepsilon$ for $\mu$-PL function $f$ (but not necessarily convex), GD with constant step-size initialized at $w_0$, needs at least*

$$K \geq \frac{H_1}{4\mu} \frac{(f(w_0) - f^*)}{\log(f(w_0) - f^*) + 1} \log\left(\frac{f(w_0) - f^*}{\epsilon}\right)$$

   *iterations.*

*Proof.* Consider constants $H_1, M > 1$ and the function

$$f(w) = \begin{cases} \frac{e^{-\sqrt{H_1}w}}{e}, & \text{if } w < -\frac{1}{\sqrt{H_1}} \\ \frac{H_1 w^2}{2} + \frac{1}{2}, & \text{if } w \in \left[-\frac{1}{\sqrt{H_1}}, \frac{1}{\sqrt{H_1}}\right] \\ \frac{e^{\sqrt{H_1}w}}{e}, & \text{if } w > \frac{1}{\sqrt{H_1}}. \end{cases}$$

This function is $(H_0, H_1)$-smooth with $H_0 \geq H_1$ and convex, thus it also belongs to the objective function class.

We consider GD for the function $f$ starting from the point

$$w_0 = \frac{\log M + 1}{\sqrt{H_1}} > 1.$$

Notice that $f(w_0) = M$ and $\|\nabla f(w_0)\| = M\sqrt{H_1}$.

If we choose the step-size $\eta$ of GD larger than $2w_0/M\sqrt{H_1}$, it holds

$$w_1 = w_0 - \eta \nabla f(w_0) < w_0 - (2w_0/M\sqrt{H_1})M\sqrt{H_1} = -w_0.$$

Thus, $w_1$ is negative and further from the optimum (which is 0) compared to $w_0$.

By the structure of the function, we can show that $x_2$ will be even further. Since the function is totally symmetric, the effect of one step of GD starting from $w_1$ is the same as if it would start from $-w_1$. Thus, it suffices to show that $\tilde{w}_1 = -w_1 - \eta \nabla f(-w_1)$ is further from 0 compared to $-w_1$. Since $|w_1| > |w_0|$, it holds $-w_1 > w_0$. We consider the function

$$g(y) = |y - \eta \nabla f(y)| - |y|$$

for $y > \frac{1}{\sqrt{H_1}}$. Then, we have

$$g(y) = \left| y - \eta\sqrt{H_1}\frac{e^{\sqrt{H_1}y}}{e} \right| - |y|.$$

It is simple to see that in the part where this function is positive and $y > \frac{1}{\sqrt{H_1}}$, it is also increasing. Since $g(w_0) > 0$, $w_0 > \frac{1}{\sqrt{H_1}}$ and $-w_1 > w_0$, we have that $g(-w_1) > 0$. This means that $|\tilde{w}_1| > |w_1|$. Using an induction argument, we can show that the iterates of GD under such step-size diverge.

We conclude, that the step-size $\eta$ for our function class must satisfy

$$\eta \le \frac{2w_0}{M\sqrt{H_1}} = \frac{2\log f(w_0) + 2}{f(w_0)H_1}. \tag{63}$$

This step-size bound will be used to derive the lower complexity bounds in all cases.

To establish lower bounds for the general and convex cases, we construct a function that contains a long, flat "runway" region where the gradient is small but non-zero. This forces any first-order method to take many small steps to traverse it.

For a parameter $\delta > 0$ (to be chosen later) and $H_0, H_1 > 0$, we define the following function $f_\delta(w)$;

The function is symmetric, $f_\delta(w) = f_\delta(-w)$, and defined for $x \ge 0$ as:

$$f_\delta(w) = \begin{cases} \frac{H_0}{2}w^2 & \text{if } 0 \le w \le X_1 \\ m(w - X_1) + \delta & \text{if } X_1 < w \le X_2 \\ Ae^{\sqrt{H_1}(w-X_2)} + B & \text{if } w > X_2. \end{cases} \tag{64}$$

To make this function twice differentiable, we choose

$$m = \sqrt{2H_0\delta}$$
$$X_1 = \sqrt{2\delta/H_0}$$
$$X_2 = X_1 + (1-\delta)/m$$
$$A = m/\sqrt{H_1}$$
$$B = 1 - A.$$

$f$ is $(H_0, H_1)$-smooth and its minimum is $f^* = f(0) = 0$.

**Lower bound in the general non-convex case:** We look for a point $w_K$ such that $\|\nabla f(w_K)\| \le \epsilon$. To establish the lower bound, we set the gradient on the runway to be slightly larger than our target $\epsilon$, for instance, $\|\nabla f(w)\| = m = 2\epsilon$.

This choice requires us to set the construction parameter $\delta$ as follows:

$$\sqrt{2H_0\delta} = 2\epsilon \implies \delta = \frac{2\epsilon^2}{H_0}.$$

An algorithm must traverse the linear runway to enter the quadratic bowl, which is the only region where $\|\nabla f(w)\| \le \epsilon$ is achievable.

GD update on the runway is $w_{k+1} = w_k - \eta\nabla f(w_k) = w_k - \eta m$, which implies that

$$w_K = w_0 - \eta K m.$$

Thus, if $w_0 = X_2$ (we start at the beginning of the runway) and $K < \frac{X_2 - X_1}{\eta m}$, then $w_K > X_1$ and we get $\|\nabla f(w_K)\| = 2\epsilon > \epsilon$. Thus, in order to get $\|\nabla f(w_K)\| \le \epsilon$, we need to have

$$K \ge \frac{X_2 - X_1}{\eta m} = \frac{1-\delta}{\eta m^2} = \frac{1 - \frac{2\epsilon^2}{H_0}}{4\eta\epsilon^2}.$$

Choosing $H_0 = 1$ (we can choose any positive constant) and plugging in the upper bound (63) for the step-size $\eta$, we get that $K$ must satisfy

$$K \geq \frac{f(w_0)H_1}{8(\log f(w_0) + 1)} \frac{1 - 2\epsilon^2}{\epsilon^2}.$$

Noticing that $f(w_0) = 1$ and $f^* = 0$, it holds $f(w_0) - f^* = 1$ and we get the desired lower bound:

$$K \geq \frac{H_1(f(w_0) - f^*)}{\log(f(w_0) - f^*) + 1} \frac{f(w_0) - f^* - 2\epsilon^2}{8\epsilon^2}.$$

**Lower bound in the convex case:**  For this scenario, the target accuracy $\epsilon$ directly maps to our construction parameter. We set $\delta = \epsilon$ (64). The function $f_\epsilon(w)$ is convex and is constructed such that the linear runway begins at the point $(X_1, \epsilon)$. An algorithm starting at some point $w_0 = X_2$ where $f(w_0) = 1$ must traverse the runway from $X_2$ down to $X_1$ to achieve the desired accuracy.

On this runway, the gradient has a constant magnitude $m = \sqrt{2H_0\epsilon}$. Similarly as before, we have that if $K < \frac{X_2 - X_1}{\eta m}$, then $w_K > X_1$ and we get $f(w_K) - f^* > \epsilon$. Thus, we need to have

$$K \geq \frac{X_2 - X_1}{\eta m} = \frac{1 - \epsilon}{\eta m^2} = \frac{f(w_0) - f^* - \epsilon}{2\eta H_0 \epsilon}$$

to achieve $\epsilon$ accuracy for the function value.

Substituting, the upper bound (63) for $\eta$ and $H_0 = 1$, we get the desired result.

**Lower bound in the PL case:**  The linear runway construction is not $\mu$-PL. For the third case, we need to construct a different function. We construct a fixed function, independent of $\epsilon$.

Let $C_0 > 0$ and $0 < \mu \leq 1$. We define a fixed connection point $w_c = \sqrt{2C_0/\mu}$. The function is symmetric and defined for $w \geq 0$ as:

$$f(w) = \begin{cases} \frac{\mu}{2}w^2 & \text{if } 0 \leq w \leq w_c \\ Ae^{\sqrt{H_1}(x - w_c)} + B & \text{if } w > w_c \end{cases} \tag{65}$$

where $A = \sqrt{2C_0\mu/H_1}$ and $B = C_0 - A$ are chosen to ensure the function is $C^1$ at $w_c$. This function is $\mu$-strongly convex (thus also $\mu$-PL) and belongs to the class of $(H_0, H_1)$ functions.

Our goal is to find again a point $w_K$ such that $f(w_K) - f^* \leq \epsilon$.

We analyze the performance of GD on the quadratic part of this function, $f(w) = \frac{\mu}{2}w^2$. An algorithm starting at $w_0 = w_c$ will have an initial function value of $f(w_0) = C_0$. The update rule with a fixed step-size $\eta$ is:

$$w_{k+1} = w_k - \eta\nabla f(w_k) = w_k - \eta(\mu w_k) = (1 - \eta\mu)w_k.$$

After $K$ iterations, we have $w_K = (1 - \eta\mu)^K w_0$. We want to find the number of iterations $K$ needed to ensure $f(w_K) \leq \epsilon$.

$$f(w_K) = \frac{\mu}{2}w_K^2 = \frac{\mu}{2}(1 - \eta\mu)^{2K}w_0^2 = f(w_0)(1 - \eta\mu)^{2K} \leq \epsilon.$$

For this to hold, we need

$$f(w_0)(1 - \eta\mu)^{2K} \leq \epsilon \implies (1 - \eta\mu)^{2K} \leq \frac{\epsilon}{f(w_0)}.$$

Taking the logarithm of both sides and using the inequality $\log(1 - z) \leq -z$:

$$2K\log(1 - \eta\mu) \leq \log\left(\frac{\epsilon}{f(w_0)}\right), \quad \text{if } -2K(\eta\mu) \leq -\log\left(\frac{f(w_0)}{\epsilon}\right).$$

Solving for $K$, we get:

*Table 1.* Detailed training details of language models and model configurations for the results in Figure 4. The implementation is based on Ajroldi (2024).

| Model | Configuration | MLP Type | Backbone | Normalization | Position Embeddings | Precision | Dropout |
|---|---|---|---|---|---|---|---|
| 70M | # Layers: 6
# heads: 8
hidden size: 512
seq. length: 1024
batch size: 256
weight decay: 0
cooldown steps: 20 %
grad clip: 1.0
tokens: 1.2B | SwiGLU (Shazeer, 2020) | PreLN transformer (Xiong et al., 2020) with skip connections | RMSnorm (Zhang & Sennrich, 2019) | MLP and Attention layers with variance: $0.02/\sqrt{\# \text{ layers}}$ Other layers: 0.02 std. dev. Biases are always initialized at zero | Mixed precision FP16 | Disabled for both hidden and attention layers |
| 160M | # Layers: 12
# heads: 12
hidden size: 1024
seq. length: 2048
batch size: 256
weight decay: 0.1
cooldown steps: 20 %
grad clip: 1.0
tokens: 1.2B | SwiGLU (Shazeer, 2020) | PreLN transformer (Xiong et al., 2020) with skip connections | RMSnorm (Zhang & Sennrich, 2019) | MLP and Attention layers with variance: $0.02/\sqrt{\# \text{ layers}}$ Other layers: 0.02 std. dev. Biases are always initialized at zero | Mixed precision FP16 | Disabled for both hidden and attention layers |
| 410M | # Layers: 6
# heads: 8
hidden size: 512
seq. length: 2048
batch size: 256
weight decay: 0.1
cooldown steps: 20 %
grad clip: 1.0
tokens: 3.2B | SwiGLU (Shazeer, 2020) | PreLN transformer (Xiong et al., 2020) with skip connections | RMSnorm (Zhang & Sennrich, 2019) | MLP and Attention layers with variance: $0.02/\sqrt{\# \text{ layers}}$ Other layers: 0.02 std. dev. Biases are always initialized at zero | Mixed precision FP16 | Disabled for both hidden and attention layers |

*Table 2.* Detailed training details of image classification and model configurations for the results in Figure 5. The implementation is based on Ajroldi (2025).

| Model | Configuration | MLP Type | Backbone | Normalization | Position Embeddings | Stochastic Depth via DropPath |
|---|---|---|---|---|---|---|
| ViT-Tiny | # Patch size: 4
# heads: 8
Embedding size: 192
# layers: 12
# heads: 3
MLP ratio: 3
Class token: True
Drop path rate: 0.1
grad clip: Null | GELU (Hendrycks & Gimpel, 2016) | PreLN transformer (Xiong et al., 2020) with skip connections | LayerNorm (Ba et al., 2016) | LayerNorm: 1 Biases: 0 Other layers: 0.02 std. dev. | Residual branches are randomly dropped with a linearly increasing drop rate across depth |

$$K \geq \frac{1}{2\eta\mu} \log\left(\frac{f(w_0)}{\epsilon}\right).$$

Substituting the upper bound (63) for the step-size $\eta$ and $f^* = 0$, we get the desired lower complexity bound.

□

# K. Experimental Details and Additional Ablations

## K.1. Experimental Setup

**Language Modeling.** Our training of language models is based on the Plain LM GitHub repository (Ajroldi, 2024) with small changes. The implementation is based on NanoGPT (Karpathy, 2022), and it includes recent improvements such as RMSNorm (Zhang & Sennrich, 2019), Rotational Positional Embeddings (Su et al., 2024), and SwiGLU activations (Shazeer, 2020). All details are reported in Table 1.

**Image Classification.** The implementation of vision tasks is based on the GitHub repository (Ajroldi, 2025) with minor changes. Similarly, we report the training details of ViT training in Table 2. It includes LayerNorm (Ba et al., 2016), GELU activations (Hendrycks & Gimpel, 2016), and drop path.

**Remark K.1.** *The results in Figures 1 and 3 are done with gradient clipping 1.0 and a small LR $10^{-4}$ to make small steps in the loss landscape from the initialization. Such an approach allows for tracking better the smoothness-loss dependency around the initialization.*

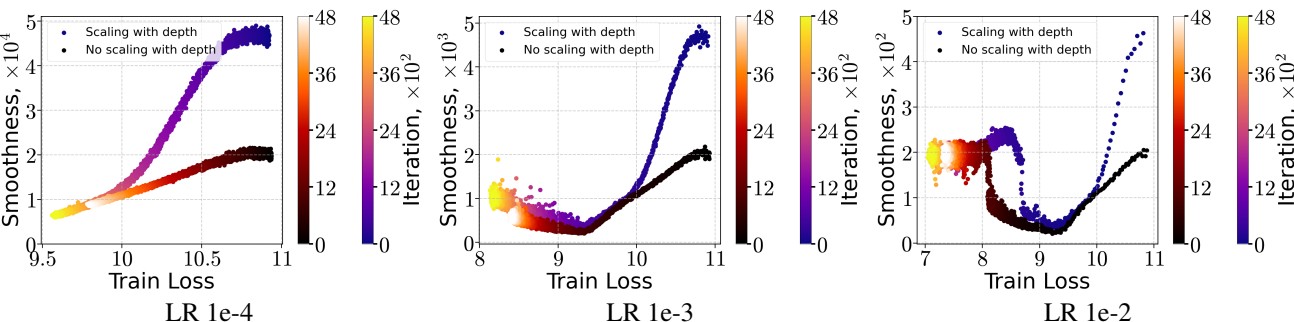

*Figure 6.* Training of 70M model on FineWeb dataset with `SGD` varying fixed learning rate and initialization scheme. Colored points correspond to depth-scaled initialization, while black points correspond to fixed-variance initialization. Color indicates training progress.

### K.2. Effect of Initialization

In this section, we examine how the choice of initialization influences the empirical verification of our condition on the 70M language model trained with clipping 1.0 and a fixed learning rate. We evaluate two initialization strategies. The first follows the approach used in modern GPT-style architectures, where the variance of the weights in the MLP or GLU blocks and in the attention output layers is scaled as $0.02/n\_\text{layer}$, with n˙layer denoting the layer index. We refer to this strategy as "scaling with depth." The second strategy uses a fixed variance of $0.02$, independent of depth, which is the default choice in many implementations. We refer to this strategy as "no scaling with depth." We estimate the smoothness throughout training using the same methodology as in Section 5. We highlight that in this set of experiments, we use gradient clipping to 1, which is a standard training technique used in practice, which allows to restrict the update magnitude and make constant steps in the landscape.

We present the results in Figure 6 varying the fixed learning rate hyperparameter of `SGD`. We observe the following results

- After an initial sharpness reduction (flattening) phase, `SGD` enters a gradual sharpening phase where the sharpness grows. In this regime, our condition does not describe the smoothness well anymore. After the gradual sharpening phase, `SGD` enters `EoS` stage where the sharpness oscillates around 2/LR stability threshold (only observed for LR 1e-2).

- Importantly, the transition from sharpness reduction to the gradual sharpening phase happens at the same loss value regardless of LR choice, which indicates a strong connection between the smoothness and the loss value.

- Our condition describes well the sharpness reduction phase, also observed in (Kalra & Barkeshli, 2024). In contrast to that work, we describe the reduction phase analytically. Using the proposed learning-rate warm-up strategy can avoid instabilities due to high values of the sharpness at the beginning and lead to better final performance.

- "Scaling with depth" strategy initializes the model closer to the origin. This results in a larger initial sharpness in comparison with "no scaling with depth" scheme. This aligns with our theoretical calculations in Proposition 3.1. We hypothesize that GPT-style initialization requires a longer learning-rate warm-up phase due to such high values of the sharpness at the beginning.

### K.3. Additional Results on Verification of the Proposed Condition

#### K.3.1. RESULTS VARYING RANDOM SEED

In this section, we demonstrate that the obtained results in Figures 1 and 3 are consistent when changing the random seed. Random seed changes the initialization of the models, thus leading to exploration of various parts of the landscape. We report the results in Figure 7. Across these runs, we observe an approximately linear decrease in the smoothness proxy with training loss early in training.

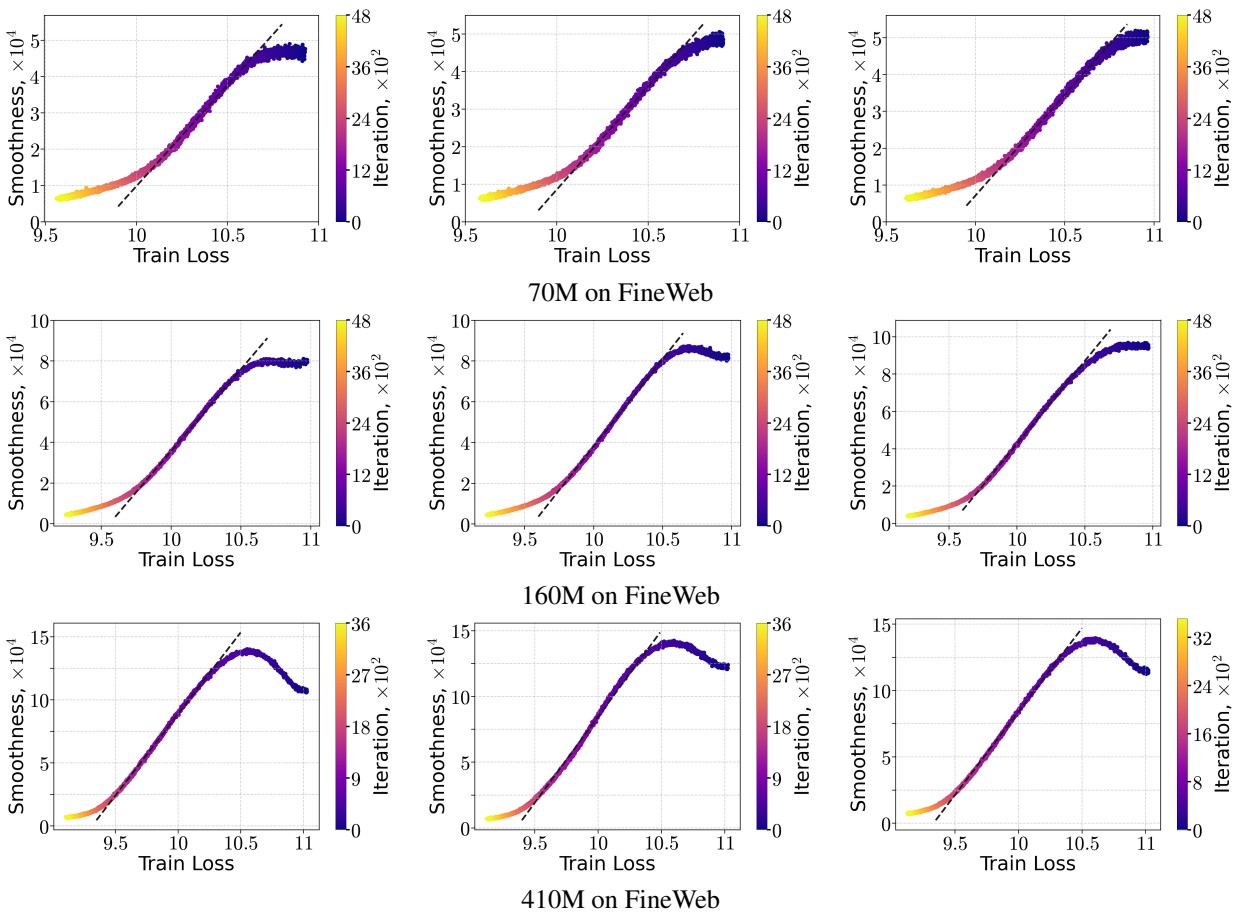

*Figure 7.* Local smoothness approximation versus training loss for language models of varying sizes and random seed on the FineWeb dataset. Models are trained with SGD at a constant learning rate of $10^{-4}$. Each dot represents the estimated local smoothness and stochastic training loss at a given iteration, with color indicating training progress, while the black dashed line shows the best linear fit. For much of early training, the relation is well-approximated by a line, aside from the very initial phase where smoothness behaves differently. This deviation likely arises because the linear fit reflects only an upper bound, suggesting that a more complex functional dependence may be necessary.

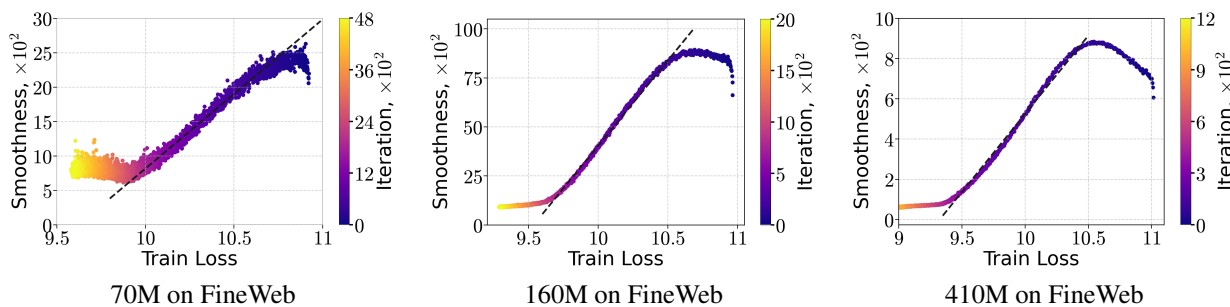

*Figure 8.* Local smoothness approximation versus training loss for language models of varying sizes and random seed on the FineWeb dataset. Models are trained with `Adam` at a constant learning rate of $10^{-7}$. Each dot represents the estimated local smoothness and stochastic training loss at a given iteration, with color indicating training progress, while the black dashed line shows the best linear fit. For much of early training, the relation is well-approximated by a line, aside from the very initial phase where smoothness behaves differently. This deviation likely arises because the linear fit reflects only an upper bound, suggesting that a more complex functional dependence may be necessary.

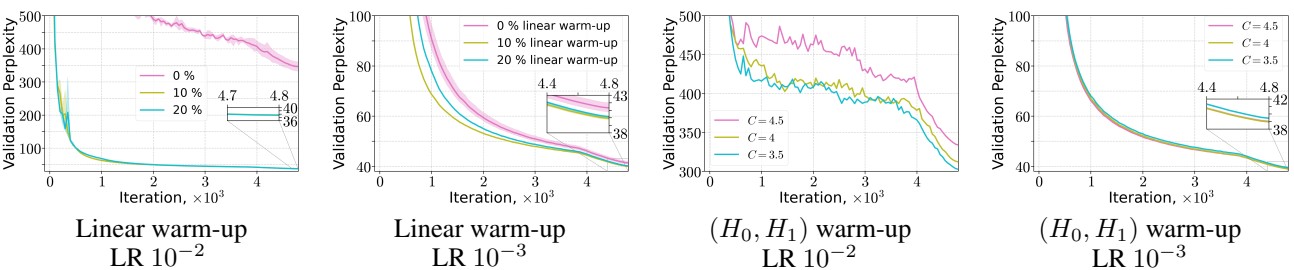

*Figure 9.* Training of 70M language model on FineWeb dataset varying the length of linear warm-up (two left figures) and threshold $C$ of $(H_0, H_1)$ warm-up (two right figures) for the peak learning rate $10^{-2}$ and $10^{-3}$.

### K.3.2. VERIFICATION WITH ADAM

Next, we switch to `Adam` optimizer to verify the proposed $(H_0, H_1)$-smoothness condition. We test the results on language models of size 70M, 160M, and 410M. The results are reported in Figure 8. Similar to the setting in the main body, we use a small constant learning rate $10^{-7}$, which allows moving slowly in the landscape. We observe that `Adam` also demonstrates a linear dependency between local smoothness approximation and train loss. However, we observe that `Adam` stays in this linear decaying part of the landscape for fewer iterations, especially for larger models, than `SGD` does. This might suggest that for `Adam` the warm-up phase should be shorter.

### K.4. Ablation Studies

#### K.4.1. PERFORMANCE VARYING WARM-UP LENGTH

**Language Modeling.** In this section, we investigate how warm-up length influences training. As shown in Figures 9-11, using a 10–20% linear warm-up yields the best validation perplexity, demonstrating that warm-up improves the final performance of the models. We also find that warm-up enables convergence even with relatively large peak learning rates $10^{-2}$ for the 70M model and $3 \cdot 10^{-3}$ for the 160M model, whereas training without warm-up performs significantly worse at these values. Similar trends have been reported by Wortsman et al. (2024). Finally, we observe that the $(H_0, H_1)$ warm-up is less robust to the choice of peak learning rate for the 70M model, resulting in higher validation perplexity. However, once the peak learning rate is properly tuned (within $10^{-3}$–$3 \cdot 10^{-3}$), it becomes less sensitive to the choice of the constant $C$.

**Image Classification with ViT.** Now we turn to the same test, but when training the ViT model on the ImageNet-32 dataset. In contrast to language modeling results, ViT with linear and $(H_0, H_1)$ warm-up strategies demonstrates similar performance. We report the results in Figure 12.

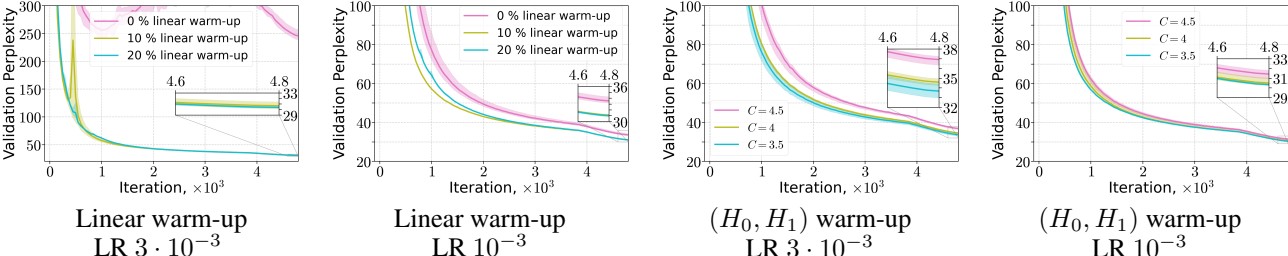

*Figure 10.* Training of 160M language model on FineWeb dataset varying the length of linear warm-up (two left figures) and threshold $C$ of $(H_0, H_1)$ warm-up (two right figures) for the peak learning rate $3 \cdot 10^{-3}$ and $10^{-3}$.

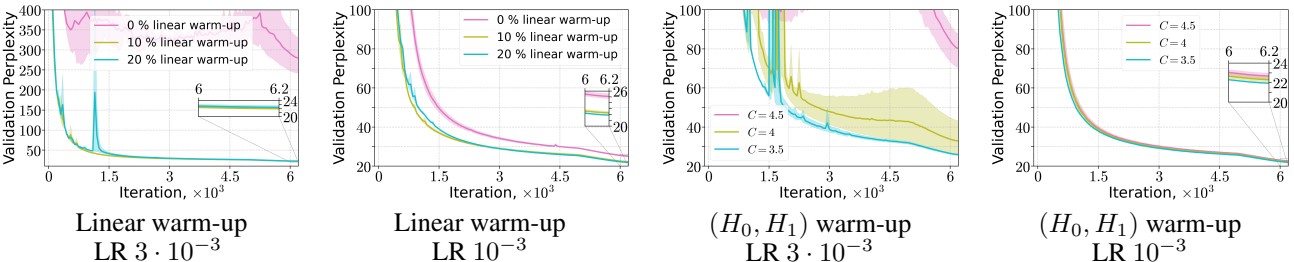

*Figure 11.* Training of 410M language model on FineWeb dataset varying the length of linear warm-up (two left figures) and threshold $C$ of $(H_0, H_1)$ warm-up (two right figures) for the peak learning rate $3 \cdot 10^{-3}$ and $10^{-3}$.

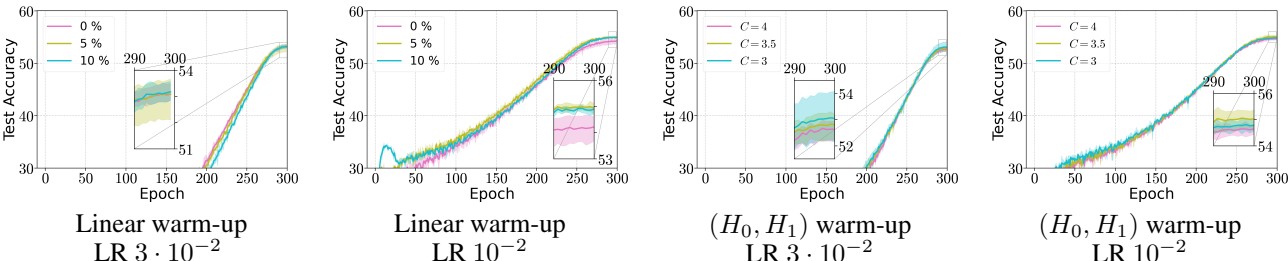

*Figure 12.* Training of ViT model on ImageNet-32 dataset varying the length of linear warm-up (two left figures) and threshold $C$ of $(H_0, H_1)$ warm-up (two right figures) for the peak learning rate $3 \cdot 10^{-2}$ and $10^{-2}$.

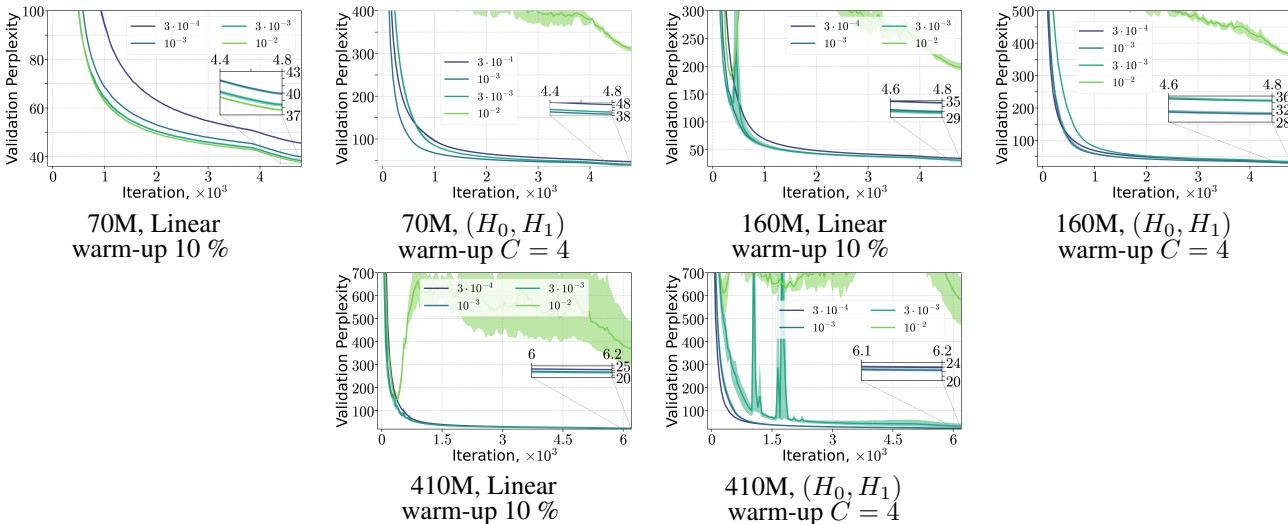

70M, Linear warm-up 10 %

70M, $(H_0, H_1)$ warm-up $C = 4$

160M, Linear warm-up 10 %

160M, $(H_0, H_1)$ warm-up $C = 4$

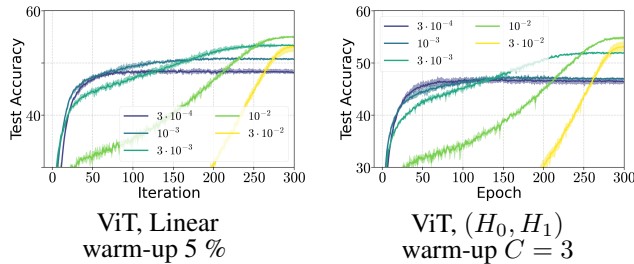

410M, Linear warm-up 10 %

410M, $(H_0, H_1)$ warm-up $C = 4$

*Figure 13.* Training of 70M and 160M language models on FineWeb dataset, varying the peak learning rate with 10 % linear warm-up and $(H_0, H_1)$ warm-up with $C = 4$.

ViT, Linear warm-up 5 %

ViT, $(H_0, H_1)$ warm-up $C = 3$

*Figure 14.* Training of ViT model on ImageNet-32 dataset, varying the peak learning rate with 5 % linear warm-up and $(H_0, H_1)$ warm-up with $C = 3$.

### K.4.2. PERFORMANCE VARYING PEAK LEARNING RATE

**Language Modeling.** We now present performance curves under different peak learning rates for all warm-up strategies: 10% linear warm-up and $(H_0, H_1)$ warm-up with $C = 4$. As shown in Figure 13, smaller models are less sensitive to high peak learning rates when using $(H_0, H_1)$ warm-up. However, for the largest 410M model, even slightly exceeding the optimal peak learning rate produces large spikes with $(H_0, H_1)$ warm-up, though `AdamW` eventually recovers. In contrast, linear warm-up proves more robust to peak learning rate selection.

**Image Classification with ViT.** Now we conduct similar tests as in the previous section. We report the results for three warm-up strategies: 5% linear warm-up and $(H_0, H_1)$ warm-up with $C = 3$. In this case, we observe that both warm-up schedules achieve similar performance; see Figure 14.

