# OpenReview forum: "Why Do We Need Warm-up? A Theoretical Perspective"
_ICML.cc/2026/Conference — ICML 2026 regular_

### Official Review · Reviewer_WTA2 · 2026-03-05

**Soundness:** 3
**Presentation:** 4
**Significance:** 4
**Originality:** 3
**Overall Recommendation:** 6
**Confidence:** 3

**Summary:**

With the goal of understanding the advantages of employing a warm-up learning rate scheduler - a procedure that progressively increases the learning rate at the beginning of training - the paper presents a new convergence analysis for (Stochastic) Gradient Descent (S)GD under $(H_0, H_1)$-smoothness. This assumption relaxes $(L_0, L_1)$-smoothness by upper bounding the spectral norm of the Hessian with a linear function of the optimality gap ($f(w)-f^*$), instead of the norm of the gradient $||\nabla f(w)||$. To motivate the use of this assumption, the authors prove that it holds for linear neural networks under some balancedness conditions and for leaky-ReLU networks under some NTK-like conditions. Beyond these theoretical results on simplified neural networks, the assumption seems to be verified also numerically for realistic neural networks. The paper proposes a warm-up step size scheme based on the $(H_0, H_1)$-smoothness condition. For functions satisfying some relaxed convexity assumptions (PL, the Aiming condition), the convergence analysis of warmed-up GD and a lower bound on the speed of convergence of fixed-step-size GD certify the advantage of the former. Finally, experimental results for transformers using both vision and text datasets are presented. In both settings, the proposed warm-up method is competitive with standard warm-up techniques while giving improved validation accuracy over not using warm-up.

**Compliance With Llm Reviewing Policy:**

Affirmed.

**Final Justification:**

The authors have addressed my comments, I am keeping my original score.

**Key Questions For Authors:**

Mostly questions related to future work:
1. Can the advantages of warm-up for GD be extended to Adam?
2. Can $(H_0, H_1)$-smoothness be proven also for other neural networks beyond those considered in Proposition 3.2 and D.1? Is (piece-wise) linearity the key to prove these results? If so, why are ReLU networks excluded from Proposition D.1 ($b_i>0$)?

**Minor Remark**

3. row 316-319 (left column), "In... schedule": what do the authors mean, that the improvement is only visible in the constants of the rate, or not even there?

**Limitations:**

Despite the authors clarify that the improvement of using warm-up is not theoretically-supported for the non-convex case, they study relaxed convexity conditions to prove this advantage. However, according to some numerical experiments in the literature, these conditions are at best satisfied only locally by neural network training losses, which is partially in contradiction with the fact that warm-up schedule is needed at the beginning of the training procedure (far from a stationary point). Based on this reasoning, it appears that the reply provided by the authors is not the final piece of the warm-up puzzle.

**Strengths And Weaknesses:**

The paper provides a theoretically-supported reply to a long-standing open question in the literature.

The convergence analysis of (S)GD under $(H_0, H_1)$-smoothness is also very relevant, independently from the warm-up question.

**Weakness**

While the theory shows the advantages of using warm-up on (S)GD, the experiments showing that the newly proposed warm-up schedule is applicable in practice are only conducted on Adam and AdamW. Beyond the timely experiments on transformers, it would be valuable to see the effect of the proposed warm-up schedule on other architectures trained via (S)GD.

---

> ### Author Rebuttal · Authors · 2026-03-30
>
> We thank the reviewer for their valuable effort and for the positive evaluation. Below is our feedback:
>
> W1: "While the theory shows the advantages ... are only conducted on Adam and AdamW."
> >Using SGD for theoretical analysis and Adam in experiments is a common practice in the optimization literature [1,2,3]. Moreover, conditions such as $\mu$-PL and Aiming have been shown to hold for the loss landscapes of neural networks [5,6]. Finally, we use Adam in our experiments because warm-up is most commonly employed in Transformer training, where Adam typically outperforms SGD [7,8]. Since our goal is to evaluate theoretically motivated warm-up schedules in realistic settings, we adopt Adam to achieve better empirical performance. Extending convergence results to modern optimizers like Adam and Muon is an exciting direction for future research.
>
> Q1: "Can the advantages of warm-up for GD be extended to Adam?"
> >Our experimental results demonstrate that the $(H_0,H_1)$ warm-up schedule can be successfully coupled with Adam, which leads to competitive performance in practice. We believe it is possible to extend theoretical analysis to Adam for general non-convex functions. However, it is not clear whether the benefits from warm-up can be kept, since our current theory demonstrates only numerical improvement in the non-convex case for gradient descent. Analyzing Adam under $(H_0,H_1)$-smoothness for structured non-convex functions is an open question that we will consider for future work.
>
> Q2: "Can $(H_0,H_1)$-smoothness ... excluded from Proposition D.1 ($b_i>0$)?"
> >In addition to Propositions 3.2 and D.1, we show that the $(H_0,H_1)$ condition holds for two-layer neural networks and one-attention-layer transformers with $\ell_2$ regularization and almost arbitrary non-linearity in Propositions 3.3 and 3.4. As of the case of neural networks without $\ell_2$ regularization, but assuming a balancedness condition (Propositions 3.2 and D.1), piece-wise linearity is indeed useful to obtain an upper bound for the Hessian norm. However, in order to obtain an $(H_0,H_1)$ condition one needs also a lower bound on the loss value. In the case of leaky-ReLU non-linearities, this is done via the bounds at the end of page 29 and beginning of page 30. Such bounds become vacuous if we allow $b_i=0$. This makes the ReLU case tricky, and that's the reason we avoided to talk about it.
>
> Q3: "row 316-319 (left column), "In... schedule": what do the authors mean, that the improvement is only visible in the constants of the rate, or not even there?"
> >We mean that the improvement is only visible in the constants of the learning rate, but not in the asymptotic behavior. We will make this point clearer.
>
> Limitation: "Despite the authors clarify ... appears that the reply provided by the authors is not the final piece of the warm-up puzzle."
> > Thank you for bringing up this point. Note that a sufficiently wide neural networks provably satisfy PL or Aiming conditions [5,6]. In particular, [5] mentions that "Importantly, we show that sufficiently wide neural networks satisfy the PL∗ condition around their initialization point, thus guaranteeing convergence". Same situation in [6], Theorems 3.1-3.3. Therefore, our theoretical analysis is applicable in such cases. We acknowledge that the amount of overparametrization required to show the validity of these assumptions might be not realistic. However, we still believe that warm-up schedule should not help for arbitrary non-convex functions, and that GD with warm-up schedule can get stuck at a sharp minima around initialization. We will incorporate these clarifications into the revision.
>
>  [1] Defazio, Aaron, and Konstantin Mishchenko. "Learning-rate-free learning by d-adaptation." International conference on machine learning. PMLR, 2023.
>
> [2] Mishchenko, Konstantin, and Aaron Defazio. "Prodigy: An expeditiously adaptive parameter-free learner." arXiv preprint arXiv:2306.06101 (2023).
>
> [3] Defazio, Aaron, et al. "The road less scheduled." Advances in Neural Information Processing Systems 37 (2024): 9974-10007.
>
> [4] Semenov, Andrei, Matteo Pagliardini, and Martin Jaggi. "Benchmarking optimizers for large language model pretraining." arXiv preprint arXiv:2509.01440 (2025).
>
> [5] Liu, Chaoyue, et al. "Aiming towards the minimizers: fast convergence of sgd for overparametrized problems." Advances in neural information processing systems 36 (2023): 60748-60767.
>
> [6] Liu, Chaoyue, Libin Zhu, and Mikhail Belkin. "Loss landscapes and optimization in over-parameterized non-linear systems and neural networks." Applied and Computational Harmonic Analysis 59 (2022): 85-116.
>
> [7] Srećković, Teodora, Jonas Geiping, and Antonio Orvieto. "Is your batch size the problem? Revisiting the Adam-SGD gap in language modeling." arXiv preprint arXiv:2506.12543 (2025).
>
> [8] Zhang, Yushun, et al. "Why transformers need adam: A hessian perspective." Advances in neural information processing systems 37 (2024): 131786-131823.

---

> > ### Author Rebuttal · Reviewer_WTA2 · 2026-04-01
> >
> > I acknowledge the rebuttal.

---

### Official Review · Reviewer_jEG9 · 2026-03-12

**Soundness:** 4
**Presentation:** 3
**Significance:** 4
**Originality:** 3
**Overall Recommendation:** 5
**Confidence:** 3

**Summary:**

This paper proposes the **(H₀, H₁)-smoothness** condition to provide a theoretical explanation for learning-rate warm-up. The condition bounds the local curvature as a linear function of the loss suboptimality, which decreases during the early stage of training and thus explains why gradually increasing the learning rate (warm-up) can be effective. The authors not only show faster convergence under this smoothness assumption, but also verify the validity of the assumption both theoretically and empirically across different model architectures. In addition, they demonstrate that the theoretically motivated learning-rate schedule exhibits performance consistent with standard warm-up strategies in practice.

**Compliance With Llm Reviewing Policy:**

Affirmed.

**Final Justification:**

The authors have addressed my concerns, and I will maintain my score.

**Key Questions For Authors:**

1. From Equation (1), it appears that as long as the loss decreases, the learning rate should increase. However, this behavior is clearly not expected in the middle and later stages of training. What causes this apparent mismatch?

2. The authors state that the paper focuses on the first phase of training. But how early does this phase actually correspond to? Is there a principled way to characterize the boundary of this stage?

3. In Figure 3, the tuned optimal (H₀, H₁) warm-up seems to keep increasing the learning rate for roughly the first 80% of training, and it does not even reach the same peak learning rate as the baseline. I suspect this may affect the fairness of the comparison.

**Limitations:**

yes

**Strengths And Weaknesses:**

**Strengths:**
The paper is theoretically novel and fairly solid. It introduces a reasonable new smoothness definition and provides an intuitive explanation of how it affects convergence speed and why it leads to a gradually increasing learning-rate schedule. In addition, the authors argue from multiple perspectives that this smoothness assumption is not difficult to satisfy in practice. Overall, the exposition is logical and clear, making the arguments convincing.

**Weaknesses:**
The paper mainly focuses on the early stage of training, but it is not very clear how “early” this stage actually is. Moreover, the experimental section is somewhat weaker. For example, the carefully designed learning-rate schedule does not outperform the heuristic linear warm-up. As a result, the practical impact of the work may be limited.

---

> ### Author Rebuttal · Authors · 2026-03-30
>
> We thank the reviewer for their valuable effort and for finding our paper novel and solid. Below is our feedback to their concerns:
>
> W1: "The paper mainly focuses on the early stage of training, but it is not very clear how “early” this stage actually is."
> >This is a good question. Characterizing the length of the warm-up period is indeed a challenging problem and requires further investigation, which we plan to pursue in future work. Nevertheless, we believe that the warm-up phase constitutes a meaningful portion of training. In practice, it typically accounts for about 5–10\% of the total training duration. Additionally, in our experiments (Figure 1), the sharpness reduction phase spans a substantial loss decrease starting from 10.8 to approximately 9.3, indicating that it captures a significant part of the training period.
>
> W2: "Moreover, the experimental section is somewhat weaker. For example, the carefully designed learning-rate schedule does not outperform the heuristic linear warm-up. As a result, the practical impact of the work may be limited."
> >The main goal of our work is to provide a theoretical characterization of the warm-up phase through the $(H_0,H_1)$-smoothness condition, and to validate this condition across different architectures, including linear models and Transformers, both in theory and practice. The experimental results are intended to demonstrate the competitiveness of theoretically motivated warm-up schedules. In this context, the fact that the $(H_0,H_1)$-based schedule matches the performance of the widely used linear warm-up strategy, that has been extensively tested in practice for many years, should not be seen as a limitation, but rather as evidence supporting the validity of our theoretical framework.
>
> Q1: "From Equation (1), it appears that as long as the loss decreases, the learning rate should increase. However, this behavior is clearly not expected in the middle and later stages of training. What causes this apparent mismatch?"
> >The learning rate schedule in Equation (1) is valid only when the $(H_0,H_1)$-smoothness condition (Definition 3.1) is tight. This happens only in the initial phase of training, where the sharpness decreases. In later stages of training, $(H_0,H_1)$-smoothness does not reliably characterize the loss landscape and one should not use the learning rate of equation (1). Therefore, our analysis applies exactly in the initial sharpness reduction phase (which is where warm-up happens), but does not apply in the later stages of progressive sharpening and edge of stability. Our iteration complexity results also apply to this initial stage, i.e. optimizers with learning rate as in equation (1) reach the turning point between sharpness reduction and progressive sharpening faster than with a fixed step-size.
>
> Q2: "The authors state that the paper focuses on the first phase of training. But how early does this phase actually correspond to? Is there a principled way to characterize the boundary of this stage?"
> >We refer the Reviewer to our response to W1.
>
> Q3: "In Figure 3, the tuned optimal $(H_0, H_1)$ warm-up seems to keep increasing the learning rate for roughly the first 80% of training, and it does not even reach the same peak learning rate as the baseline. I suspect this may affect the fairness of the comparison."
> > In Figure 3 (right), we demonstrate the effective learning rate when training a 70M transformer model. The $(H_0, H_1)$ warm-up schedule with $C=3.5$ the reviewer refers to results in sub-optimal performance. The best performance is achieved with $C=4$, which we plot in Figure 3 (left). This result demonstrates that the choice $C=3.5$ is too optimistic, and warm-up length should be shorter from the loss perspective for a better final performance.

---

> > ### Author Rebuttal · Reviewer_jEG9 · 2026-04-04
> >
> > Thank you for the authors’ response. I will maintain my positive assessment.

---

### Official Review · Reviewer_n25f · 2026-03-17

**Soundness:** 4
**Presentation:** 4
**Significance:** 4
**Originality:** 3
**Overall Recommendation:** 4
**Confidence:** 4

**Summary:**

This paper proposes an analytical framework for understanding the role of learning rate warm-up in training deep neural networks. The core idea is to replace classical smoothness assumptions with an $(H_0, H_1)$-type condition that bounds the Hessian in terms of function-value suboptimality, thereby capturing the geometry of the loss landscape during the early phase of training. Under this condition, the authors derive step size schedules that naturally exhibit a warm-up behavior and establish convergence guarantees for gradient-based methods. The paper further supports the theory with empirical evidence across a range of models and tasks, showing that the proposed condition approximately holds in practice and that the resulting schedules perform competitively with standard heuristics.

Overall, I found the paper interesting and well-executed. I am generally quite fond of this line of work that aims to identify the "right" assumptions under which meaningful theoretical analysis becomes possible, while also making an effort to empirically validate whether those assumptions are reflected in modern deep learning practice.

**Compliance With Llm Reviewing Policy:**

Affirmed.

**Key Questions For Authors:**

.

**Limitations:**

.

**Strengths And Weaknesses:**

I find the approach of introducing a refined condition that depends on the current suboptimality, rather than attempting to analyze warm-up under classical global smoothness assumptions, compelling. And I am fond of this trend in optimization and learning theory: tailoring assumptions to the regimes that actually arise in practice, rather than insisting on overly rigid classical conditions. I also like the experimental validation of the assumption and the subsequent analyses of the warm-up schedule.

At the same time, there are some weaknesses. Namely the rubric of proposing a refined structural assumption that enables tractable analysis and then validating it empirically, is not entirely new. Also, the optimization-theoretic techniques used in the proof are not exceptionally novel. Relatedly, the empirical results, while solid, mostly demonstrate competitiveness rather than clear superiority over standard warm-up schedules.

Second, the theoretical results rely on additional conditions such as variants of PL or interpolation-type assumptions and are primarily developed for gradient methods, whereas the experiments focus on adaptive optimizers such as Adam or AdamW. This gap is acknowledged, but this slightly weakens the results.

---

> ### Author Rebuttal · Authors · 2026-03-30
>
> We thank the reviewer for their time and thorough critique. We are pleased to see that they found our approach compelling and appreciate the move beyond classical assumptions. Below is our feedback to their concerns:
>
> W1: "The rubric of proposing a refined structural assumption that enables tractable analysis and then validating it empirically, is not entirely new."
> >Such an approach is indeed common in the literature; however, in the context of warm-up, existing studies are mostly empirical. Our main contribution is identifying exactly the right structural condition that is simultaneously 1) realistic enough for deep models 2) amenable to optimization analysis, and 3) can explain the need for learning rate warm-up. We believe that combining these three elements presents a significant degree of novelty.
>
> W2: "The optimization-theoretic techniques used in the proof are not exceptionally novel"
> > Although some of our theoretical proving techniques are based on prior works (e.g., Zhang et al., 2020), we provide new mathematical results in Lemmas 5 and 6, that significantly differ from the analysis under the $(L_0,L_1)$-smoothness condition. Next, we provide lower bounds under $(H_0,H_1)$-smoothness and $\mu$-PL conditions, which was not done before. Finally, our theoretical analysis also covers the convergence under Aiming condition. While not the primary focus of novelty, carefully integrating all components in the setting of $(H_0,H_1)$-smoothness is still a meaningful and non-trivial task.
>
> W3: "Relatedly, the empirical results, while solid, mostly demonstrate competitiveness rather than clear superiority over standard warm-up schedules":
> >The primary goal of our experimental section is not to design superior practical warm-up schedules, but rather to demonstrate the competitiveness of theoretically motivated ones. We believe that the comparable performance between the schedule derived from our theory and those commonly used in practice provides strong justification and supports the validity of our approach.
>
> W4: "The theoretical results rely on additional conditions such as variants of PL or interpolation-type assumptions and are primarily developed for gradient methods, whereas the experiments focus on adaptive optimizers such as Adam or AdamW"
> >Using SGD for theoretical analysis and Adam in experiments is a common practice in the optimization literature [1,2,3]. Moreover, conditions such as $\mu$-PL and Aiming have been shown to hold for the loss landscapes of neural networks [5,6]. Finally, we use Adam in our experiments because warm-up is most commonly employed in Transformer training, where Adam typically outperforms SGD [7,8]. Since our goal is to evaluate theoretically motivated warm-up schedules in realistic settings, we adopt Adam to achieve better empirical performance. Extending convergence results to modern optimizers like Adam and Muon is an exciting direction for future research.
>
>  [1] Defazio, Aaron, and Konstantin Mishchenko. "Learning-rate-free learning by d-adaptation." International conference on machine learning. PMLR, 2023.
>
>  [2] Mishchenko, Konstantin, and Aaron Defazio. "Prodigy: An expeditiously adaptive parameter-free learner." arXiv preprint arXiv:2306.06101 (2023).
>
>  [3] Defazio, Aaron, et al. "The road less scheduled." Advances in Neural Information Processing Systems 37 (2024): 9974-10007.
>
> [4] Semenov, Andrei, Matteo Pagliardini, and Martin Jaggi. "Benchmarking optimizers for large language model pretraining." arXiv preprint arXiv:2509.01440 (2025).
>
> [5] Liu, Chaoyue, et al. "Aiming towards the minimizers: fast convergence of sgd for overparametrized problems." Advances in neural information processing systems 36 (2023): 60748-60767.
>
> [6] Liu, Chaoyue, Libin Zhu, and Mikhail Belkin. "Loss landscapes and optimization in over-parameterized non-linear systems and neural networks." Applied and Computational Harmonic Analysis 59 (2022): 85-116.
>
> [7] Srećković, Teodora, Jonas Geiping, and Antonio Orvieto. "Is your batch size the problem? Revisiting the Adam-SGD gap in language modeling." arXiv preprint arXiv:2506.12543 (2025).
>
> [8] Zhang, Yushun, et al. "Why transformers need adam: A hessian perspective." Advances in neural information processing systems 37 (2024): 131786-131823.

---

> > ### Author Rebuttal · Reviewer_n25f · 2026-04-06
> >
> > Thank you for the response. I will maintain my score, leaning towards acceptance.

---

### Decision · Program_Chairs · 2026-04-30

**Decision:**

Accept (regular)

**Comment:**

The paper provides a principled explanation for why warm-up improves training. By relying on a generalization of the (L_0,L_1)-smoothness assumption, it shows, both theoretically and empirically, that this condition is satisfied by common neural architectures and accurately captures the curvature of the optimization landscape early in training. The proposed learning rate yields provably faster convergence guarantees than using a fixed learning rate.

All reviewers find the results of this work interesting and are positive about the paper's acceptance. They give scores of 4, 5, and 6. The authors provided more details during the rebuttal and discussion period, and all reviewers agreed that their concerns were fully resolved.

For the camera-ready, the AC advises the authors to incorporate the feedback received during the review process into the revised manuscript. The paper is well written and has clear contributions that could be of interest to the general optimization and ML communities.  I suggest acceptance of this work.